# TabArena: A Living Benchmark for Machine Learning on Tabular Data

Nick Erickson[1]    Lennart Purucker[2]    Andrej Tschalzev[3]    David Holzmüller[4,5,6]
Prateek Mutalik Desai[1]    David Salinas[8,2]    Frank Hutter[7,8,2]

[1]Amazon Web Services    [2]University of Freiburg    [3]University of Mannheim    [4]INRIA Paris
[5]Ecole Normale Supérieure    [6]PSL Research University    [7]Prior Labs    [8]ELLIS Institute Tübingen
`mail@tabarena.ai`

## Abstract

With the growing popularity of deep learning and foundation models for tabular data, the need for standardized and reliable benchmarks is higher than ever. However, current benchmarks are static. Their design is not updated even if flaws are discovered, model versions are updated, or new models are released. To address this, we introduce TabArena, the first continuously maintained living tabular benchmarking system. To launch TabArena, we manually curate a representative collection of datasets and well-implemented models, conduct a large-scale benchmarking study to initialize a public leaderboard, and assemble a team of experienced maintainers. Our results highlight the influence of validation method and ensembling of hyperparameter configurations to benchmark models at their full potential. While gradient-boosted trees are still strong contenders on practical tabular datasets, we observe that deep learning methods have caught up under larger time budgets with ensembling. At the same time, foundation models excel on smaller datasets. Finally, we show that ensembles across models advance the state-of-the-art in tabular machine learning. We observe that some deep learning models are overrepresented in cross-model ensembles due to validation set overfitting, and we encourage model developers to address this issue. We launch TabArena with a public leaderboard, reproducible code, and maintenance protocols to create a living benchmark available at `https://tabarena.ai`.

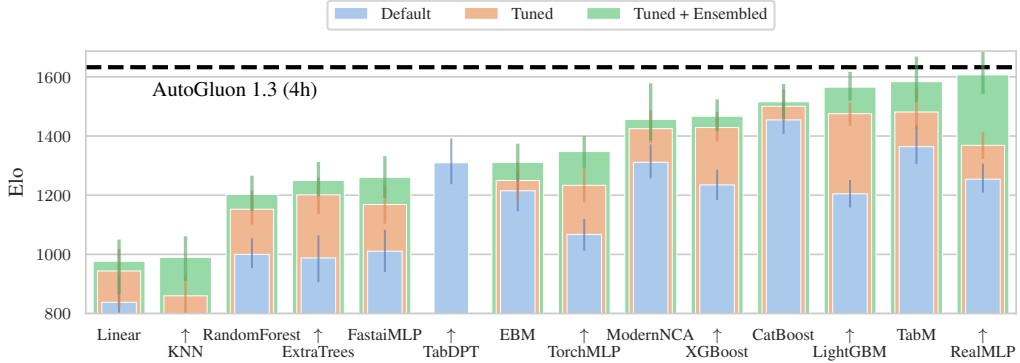

Figure 1: **TabArena-v0.1 Leaderboard.** We evaluate models under default parameters, tuning, and weighted ensembling [1] of hyperparameters. Since TabICL and TabPFNv2 are not applicable to all datasets, we evaluate them on subsets of the benchmark in Figure 4 .

# 1 Introduction

Benchmarking tabular machine learning models is an arduous and error-prone process. With the rise of deep learning and foundation models for tabular data [2, 3, 4, 5, 6], benchmarking has become even more challenging for researchers and practitioners alike. While several benchmarks have been proposed in recent years, there is increasing skepticism towards data curation and evaluation protocols utilized [7, 8, 9]. Most importantly, many datasets used in benchmarks are outdated, contain problematic licenses, do not represent real tabular data tasks, or are biased through data leaks [7, 10]. Despite the growing awareness of such issues, benchmarks are rarely maintained after publication. Issues uncovered in follow-up studies are not addressed, and baselines for the state-of-the-art are stuck in time. Consequently, follow-up benchmarks reproduce shortcomings of prior work and do not compare to the actual state-of-the-art [11].

To address these issues, we argue for a paradigm shift from the currently used static benchmarks to a sustainable, living benchmarking system treated as software that is versioned, professionally maintained, and gradually improved by the community as an open-source project. With this goal in mind, we introduce `TabArena`.

`TabArena` is a living benchmarking system that makes benchmarking tabular machine learning models a reliable experience. TabArena is realized through **our four main contributions**:

- ➢ We investigate 1053 datasets used in tabular data research and carefully, manually curate a set of 51 datasets out of these, representing real-world tabular data tasks.
- ➢ We curate 16 tabular machine learning models, including 3 tabular foundation models, and run large-scale experiments in well-tested modeling pipelines used in practice. In total, we trained ~25 000 000 instances of models across all our experiments.
- ➢ We instantiate a public leaderboard available on `tabarena.ai`, release precomputed results for fast comparisons, and provide reproducible code to benchmark new methods.
- ➢ We assemble a team of maintainers from different institutions with experience in maintaining open-source initiatives to keep the living benchmark up to date.

In this paper, we detail our curation protocols for building a sophisticated living benchmark and investigate the results of `TabArena` version 0.1, representing our initialization of the leaderboard.

**TabArena-v0.1 Focus Statement.** `TabArena` focuses on evaluating predictive machine learning models for tabular data. Our long-term vision is to make `TabArena` representative for all use cases of tabular data. For `TabArena`-v0.1, we initialize the benchmark focusing on the most common type of tabular machine learning problem: **Tabular classification and regression for independent and identically distributed (IID) data, spanning the small to medium data regime**. We explicitly leave for future work use cases such as non-IID data (e.g., temporal dependencies, subject groups, or distribution shifts); few-shot predictions, or very small data (e.g., less than 500 training samples); large data (e.g., more than 250 000 training samples); or other machine learning tasks such as clustering, anomaly detection, subgroup discovery, or survival analysis.

**Our results demonstrate:** (**A**) The best performance for individual tabular machine learning models is generally achieved by post-hoc ensembling tuned hyperparameter configurations; (**B**) With tuning and ensembling, the best deep learning methods are equal to or better than gradient-boosted decision trees; (**C**) Tabular foundation models dominate for small data with strong in-context learning performance even without tuning; (**D**) Ensemble pipelines leveraging various models are the state-of-the-art for tabular data, but the best individual models do not contribute equally to the ensemble pipeline.

# 2 TabArena

`TabArena` is a living benchmark because of our *protocols*, which govern the curation of (**2.1**) models and hyperparameter optimization, (**2.2**) datasets, and (**2.3**) evaluation design. Through continuous application and refinement of these protocols, we ensure `TabArena` remains current and maintained.

## 2.1 Models and Hyperparameter Optimization Protocol

`TabArena` is implemented as an extendable platform to support a wide range of machine learning models on tabular data. For instantiating `TabArena`, we curate 14 state-of-the-art (foundation) models

and two simple baselines. `TabArena` is created as a platform to benchmark each model to its full potential. Therefore, every included model represents a well-known baseline or was implemented in dialogue with the authors. Furthermore, we only run models on datasets within their restrictions to represent them fairly. This only affects TabPFNv2, which is restricted to datasets with up to $10,000$ training samples, $500$ features, and $10$ classes for classification tasks, and TabICL, which is constrained to classification tasks with up to $100,000$ training samples and $500$ features.

`TabArena` models are powered by three components: **(1)** implementation in a well-tested modeling framework used in real-world applications; **(2)** curated hyperparameter optimization protocols; **(3)** improved validation and ensembling strategies, including ensembling over instances of a single model class. Table 1 provides an overview of the models benchmarked in `TabArena`-v0.1.

Table 1: **TabArena-v0.1 Models.** We show all models included in our initialization of `TabArena`, the source of the search space, and a short version of the name. Moreover, we specify the model types: tree-based (🌳), neural network (❊), pretrained foundation models (◖), and baseline (✏).

| Model | Short Name | Search Space | Type |
|---|---|---|---|
| Random Forests [12] | RandomForest | Prior Work + Us | 🌳 |
| Extremely Randomized Trees [13] | ExtraTrees | Prior Work + Us | 🌳 |
| XGBoost [14] | XGBoost | Prior Work + Us | 🌳 |
| LightGBM [15] | LightGBM | Prior Work + Us | 🌳 |
| CatBoost [16] | CatBoost | Prior Work + Us | 🌳 |
| Explainable Boosting Machine [17, 18] | EBM | Authors | 🌳 |
| FastAI MLP [19] | FastaiMLP | Authors | ❊ |
| Torch MLP [19] | TorchMLP | Authors | ❊ |
| RealMLP [20] | RealMLP | Authors | ❊ |
| TabM$_{\text{mini}}^{\dagger}$ [9] | TabM | Authors | ❊ |
| ModernNCA [21] | ModernNCA | Authors | ❊ |
| TabPFNv2 [5] | TabPFNv2 | Authors | ◖ |
| TabICL [22] | TabICL | – | ◖ |
| TabDPT [23] | TabDPT | – | ◖ |
| Linear / Logistic Regression | Linear | Prior Work + Us | ✏ |
| K-Nearest Neighbors | KNN | Prior Work + Us | ✏ |

**Implementation Framework.** For implementing models, we rely on functionalities from AutoGluon [19], an established machine learning framework used in practical applications. Each model is implemented within the standardized `AbstractModel` framework, which aligns with the scikit-learn [24] API, and includes: **(1)** model-agnostic preprocessing, **(2)** support for (inner) cross-validation with ensembling, **(3)** hyperparameter optimization, **(4)** evaluation metrics, **(5)** fold-wise training parallelization, **(6)** (customizable) model-specific preprocessing pipeline, **(7)** (customizable) early stopping and validation logic, and **(8)** unit tests. As a result, any model implemented in `TabArena` can be readily deployed for real-world use cases or within predictive machine learning systems. Moreover, the pipeline logic encompassing models within `TabArena` is implemented in a tested framework regularly used in real-world applications. Appendix C.1 summarizes further implementation details, and Appendix E.2 includes a detailed protocol for contributing models.

**Cross-validation and Post-hoc Ensembles.** As can be seen in various Kaggle competitions and academic studies, cf. [8, 20, 25, 26], for most datasets, peak performance requires ensembling strategies. Therefore, we default to using 8-fold cross-validation (with class-wise stratification for classification) and then employ cross-validation ensembles [27]; which we describe in detail in Appendix C.3. For all foundation models, we refit on training and validation data instead of using cross-validation ensembles, following recommendations from the authors of TabPFN and TabICL. In addition, we evaluate each tunable model using post-hoc ensembling [1] of different hyperparameter configurations, denoted as `Tuned + Ensembled`; further details are provided in Appendix C.4.

**Hyperparameter Optimization.** For each model, we curate a strong hyperparameter search space; for full details, see Appendix C.2. Where possible, we started with the search spaces from the original paper and finalized them in dialogue with the models' authors. Otherwise, we curated

Table 2: **Comparison of Tabular Benchmarks.** We systematically compare prior tabular benchmark studies across six characteristics. **Inner and outer splits:** the number of splits used for inner or outer validation: 1 for holdout validation; - if the benchmark does not specify; and any other number specifies the total number of splits from (repeated) cross-validation. If a set is given, the benchmark uses different splits for different datasets. **Ensembling:** Whether the benchmark studies ensembling of configurations for individual models (✓), uses any other ensembling ((✓)), or uses no ensembling at all (✗). **Manual Curation:** Whether the benchmark filters datasets based on criteria beyond simple automation, (✓) or not (✗). **Datasets remaining:** the number of datasets remaining after filtering by our criteria. **Results available:** Whether the benchmark shares no re-usable results (✗), only metric results ((✓)), or metric results and predictions (✓). **HPO Limit:** How hyperparameter optimization was limited in the number of configurations and/or hours.

| Benchmark | #splits | | Ensembling | Manual curation | #datasets remaining | Results available | HPO Limit | |
|---|---|---|---|---|---|---|---|---|
| | inner | outer | | | | | #confs. | #hours |
| Bischl et al. [28, 29] | - | 10 | ✗ | ✓ | 9/72 | (✓) | - | - |
| Gorishniy et al. [30] | 1 | 1 | (✓) | ✓ | 1/11 | ✗ | 100 | 6 |
| Shwartz-Ziv and Armon [31] | 1 | {1, 3} | (✓) | ✗ | 1/11 | ✗ | 1000 | - |
| Grinsztajn et al. [32] | 1 | {1, 2, 3, 5} | ✗ | ✓ | 12/47 | (✓) | 400 | - |
| McElfresh et al. [33] | 1 | 10 | ✗ | ✗ | 13/196 | (✓) | 30 | 10 |
| Fischer et al. [34] | {1, 3, 10} | {1, 10, 100} | ✗ | ✓ | 8/35 | (✓) | {-, 500} | - |
| Gijsbers et al. [35] | - | 10 | (✓) | ✓ | 15/104 | (✓) | - | 4 |
| Kohli et al. [7] | 1 | 1 | ✗ | ✓ | 17/187 | ✗ | 100 | {3, -} |
| Tschalzev et al. [8] | 10 | 1 | (✓) | ✓ | 1/10 | ✗ | 100 | - |
| Holzmüller et al. [20] | 1 | 10 | (✓) | ✓ | 10/118 | ✓ | 50 | - |
| Ye et al. [36] | 1 | 1 | ✗ | ✗ | 39/300 | (✓) | 100 | - |
| Rubachev et al. [10] | 1 | 1 | (✓) | ✓ | 0/8 | (✓) | 100 | - |
| Salinas and Erickson [37] | 8 | 3 | ✓ | ✗ | 19/200 | ✓ | 200 | 200 |
| **TabArena (Ours)** | 8 | {9, 30} | ✓ | ✓ | 51/51 | ✓ | 200 | 200 |

search spaces from prior work. We evaluate 1 default and a fixed set of 200 randomly-sampled hyperparameter configurations for all models, except for TabICL and TabDPT. TabICL and TabDPT do not specify hyperparameter optimization (HPO) in the original paper and implementation; thus, we restrict ourselves to evaluating only their default performance. Each hyperparameter configuration is validated using 8-fold (inner) cross-validation. We use the score of this (inner) cross-validation to select the best hyperparameter configuration.

For practical reasons, we restrict the time to evaluate one configuration on one train split of a dataset to 1 hour. Our analysis in Appendix A.2 shows that this limit rarely takes effect.

In a living benchmark, we expect users with different hardware to submit to the leaderboard. Thus, we do not constrain the hardware used to evaluate a configuration. We log the hardware used during benchmarking to enable analysis of the impact of computing power. Our recommended hardware for evaluating a configuration is 8 CPU cores, 32 GB of RAM, and 100 GB of disk space. Furthermore, we recommend using 1 GPU with 48 GB VRAM for GPU-scaling models, such as foundation models.

## 2.2 Datasets Protocol

Many existing benchmarks were curated using semi-automated procedures to collect datasets according to simple characteristics, often to obtain as many datasets as possible. In contrast, we reject the notion of automatically collecting datasets without any sanity check. Instead, we focus on carefully, manually curating a representative collection of datasets. Although some previous benchmarks manually curated data, most of the included datasets do not meet the criteria of IID predictive tabular datasets, as seen in Table 2. Our work indicates a turning point with the most extensive and conscientious manual curation effort for machine learning on IID tabular data so far. We define criteria for data selection according to our focus statement and filter 1053 datasets used in 14 prior benchmarks accordingly. Figure 2 describes our selection process. Notably, only the deduplication step and size filters can be automated. Faithfully applying the other criteria requires manual human effort per-dataset, demonstrating the downsides of automated data curation procedures.

**Dataset Selection Criteria.** We comprehensively describe our selection criteria in Appendix B.1. In short, we selected datasets that fulfilled the following requirements: (**1**) The dataset and its predictive machine learning task are unique within our benchmark; (**2**) The dataset is IID, that is, a random split is appropriate for the underlying original task; (**3**) The dataset is not from a non-tabular

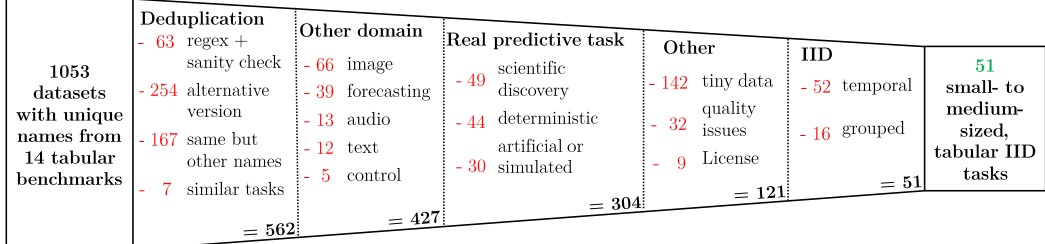

Figure 2: **Data Curation Results.** The figure shows why and how many datasets we filter based on our criteria. We filter datasets that are duplicates, not from a tabular domain, not a real predictive task, tiny, have quality or license issues, and are not IID.

modality, such as images, where it is unclear whether tabular machine learning is a reasonable alternative to domain-specific methods; (**4**) The dataset stems from a real random distribution, and is not generated, e.g., from a deterministic function; (**5**) The dataset was published explicitly for a predictive modeling task in a real-world application; (**6**) The dataset is small-to-medium-sized, i.e., it has at least $500$ and at most $250\,000$ train samples; (**7**) We can use a version of the dataset without pre-applied problematic preprocessing, such as irreversible data leaks; (**8**) The dataset was originally published with a license allowing for scientific usage; (**9**) The dataset and its structured metadata can be automatically downloaded via a public API, or we are allowed to upload the dataset to a public API; (**10**) The dataset and its predictive task do not raise (obvious) ethical concerns.

As several criteria involve subjective judgment and human interpretation, we publicly share our curation insights at `tabarena.ai/dataset-curation`, including per-dataset notes detailing our observations, identified characteristics, and final assessments. To enhance the quality of `TabArena`'s datasets, we actively encourage the community to review and critique our evaluations.

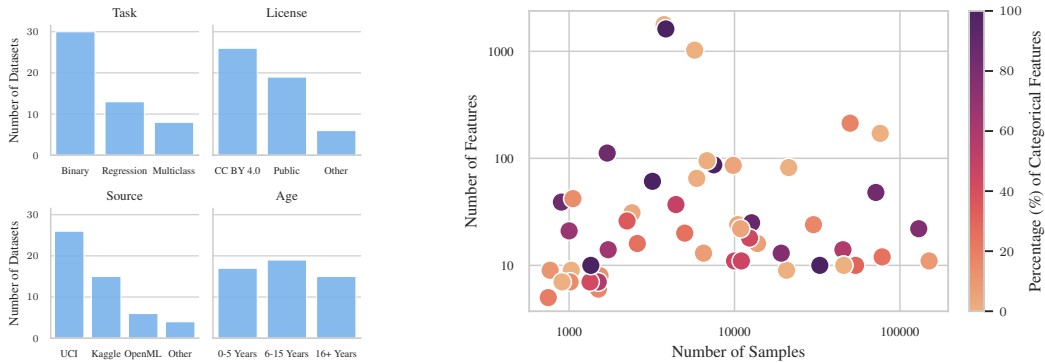

Figure 3: **Characteristics of Datasets in TabArena.** On the left, we show the number of datasets per task type, license, source of the dataset, and age group. On the right, we show the number of features (columns) and samples (rows), as well as the percentage of categorical features per dataset.

Figure 3 summarizes dataset characteristics; we share per-dataset details and domain coverage in Appendix B.2. Throughout our curation process, we noticed two trends. First, across benchmarks, the versions of datasets were often inconsistent. Benchmarks used the same name and source for a dataset, but with different preprocessing, features, or even targets. Thus, dataset-specific performance comparisons across benchmarks were often invalid. Second, the number of datasets that are truly suitable for benchmarking IID tabular data approaches is surprisingly small after carefully inspecting the tasks represented by datasets. We share more noteworthy observations in Appendix B.3.

**Call for Data Contributions.**   `TabArena` is a living benchmark, and the datasets we curated are a significant part of this living, continuously maintained system. As mentioned above, we invite the community to scrutinize our curation efforts. Moreover, to keep `TabArena` up-to-date, we implore the broader scientific community to share their tabular data publicly. We are also actively looking for contributions of data outside of `TabArena`-v0.1's focus, i.e., data that is non-IID, tiny, or large, as well as tasks where extracing features from other modalities is still considered a reasonable approach.

Likewise, we invite the community to curate additional existing datasets. We provide a detailed protocol to contribute a new dataset to `TabArena` in Appendix E.3.

## 2.3 Evaluation Design Protocol

The `TabArena` leaderboard aims to assess the state-of-the-art for predictive machine learning on tabular data. With this in mind, we design the evaluation to produce a reliable and representative leaderboard. To guard against randomness in the data and method significantly impacting our conclusions, we repeat our experiments per dataset. We detail the various sources of randomness in `TabArena` in Appendix D.2. We employ a dataset-specific repetition strategy using more repeats for smaller datasets and fewer for larger datasets, because the significance of randomness decreases with dataset size. The strategy is as follows: (**I**) for datasets with less than 2 500 samples, we use 10 times repeated 3-fold outer cross-validation; (**II**) for all other datasets, we use 3 repeats. For classification tasks, we use class-wise stratified cross-validation.

**Evaluation Metric.** We evaluate models using the Elo rating system [38]. Elo is a pairwise comparison-based rating system where each model's rating predicts its expected win probability against others, with a 400-point Elo gap corresponding to a 10 to 1 (91%) expected win rate. We calibrate 1000 Elo to the performance of our default random forest configuration across all figures, and perform 200 rounds of bootstrapping to obtain 95% confidence intervals, similar to Chiang et al. [39], see Appendix D.1. In our main results, Elo scores are computed using ROC AUC for binary classification, log-loss for multiclass classification, and RMSE for regression.

We aim to provide a ranking of models as part of `TabArena`. Every aggregation metric used for ranking has its own pitfalls, and none is perfect; however, we argue that Elo aligns most closely with our goals and has therefore been selected as our primary evaluation metric. Elo is based on pairwise comparison scoring, which only considers wins, ties, or losses and neglects the magnitude of performance differences. This can be a disadvantage, as minor performance differences may be considered irrelevant by practitioners in some applications. At the same time, this means that each dataset contributes equally to Elo, hence the aggregation is not biased towards certain domains or dataset properties (e.g., small or non-noisy datasets). For `TabArena`, this is a key advantage because we want to create a benchmark whose results are representative of all domains and datasets.

While we use Elo as our primary evaluation metric, we also track and present additional aggregation metrics in Section 3 and on the leaderboard. Users can re-rank the leaderboard according to alternative metrics, such as Improvability. Improvability measures the performance of methods relative to the best method, and is therefore sensitive to the magnitude of performance differences; see Appendix A.1 for details. We also share scripts to generate the leaderboard, evaluation plots, and inspect the results.

**TabArena Reference Pipeline.** Following the recommendations of Tschalzev et al. [11], we include a reference pipeline in our benchmark. This reference pipeline is applied independently of the tuning protocol and constraints we constructed for models within `TabArena`. It aims to represent the performance easily achievable by a practitioner on a dataset.

We select the predictive machine learning system AutoGluon [19] (version 1.3, with the `best_quality` preset and 4 hours for training) as the first official `TabArena` reference pipeline.

**Additional Metadata.** Next to the main results, we save an extensive amount of additional metadata to enable future research and deeper model studies. We save the training time, inference time, precomputed results for various metrics, hyperparameters, hardware specification, and model predictions. We save the validation and test predictions of the final model, and of all models trained per-fold during inner cross-validation.

**Living Benchmark.** `TabArena` marks the start of an open-source initiative towards a continuously updated, collectively shared assessment of the state-of-the-art in tabular machine learning. Therefore, as one of the most crucial parts of `TabArena`, we define a protocol for researchers and practitioners to submit models to the live leaderboard. The protocol is detailed in Appendix E.4

## 3 Results

To initialize the leaderboard, we run 16 curated models (Section 2.1) on 51 curated datasets (Section 2.2) within `TabArena`'s evaluation design (Section 2.3). Likewise, we evaluate AutoGluon, the reference pipeline, on all datasets. We run TabM, ModernNCA, and the foundation models on GPU

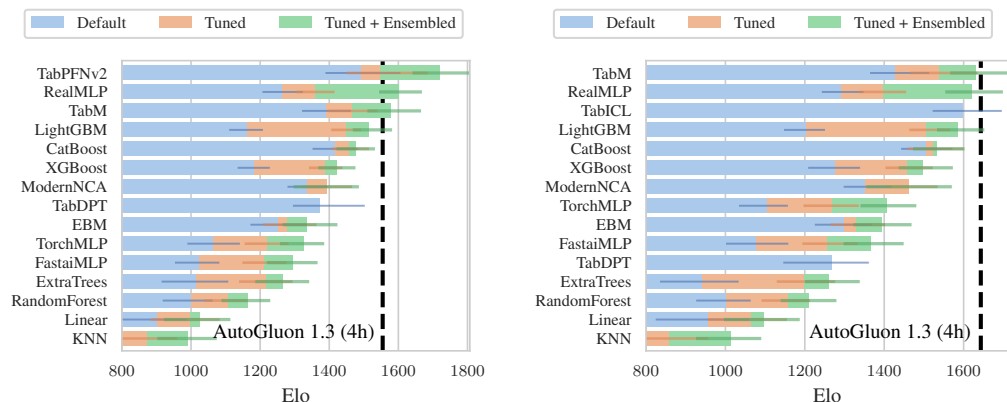

Figure 4: **Leaderboard for TabPFNv2-compatible (left) and TabICL-compatible (right) datasets.** For TabPFNv2, we obtain 33 datasets ($\leq$ 10K training samples, $\leq$ 500 features). For TabICL, we obtain 36 *classification* datasets ($\leq$ 100K, $\leq$ 500). Everything but the datasets is identical to Figure 1.

and all other models, as well as AutoGluon, on CPU. We give all runs 32 GB of RAM and 100 GB of disk space. We perform CPU runs in the cloud via Amazon Web Services on an M6i.2xlarge EC2 instance (eight cores Intel Xeon CPU). We perform GPU runs on an NVIDIA L40S with 48 GB VRAM and eight cores of an AMD EPYC 9334 CPU. Based on our logs, the compute time for all our experiments without parallelization, including overhead such as scheduling, is ∼15 wall-clock years.

## 3.1 Assessing Peak Performance: The TabArena-v0.1 Leaderboard

Figure 1 shows our main results, the leaderboard of `TabArena`-v0.1, which includes the performance of methods with default parameters, a single configuration on validation data after hyperparameter tuning, and weighted post-hoc ensembling [1] of different hyperparameter configurations. Furthermore, we present one leaderboard encompassing all models in Appendix A.1, where we impute missing datasets, representing the live leaderboard available on `tabarena.ai`. To provide a wide range of perspectives, we also share additional versions of the leaderboard in Appendix A.1, such as using alternative metrics or task-wise dataset subsets. Finally, we analyze the statistical significance of our comparison in Appendix A.5.

**Post-Hoc Ensembled Deep Learning Models Dominate the Leaderboard.** In line with previous work [33], CatBoost is ranked first in the conventional tuning regime (Figure 1). However, the `TabArena`-v0.1 leaderboard reveals that after post-hoc ensembling, neural networks are the strongest single models on average in `TabArena`-v0.1. Our results show that the peak performance of models is misrepresented unless post-hoc ensembling is used. This is evident in the observation that the top three models in our leaderboard (TabM, LightGBM, RealMLP; see Figure 1) would all be worse than the actual fourth-best model (CatBoost) without post-hoc ensembling. While practitioners must trade off the increased predictive performance with the increased inference cost (see Figure 6) [40], peak performance requires post-hoc ensembling. Furthermore, this shows that compared to existing benchmarks, `TabArena` is closer to correctly representing the currently achievable peak performance.

**Tabular Foundation Models Lead on Small Datasets.** As most foundation models are currently limited in their applicability, we evaluate them separately in Figure 4, presenting the leaderboard after restricting the datasets to the constraints of TabPFNv2 or TabICL. TabPFNv2 outperforms related approaches by a large margin, establishing tabular foundation models as the go-to solution for datasets within their constraints. Moreover, TabPFNv2 with tuning and post-hoc ensembling again outperforms AutoGluon, confirming the results by Hollmann et al. [5]. We expect tabular foundation models with wider applicability to be released and included in future versions of `TabArena`.

**Pareto Fronts and Tuning Trajectories Reveal Efficiency Tradeoffs.** The Pareto front of improvability and inference time in Figure 5 (left) reveals that tuned EBM (EBM-T) and tuned CatBoost (CatBoost-T) shine at inference time. The next Pareto points with strong improvements come with an increase in inference time of ∼ 15× for TabM-TE and ∼ 100× for RealMLP-TE. Figure 5

(right) shows that gradient-boosted trees have strong performance given their training cost. RealMLP only starts to dominate them after a considerable amount of training time with an ensemble of 25+ configurations. For additional discussion of the tuning trajectories, see Appendix A.6.

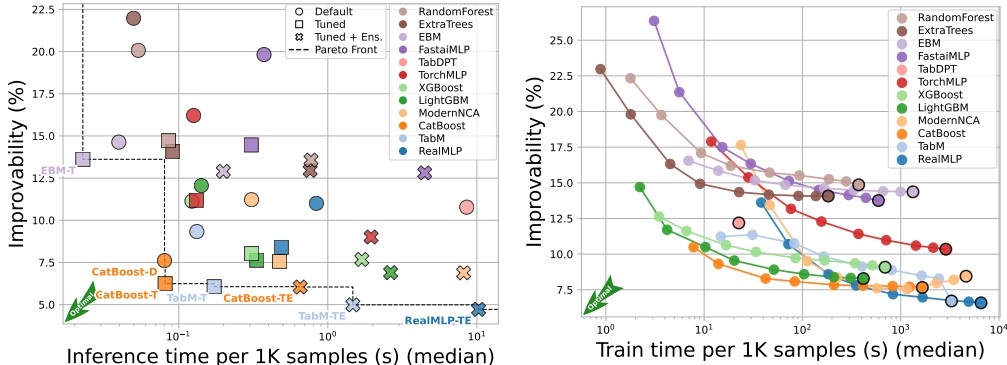

Figure 5: **(Left) Pareto front of improvability and inference time.** We report the median inference time per 1000 samples across all datasets. **(Right) Improvability tuning trajectories.** Time is shown as the tuning time with points from left to right marking ensembles of increasing numbers of random configurations (1, 2, 5, 10, 25, 50, 100, 150, 201). The trajectories are sampled 20 times from all trials and averaged. The right-most highlighted points use all configurations.

## 3.2 Holistic Benchmarking of Peak Performance with TabArena

In this subsection, we demonstrate how the design choices behind `TabArena` enable users to assess peak performance and evaluate the utility of models in ensembling pipelines.

**Abandoning the Holdout Validation Fallacy.** The rich metadata we save for `TabArena` allows us to study what-if cases for our evaluation design. Figure 6 shows that when using holdout validation instead of cross-validation for model selection, all models are greatly underestimated, and performance is biased in favor of models that already use ensembling. This aligns with prior work [11, 41, 42] and demonstrates the importance of nested cross-validation in our benchmark design.

**Reducing Time and Hardware Constraints for Accurate Benchmarking.** While `TabArena` enables a more accurate assessment of peak performance, it does so at a non-negligible computational cost. We hypothesize that prior benchmarks did not assess peak performance mainly due to time and hardware constraints. Thus, we share our result artifact to reduce time and hardware constraints for accurate benchmarking. As a result, users can cheaply compare novel models, and simulated

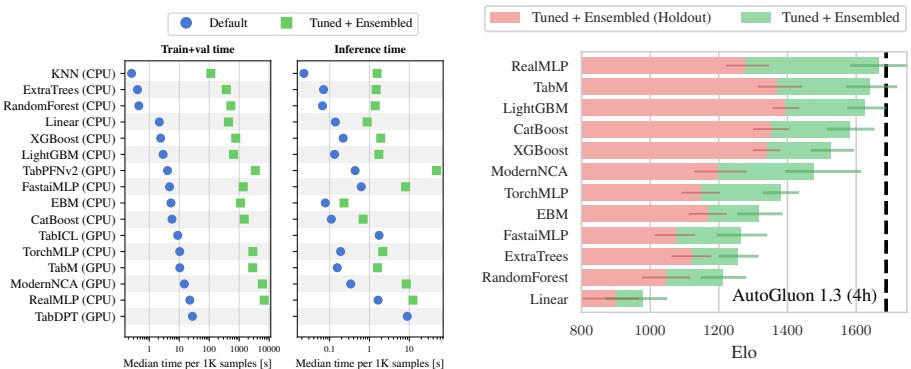

Figure 6: **(Left) Model Efficiency.** Median training times with cross-validation across TabPFNv2- and TabICL-compatible datasets (see Figure 4). **(Right) Holdout vs. Cross-Validation.** Predictive performance of a model with tuning and ensembling when using holdout or cross-validation.

ensembles thereof, to existing models and ensemble pipelines (see Appendix E.1).

Furthermore, we introduce and continually maintain `TabArena-Lite`, a subset of `TabArena` that consists of all datasets with one outer fold (see Appendix A.4). We intend `TabArena-Lite` to be used in academic studies and find any novel model that significantly outperforms other models on at least one dataset, even if it is not the state-of-the-art on average, worthy of publication.

**Ensembles Across Models and Individual Contributions.** While our public leaderboard primarily ranks models, we want to raise awareness that this is a limited perspective that wrongly suggests that lower-ranked models are unnecessary. In Figure 7 (left), we show that a simulated ensembling pipeline using all models in `TabArena` outperforms all individual models and AutoGluon, further advancing the state-of-the-art. In Figure 7 (right), we show the average weights of different models in the `TabArena` ensemble. Notably, models with the highest performance on the leaderboard are not necessarily the ones with the highest weights, likely because the ensemble construction favors models overfitting the validation data [43, 44], such as ModernNCA and RealMLP as seen in Appendix A.6.

**A Sober Look at the "GBDT vs. Deep Learning" Debate.** Prior benchmarks dedicated an enormous amount of work toward debating whether gradient-based decision trees (GBDTs) or deep learning is better across benchmarks and model studies [31, 32, 33]. We argue that the battle between GBDTs and deep learning is a false dichotomy, as both model families contribute to ensembles that strongly outperform individual model families (Figure 7). Thus, we posit that it is more important to find new models that work well in ensembles than models that beat GBDTs. `TabArena` enables such research by allowing users to simulate ensemble performance per and across models based on our precomputed result artifacts. Finally, we hypothesize that a major reason for practitioners to rely on GBDTs instead of deep learning is the insufficient code quality and maintenance of deep learning methods compared to GBDT frameworks [45]. `TabArena` also tackles this problem by wrapping existing deep learning methods in an easy-to-use interface, see Appendix E.5. However, other reasons may also contribute to the slow adoption of deep learning methods, such as longer training time (especially on CPU), the lack of some functionalities, habit, the existence of more educational material for GBDTs, and overclaims of deep learning performance in academic papers.

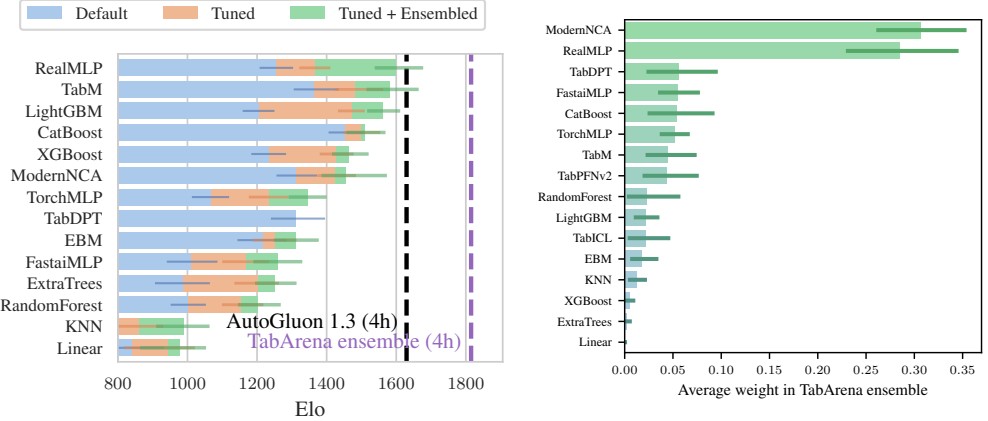

Figure 7: **(Left) TabArena Ensemble.** Simulated performance of an ensemble using all models in `TabArena` compared to the leaderboard from Figure 1. **(Right) Individual Contributions.** Contributions of models to the `TabArena` ensemble in terms of average weight across all datasets.

## 4 Related Work

Many tabular benchmarks have been proposed over the recent years, including OpenML-CC18 [28], PMLB(Mini) [46, 47, 48], the Grinsztajn et al. [32] benchmark, TabZilla [33], the AutoML Benchmark [35], TabRepo [37], Tschalzev et al. [8], TabRed [10], the Zabërgja et al. [49] benchmark, and TALENT [50]. In addition, many studies used their own, alternative evaluation frameworks [5, 9, 20, 26, 30, 51]. This shows that the community does not have a consistent shared understanding of how to benchmark tabular models and, more importantly, no platform to align and collectively improve benchmarking efforts. `TabArena` aims to provide both.

Recent studies revealed further flaws in tabular benchmarking: datasets are often outdated [7], non-IID

data is used inappropriately [10], many datasets are not originally tabular [7, 10], and inappropriate evaluation protocols are frequent [8, 11]. `TabArena` addresses these concerns through extensive data curation efforts and evaluation in sophisticated ensembling pipelines. As seen in Table 2, `TabArena` consolidates the individually outstanding aspects of previous benchmarks.

While some existing tabular benchmarks outline guidelines and commit to adding datasets or models, none include maintenance protocols or have been updated to address issues discovered in data and evaluation quality. To the best of our knowledge, the AutoML benchmark [35] and the TALENT benchmark [36] are the most similar to the vision of `TabArena`; both benchmarks received updates and are actively maintained. `TabArena` is the first living tabular benchmark for machine learning models with active maintenance protocols and a public leaderboard for tabular data.

Our work is inspired by benchmarking efforts from other domains, such as ChatBot Arena [39], RewardBench [52], LiveBench [53], the Huggingface Open LLM Leaderboard [54, 55], and GIFT-Eval [56]. The most significant similarity between these efforts and `TabArena` is that we also provide a live leaderboard. Otherwise, `TabArena` is clearly distinct through our focus on predictive machine learning for tabular data and the resulting differences in models, datasets, and evaluation design.

## 5   Conclusion

We introduced `TabArena`, the first living benchmark for machine learning on small to medium-sized tabular data. We described the core components of `TabArena`: models and hyperparameter optimization, datasets, and evaluation design protocols. As part of `TabArena`, we share rich metadata and reproducible code with the community, allowing the community to benchmark and evaluate new models in a standardized way. We then instantiated `TabArena` by curating 16 models and 51 representative real-world tabular data tasks out of 1053 available datasets for running a large-scale benchmark consisting of ~25 000 000 individual runs that took ~15 years of wall-clock time. To the best of our knowledge, our results are the most representative assessment of the state-of-the-art for tabular data to date. We found that deep learning models such as TabM and RealMLP, as well as foundation models for small data, perform similar to or better than gradient-boosted decision trees. Moreover, post-hoc ensembling per model and across models can dramatically improve performance.

**Limitations and Societal Impact.**   As we envision a living benchmarking system that will evolve over time, some limitations can be seen as future work, while others stem from fundamental trade-offs in benchmark design choices. Our (current) limitations are: (1) We use a fixed set of 200 random hyperparameter configurations to enable the study of ensemble pipelines. This prevents analyzing the variance of random hyperparameter choices [57] and studying more advanced hyperparameter optimization strategies. (2) We use a time limit per configuration; thus, our results depend on the hardware used in edge cases where the time limit is reached. Using different hardware across models (and in the future across users) reduces the comparability in such cases. (3) Our strict selection criteria for datasets makes `TabArena` more representative for real-world use-cases, but reduces the number of datasets drastically. We emphasize the need to work with the community to curate a more representative, high-quality collection of datasets with more diversity and statistical power. (4) Lastly, we assess predictive performance without feature engineering on top of the existing dataset state. Feature engineering could further boost predictive performance and change the model ranking.

A fundamental limitation of open-source benchmarking is that foul play or dataset contamination could compromise the leaderboard. Model providers could overfit the hyperparameters of their model on the `TabArena` datasets, or use them for pretraining a foundation model. We posit that keeping the `TabArena` alive is the best way to handle these challenges; see Appendix E.6 for a discussion.

Our work has a broader positive societal impact by improving the trustworthiness and reliability of academic benchmarks. Moreover, `TabArena` provides practical guidance to practitioners for small-to medium-sized IID tabular data. Lastly, while the upfront computational costs of `TabArena` imply a negative environmental impact, we argue that our secondary contributions, such as sharing result artifacts, will offset the negative environmental impact; we elaborate on this in Appendix D.3.

**Future Work.**   Our vision for `TabArena`-v1.0 is a sophisticated benchmark for classification and regression for *any* tabular dataset for the entire community. Our specific next steps towards this goal are: supporting tiny, large, and non-IID data, integrating new models, and curating more datasets.

**To conclude,** `TabArena` is a significant step towards making benchmarking tabular machine learning models a straightforward and reliable process. We look forward to seeing our living benchmark grow and evolve in cooperation with the tabular machine learning community.

## Acknowledgments and Disclosure of Funding

We thank the authors of all models with whom we were in contact for feedback and code contributions regarding the implementations of their models, namely, not among the authors of this manuscript: Yury Gorishniy; Paul Koch; Jingang Qu; and Han-Jia Ye. Furthermore, we thank the dataset donors for sharing their data, especially those under a public license. Moreover, we thank the community, particularly OpenML, Kaggle, and Hugging Face, for developing tools that facilitate the sharing of tabular data. Finally, we thank the reviewers for their constructive feedback and contribution to improving the paper.

L.P. acknowledges funding by the Deutsche Forschungsgemeinschaft (DFG, German Research Foundation) under SFB 1597 (SmallData), grant number 499552394. A.T. acknowledges funding by the Ministry of Economic Affairs, Labour and Tourism Baden-Württemberg. F.H. acknowledges the financial support of the Hector Foundation. This research was funded by the Deutsche Forschungsgemeinschaft (DFG, German Research Foundation) under grant number 539134284, through EFRE (FEIH_2698644) and the state of Baden-Württemberg.

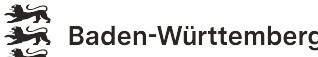 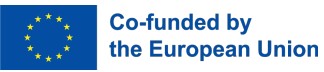

## Author Contributions

N.E. led the code development, co-led the model curation, led the CPU experiments, contributed to writing the paper, contributed to the management of the collaboration, led the evaluation, and co-led visualizations. L.P. contributed to the code development, contributed to the model curation, led the GPU experiments, led the writing of the paper, led the management of the collaboration, co-led the dataset curation, contributed to the evaluation, and contributed to the visualizations. A.T. helped with the code development, co-led the dataset curation, contributed to writing the paper, contributed to the evaluation, and contributed to the visualizations. D.H. contributed to the code development, co-led the model curation, contributed to writing the paper, contributed to the evaluation, and co-led visualizations. P.M.D. contributed to the code development. D.S. helped with visualizations, helped with supervising the project, and contributed to reviewing and editing the paper. F.H. supervised L.P. and helped with reviewing and editing the paper.

## Competing Interests

D.H. is one of the authors of RealMLP and one of the authors of TabICL. D.S. and N.E. are the authors of TabRepo. N.E., L.P., and P.M.D. are developers of AutoGluon, and in extension, the current maintainers of FastAI MLP and Torch MLP. L.P. and F.H. are a subset of the authors of TabPFNv2. L.P. is an OpenML core contributor. F.H. is affiliated with PriorLabs, a company focused on developing tabular foundation models. The authors declare no other competing interests.

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

# Appendices

**Table of Contents.**

# A  Additional Experiments

## A.1  Alternative Leaderboard Versions

**Aggregation Methods.**   Here, we provide more leaderboard variants, using different aggregation strategies. Specifically, we obtain errors $\text{err}_i$ for each dataset $i$ by averaging error metrics (1-AUROC for binary, logloss for multiclass, and RMSE for regression) over all outer folds. We then aggregate these errors as follows:

- **Elo**: As described in Section 2.3.
- **Normalized score**: Following Salinas and Erickson [37], we linearly rescale the error such that the best method has a normalized score of one, and the median method has a normalized score of 0. Scores below zero are clipped to zero. These scores are then averaged across datasets.
- **Average rank**: Ranks of methods are computed on each dataset (lower is better) and averaged.
- **Harmonic mean rank**: Taking the harmonic mean of ranks,

$$\frac{1}{\frac{1}{N}\sum_{i=1}^{N}(1/\text{rank}_i)},$$

  more strongly favors methods having very low ranks on some datasets. It therefore favors methods that are sometimes very good and sometimes very bad over methods that are always mediocre, as the former are more likely to be useful in conjunction with other methods.
- **Improvability**: We introduce improvability as a metric that measures how many percent lower the error of the best method is than the current method on a dataset. This is then averaged over datasets. Formally, for a single dataset,

$$\text{Improvability} := \frac{\text{err}_i - \text{best\_err}_i}{\text{err}_i} \cdot 100\% \,.$$

  Improvability is always between $0\%$ and $100\%$.

**Results.**   Figure A.1 presents a leaderboard including all models. We impute the results for models on datasets where they are not applicable with the results of RandomForest (default). We choose the default random forest since it is a fast baseline that is sufficiently but not unreasonably weak, to penalize models that are not applicable to all datasets. Table A.1 presents the same data in tabularized format, akin to the current version of the live leaderboard at tabarena.ai. Table A.1 further includes several additional metrics to asses peak average performance, some of which change the ranking (see the color coding) as they are less influenced by model-wise negative outlier results introduced by imputation.
We further investigate our results by presenting the leaderboard across task types. We show the results per task type by computing the results only with datasets from: binary classification in Figure A.2, multiclass classification in Figure A.3, and regression in Figure A.4.
In addition, we show pairwise win rate comparisons of the models in Figure A.5. Among the models with the best performance in our main analysis, the win rates over strong competitors are rather moderate. This undermines that there is no one-size-fits-all solution.
Next, Figure A.6 and Figure A.7 present the results for the TabPFNv2-compatible and TabICL-compatible datasets, but also impute TabPFNv2/TabICL to enable a more direct comparison between these two foundation models. Finally, Figure A.8 presents the result only with datasets for which both TabPFNv2 and TabICL are compatible.

## A.2  Analyzing Training Time Limit

In our experiments, we restrict the time to evaluate one configuration on one train split of a dataset to 1 hour. Thus, a model must finish training (across all $8$ inner folds) within 1 hour, or its training will be gracefully stopped early. Figure A.9 presents the training runtime for all hyperparameter configurations for all models by visualizing what proportion of configurations (x-axis) took how many seconds for training (y-axis).

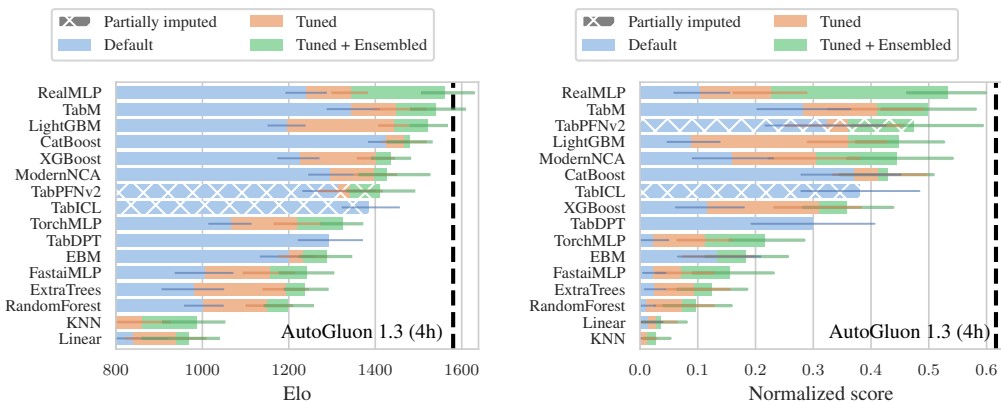

Figure A.1: **TabArena-v0.1 leaderboard with imputation for TabPFNv2 and TabICL, Elo (left) and normalized scores (right).** For TabPFNv2 and TabICL, on datasets where they are not applicable, we impute their results with RandomForest (default).

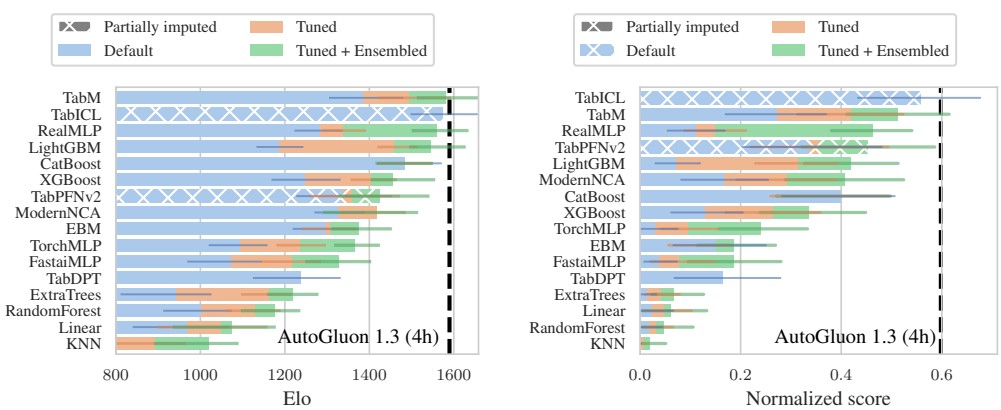

Figure A.2: **Benchmark results on binary classification with Elo (left) and normalized scores (right).** For TabPFNv2 and TabICL, on datasets where they are not applicable, we impute their results with RandomForest (default).

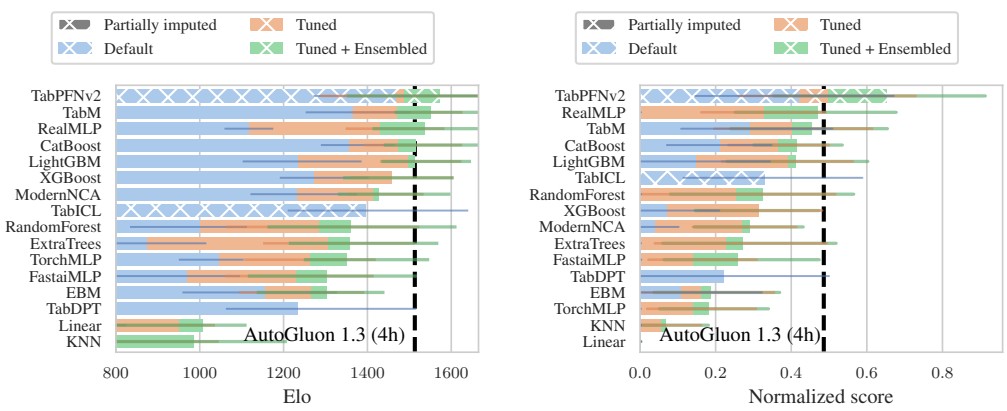

Figure A.3: **Benchmark results on multiclass classification with Elo (left) and normalized scores (right).** For TabPFNv2 and TabICL, on datasets where they are not applicable, we impute their results with RandomForest (default).

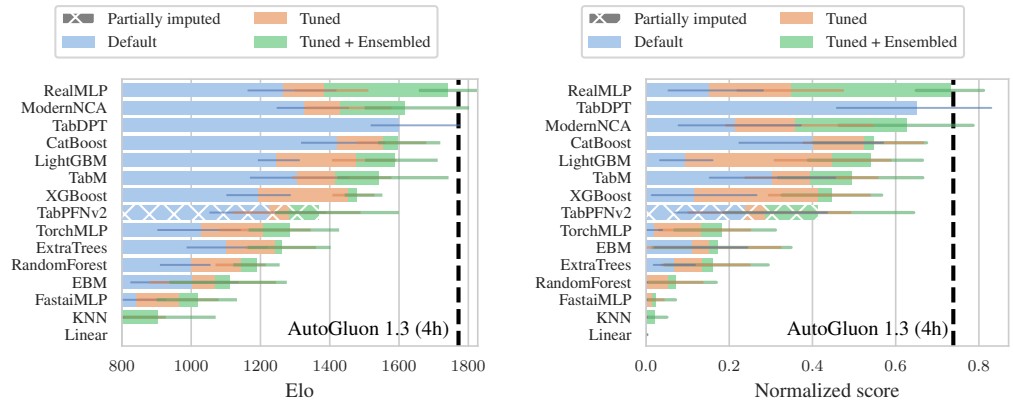

Figure A.4: **Benchmark results on regression with Elo (left) and normalized scores (right).** For TabPFNv2 and TabICL, on datasets where they are not applicable, we impute their results with RandomForest (default).

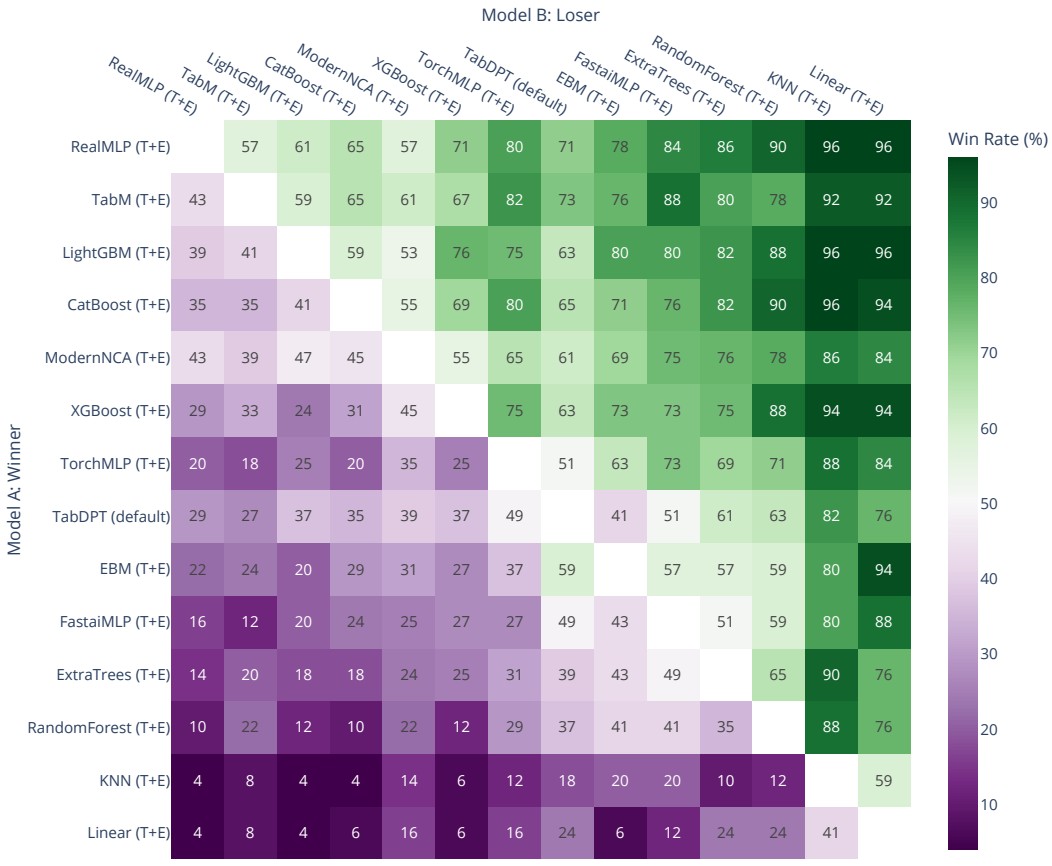

Figure A.5: **Pairwise win rate comparison for all datasets.** Higher numbers correspond to a better win rate for the model on the y-axis.

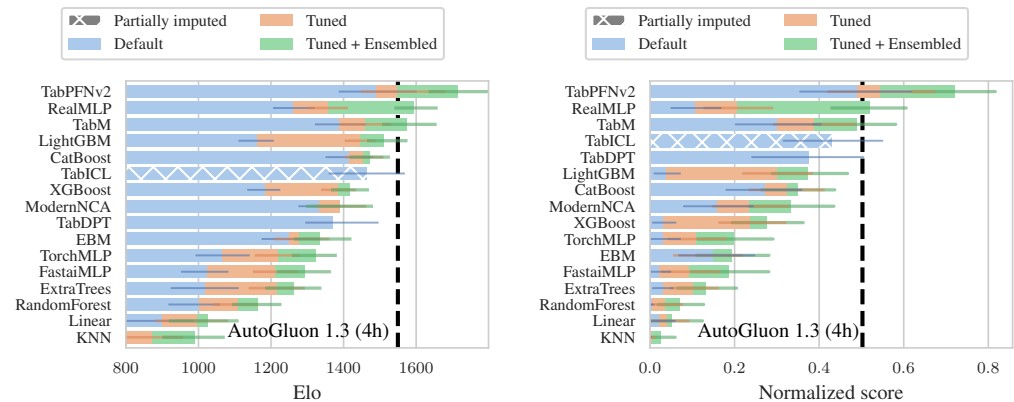

Figure A.6: **Benchmark results on TabPFNv2-compatible datasets with imputed results for TabICL, using Elo (left) and normalized scores (right).** On datasets where TabICL is not applicable, we impute its results with RandomForest (default).

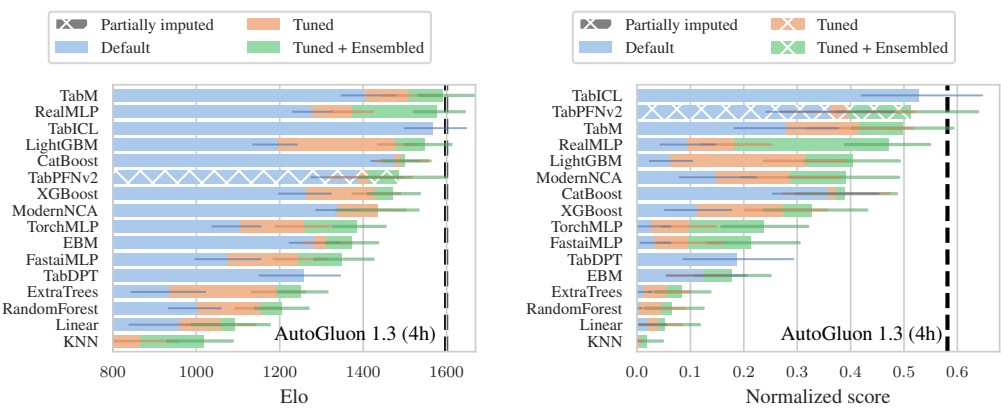

Figure A.7: **Benchmark results on TabICL-compatible datasets with imputed results for TabPFNv2, using Elo (left) and normalized scores (right).** On datasets where TabPFNv2 is not applicable, we impute its results with RandomForest (default).

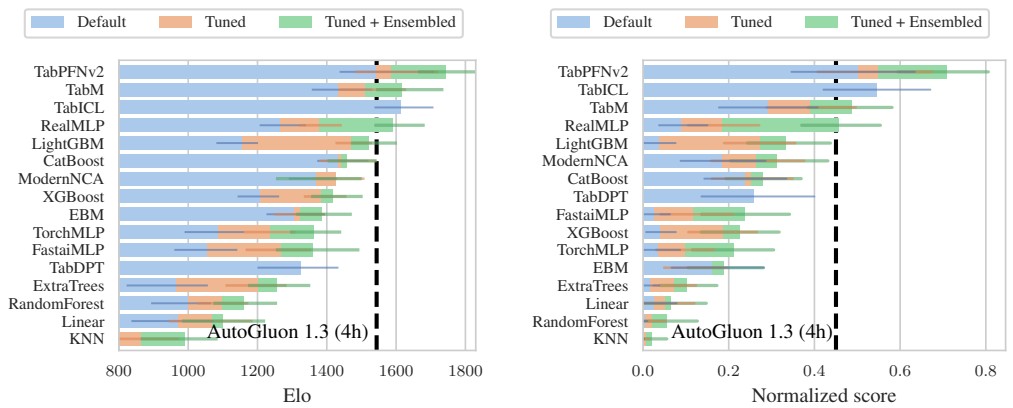

Figure A.8: **Benchmark results on TabPFNv2- and TabICL-compatible datasets using Elo (left) and normalized scores (right).**

Table A.1: **TabArena-v0.1 Leaderboard.** We show default (D), tuned (T), and tuned + ensembled (T+E) performances. Results of TabPFNv2 and TabICL are imputed with RandomForest (default) for datasets on which they were not run. Times are median times per 1K samples across datasets, averaged over all outer folds per dataset. The best three values in columns are highlighted with gold, silver, and bronze colors. For Elo values, we also indicate their approximate 95% confidence intervals obtained through bootstrapping.

| Model | Elo (↑) | Norm. score (↑) | Avg. rank (↓) | Harm. mean rank (↓) | #wins (↑) | Improvability (↓) | Train time per 1K [s] | Predict time per 1K [s] |
|---|---|---|---|---|---|---|---|---|
| RealMLP (T+E) | $1564_{-57,+73}$ | 0.532 | 8.6 | 4.9 | 1.0 | 6.4% | 6564.71 | 10.26 |
| TabM (T+E) | $1541_{-61,+74}$ | 0.499 | 9.4 | 4.4 | 4.0 | 6.7% | 3285.87 | 1.47 |
| LightGBM (T+E) | $1527_{-45,+49}$ | 0.448 | 9.9 | 5.5 | 2.0 | 8.3% | 416.98 | 2.64 |
| CatBoost (T+E) | $1482_{-55,+57}$ | 0.428 | 11.7 | 7.3 | 0.0 | 7.6% | 1658.41 | 0.65 |
| CatBoost (T) | $1469_{-49,+56}$ | 0.411 | 12.2 | 6.1 | 2.0 | 7.8% | 1658.41 | 0.08 |
| TabM (T) | $1447_{-52,+73}$ | 0.410 | 13.2 | 6.7 | 1.0 | 7.7% | 3285.87 | 0.17 |
| LightGBM (T) | $1446_{-37,+39}$ | 0.359 | 13.2 | 10.7 | 0.0 | 9.0% | 416.98 | 0.33 |
| XGBoost (T+E) | $1440_{-47,+50}$ | 0.358 | 13.5 | 8.2 | 1.0 | 9.1% | 693.49 | 1.69 |
| CatBoost (D) | $1427_{-43,+52}$ | 0.369 | 14.1 | 7.0 | 2.0 | 9.0% | 6.83 | 0.08 |
| ModernNCA (T+E) | $1425_{-67,+103}$ | 0.444 | 14.2 | 5.0 | 3.0 | 8.4% | 4621.67 | 8.15 |
| TabPFNv2 (T+E) | $1405_{-76,+83}$ | 0.474 | 15.1 | 3.1 | 11.0 | 8.3% | 3030.15 | 21.44 |
| XGBoost (T) | $1404_{-44,+49}$ | 0.309 | 15.1 | 12.2 | 0.0 | 9.4% | 693.49 | 0.31 |
| ModernNCA (T) | $1396_{-44,+55}$ | 0.305 | 15.5 | 8.6 | 1.0 | 9.1% | 4621.67 | 0.47 |
| TabICL (D) | $1383_{-64,+71}$ | 0.380 | 16.1 | 4.5 | 6.0 | 8.9% | 6.63 | 1.48 |
| TabM (D) | $1343_{-55,+68}$ | 0.282 | 18.0 | 11.9 | 0.0 | 10.7% | 10.49 | 0.13 |
| RealMLP (T) | $1342_{-45,+43}$ | 0.226 | 18.1 | 15.3 | 0.0 | 9.8% | 6564.71 | 0.49 |
| TabPFNv2 (T) | $1336_{-72,+77}$ | 0.359 | 18.4 | 5.7 | 1.0 | 10.4% | 3030.15 | 0.46 |
| TorchMLP (T+E) | $1327_{-53,+49}$ | 0.215 | 18.8 | 13.6 | 0.0 | 10.3% | 2874.67 | 1.95 |
| TabPFNv2 (D) | $1309_{-81,+87}$ | 0.323 | 19.7 | 5.5 | 4.0 | 11.4% | 3.36 | 0.31 |
| ModernNCA (D) | $1294_{-48,+58}$ | 0.158 | 20.5 | 12.3 | 1.0 | 12.6% | 14.87 | 0.31 |
| TabDPT (D) | $1290_{-72,+79}$ | 0.298 | 20.7 | 4.8 | 7.0 | 12.0% | 22.53 | 8.55 |
| EBM (T+E) | $1286_{-63,+59}$ | 0.182 | 20.8 | 11.6 | 1.0 | 14.4% | 1331.68 | 0.20 |
| FastaiMLP (T+E) | $1242_{-64,+64}$ | 0.156 | 23.0 | 13.4 | 0.0 | 13.7% | 593.24 | 4.47 |
| RealMLP (D) | $1238_{-47,+51}$ | 0.103 | 23.2 | 19.6 | 0.0 | 12.4% | 21.86 | 0.84 |
| ExtraTrees (T+E) | $1236_{-50,+54}$ | 0.123 | 23.4 | 15.1 | 0.0 | 14.1% | 183.02 | 0.76 |
| EBM (T) | $1231_{-55,+55}$ | 0.132 | 23.6 | 16.3 | 0.0 | 15.0% | 1331.68 | 0.02 |
| XGBoost (D) | $1225_{-50,+49}$ | 0.115 | 23.9 | 18.3 | 0.0 | 12.4% | 1.94 | 0.12 |
| TorchMLP (T) | $1220_{-54,+59}$ | 0.111 | 24.1 | 20.3 | 0.0 | 12.4% | 2874.67 | 0.13 |
| EBM (D) | $1198_{-66,+64}$ | 0.133 | 25.2 | 13.2 | 1.0 | 16.0% | 4.67 | 0.04 |
| RandomForest (T+E) | $1197_{-56,+62}$ | 0.096 | 25.2 | 13.0 | 1.0 | 14.9% | 373.18 | 0.77 |
| LightGBM (D) | $1197_{-45,+44}$ | 0.088 | 25.3 | 22.3 | 0.0 | 13.3% | 1.96 | 0.14 |
| ExtraTrees (T) | $1190_{-52,+55}$ | 0.093 | 25.6 | 17.7 | 0.0 | 15.1% | 183.02 | 0.09 |
| FastaiMLP (T) | $1154_{-61,+60}$ | 0.070 | 27.3 | 21.8 | 0.0 | 15.4% | 593.24 | 0.31 |
| RandomForest (T) | $1149_{-49,+59}$ | 0.071 | 27.5 | 15.1 | 1.0 | 15.9% | 373.18 | 0.09 |
| TorchMLP (D) | $1065_{-49,+50}$ | 0.022 | 31.2 | 28.5 | 0.0 | 17.2% | 9.99 | 0.13 |
| FastaiMLP (D) | $1007_{-71,+66}$ | 0.022 | 33.4 | 30.4 | 0.0 | 20.6% | 2.86 | 0.37 |
| RandomForest (D) | $1000_{-44,+50}$ | 0.009 | 33.7 | 31.8 | 0.0 | 21.0% | 0.43 | 0.05 |
| KNN (T+E) | $984_{-79,+66}$ | 0.027 | 34.2 | 30.7 | 0.0 | 23.3% | 129.01 | 1.80 |
| ExtraTrees (D) | $980_{-75,+72}$ | 0.024 | 34.3 | 30.6 | 0.0 | 22.8% | 0.25 | 0.05 |
| Linear (T+E) | $967_{-110,+72}$ | 0.036 | 34.8 | 26.8 | 0.0 | 28.9% | 237.58 | 0.42 |
| Linear (T) | $936_{-115,+72}$ | 0.027 | 35.8 | 30.1 | 0.0 | 29.7% | 237.58 | 0.08 |
| KNN (T) | $858_{-93,+64}$ | 0.011 | 38.0 | 36.3 | 0.0 | 28.8% | 129.01 | 0.18 |
| Linear (D) | $836_{-136,+84}$ | 0.014 | 38.5 | 30.8 | 0.0 | 32.7% | 1.19 | 0.12 |
| KNN (D) | $614_{-133,+102}$ | 0.000 | 42.2 | 41.7 | 0.0 | 43.0% | 0.19 | 0.04 |

We observe that for all models, less than $1\%$ of all configurations reach the time limit of 1 hour. We further investigate the time limit for the GPU-optimized models in Appendix A.3. For EBMs, we notice that the training was not stopped early at the 1 hour time limit, positively influencing its results. As this only concerns a small fraction of hyperparameter trials, we did not rerun the training for EBM.

## A.3  Tabular Deep Learning on GPU vs. CPU

In our main experiments, we ran TabM and ModernNCA on GPU. To further investigate the impact of hardware choice on these models, we also ran TabM and ModernNCA on CPU. Moreover, we

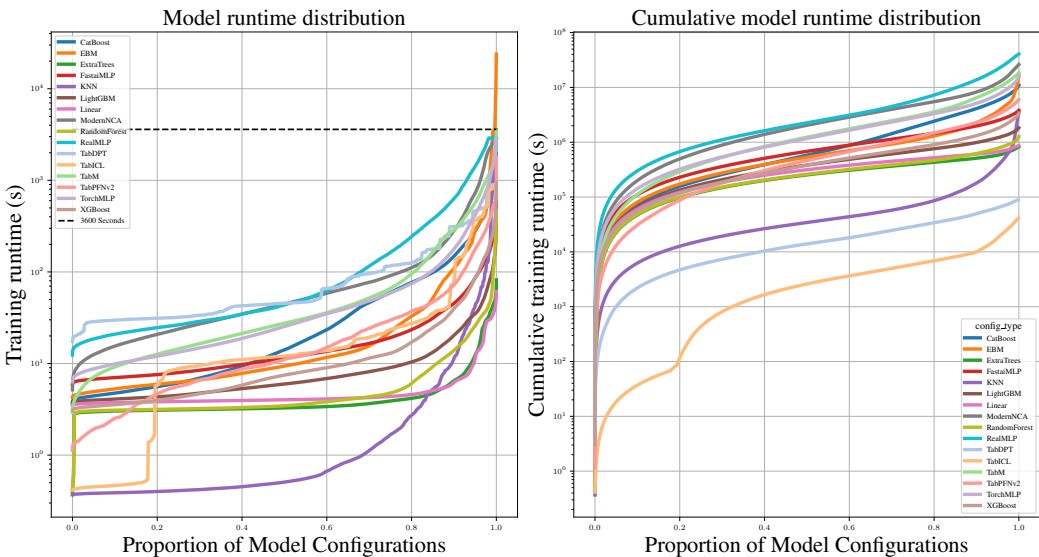

Figure A.9: **Training Runtime Analysis, Runtime Distribution (left) and Cumulative Total Runtime (right) Across Hyperparameter Configurations.** We show the training runtime in seconds for the hyperparameter configurations across models.

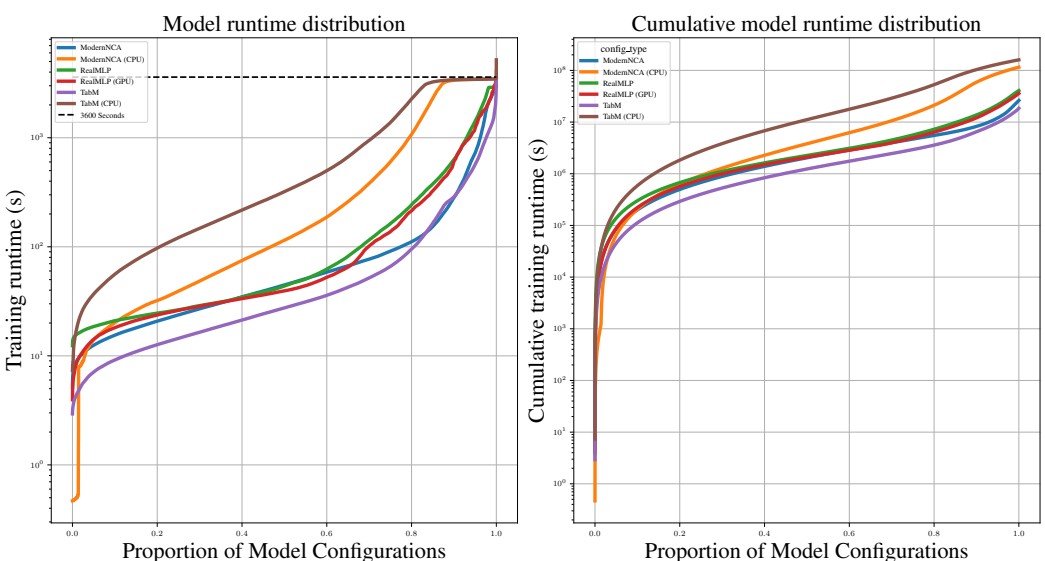

Figure A.10: **Runtime Analysis of RealMLP, TabM, and ModernNCA on CPU vs. GPU.** We show the training runtime distribution of RealMLP, TabM, and ModernNCA trained on CPU and GPU across all TabArena datasets. For ModernNCA and TabM on CPU, approximately $16\%$ of runs are early stopped due to the 1-hour time limit. For GPU, less than $0.1\%$ of runs are early stopped due to the time limit.

ran RealMLP on GPU. Figure A.10 demonstrates that the hyperparameter configurations of TabM and ModernNCA train much faster and RealMLP slightly faster on GPU than on CPU. We conclude from this ablation that training TabM and ModernNCA on CPU with a time limit of 1 hour would negatively influence their predictive performance. While the influence is marginal for RealMLP, it seems non-marginal for TabM and ModernNCA.

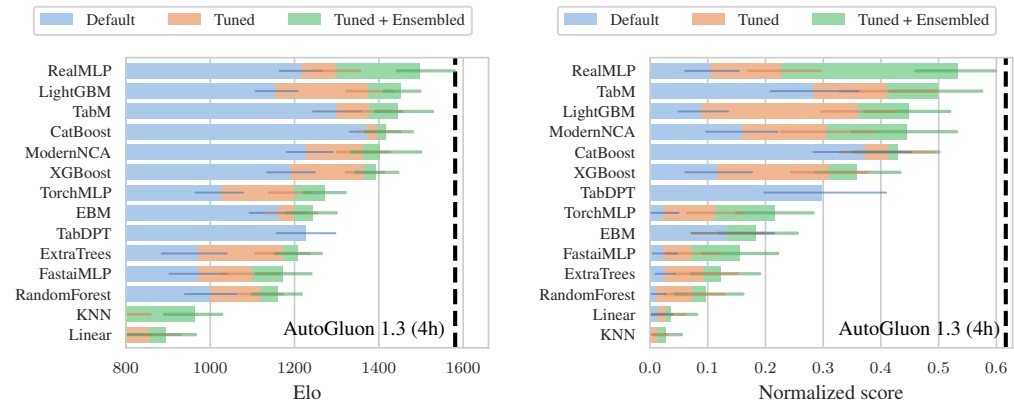

Figure A.11: **Benchmark results on** `TabArena-Lite` **using Elo (left) and normalized scores (right).** Our main leaderboard with `TabArena-Lite`, a subset of `TabArena` consisting of all datasets, but only with one outer fold.

## A.4 TabArena-Lite

Benchmarking can quickly become very expensive, especially with a sophisticated protocol to guarantee robust results. To reduce the cost of benchmarking, we introduce `TabArena-Lite`. `TabArena-Lite` is a continually maintained subset of `TabArena` that consists, in its first version, of all datasets with one outer fold. Figure A.11 shows results on `TabArena-Lite`, using 200 hyperparameter configurations per model, but only a single outer fold for all datasets. The results are similar to the results on `TabArena` in Figure 1, showing that `TabArena-Lite` is a good indicator of model performance.

To further reduce the cost of benchmarking, we also recommend running new models on `TabArena-Lite` with one default hyperparameter configuration and optionally with a lower number of random hyperparameter configurations (e.g., 25). As all other models in `TabArena` are tuned, a less heavily tuned model that performs comparably could already show that a new model is promising. We designate `TabArena-Lite` to be used in academic studies and find any novel model that outperforms other models on at least one dataset, even if it is not among the best on average, a valuable publication. Furthermore, we as maintainers use the performance on `TabArena-Lite` to prioritize the integration of new models into `TabArena`. We envision `TabArena-Lite` also as a living, continuously updated subset. Ideally, future work could determine a method that finds the optimal and most representative subset of partitions and datasets in `TabArena` to populate `TabArena-Lite`.

## A.5 Investigating Statistical Significance

We investigate the statistical significance between models by using critical difference diagrams (CDDs) [58] to represent the results of a Friedman test and then a Nemenyi post-hoc test ($\alpha = 0.05$) from AutoRank[1] [59]. Figures A.12 to A.14 show the CDDs for the full benchmark, TabPFNv2-compatible datasets, and TabICL-compatible datasets with respect to the peak performance of the models, i.e., tuned + ensembled where available. We further investigate statistical significance per-dataset in Appendix F.

We observe that there always exists a group of not statistically significantly different top models containing at least one deep learning model and GBDT, and when available, TabPFNv2 and TabICL.

---

[1]`https://github.com/sherbold/autorank`

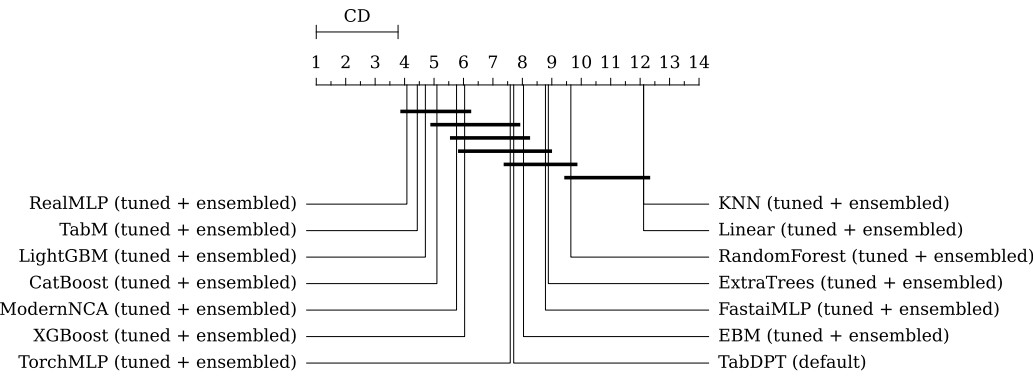

Figure A.12: **Critical Difference Diagram for tuned+ensembled methods on the full benchmark.** Lower ranks are better; horizontal bars connect methods that are not statistically significantly different.

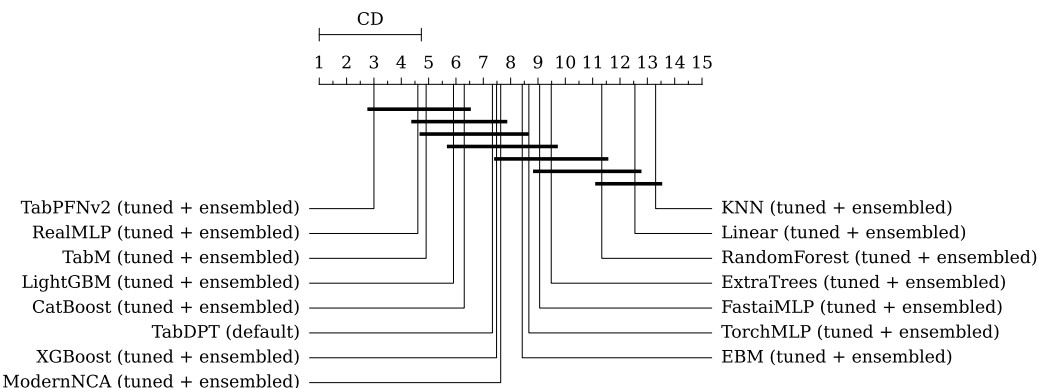

Figure A.13: **Critical Difference Diagram for tuned+ensembled methods on TabPFNv2-compatible datasets.** Lower ranks are better; horizontal bars connect methods that are not statistically significantly different.

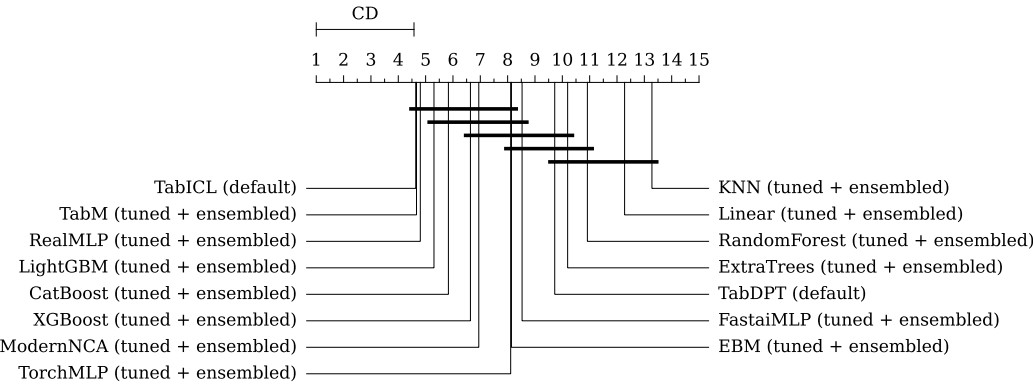

Figure A.14: **Critical Difference Diagram for tuned+ensembled methods on TabICL-compatible datasets.** Lower ranks are better; horizontal bars connect methods that are not statistically significantly different.

## A.6   Investigating Tuning Trajectories and Validation Overfitting

In Figure 5 (right), most models appear to have saturated their performance by 201 tuning configs, indicating that further increasing the amount of configurations beyond 201 is unlikely to lead to significant improvements without also expanding the model search spaces. ModernNCA's performance peaks at 25 configurations and then progressively degrades with further tuning, which can be attributed to validation data overfitting during tuning; also often called overtuning [41, 42]. This can be observed in Figure A.15, where despite ModernNCA being the 6th place model in terms of Elo when evaluated on the test data, it is the 1st place model when evaluated on the validation data. In other words, we observe that overtuning occurs since overfitting on the validation data results in reduced test performance, despite continued improvement in the validation score. Furthermore, the degree of overfitting for each method varies widely. As seen in Figure A.16, neural network models overfit much more than other methods, even with far fewer configurations. ModernNCA, in particular, only needs 10 random configurations ensembled together to overfit more than any non-deep learning method with 201 ensembled configurations. TabM stands out as the sole exception, demonstrating remarkable resilience to overfitting, possibly due to its internal ensembling, joint early stopping for the whole ensemble, and moderately-sized hyperparameter search space.

With these insights, we can now explain why the TabArena ensemble overwhelmingly favors ModernNCA and RealMLP over other models (Figure 7). The reason is not necessarily because these models are the correct ones to choose, but rather because the ensemble is optimizing the validation performance as a proxy for the test performance, and the models that are most overfit on the validation data will naturally be over-selected by the ensembling algorithm (and likewise for model selection). Similarly, models that perform very well on test data while avoiding overfitting (such as TabM) are heavily under-selected by the ensemble. Despite these issues, the TabArena ensemble still drastically outperforms all methods, including AutoGluon. This indicates that a novel overfitting-aware method for selecting configurations in the TabArena ensemble could yield significantly improved performance, further advancing the state of the art. We leave the exploration of such algorithms and other solutions for overtuning to future work.

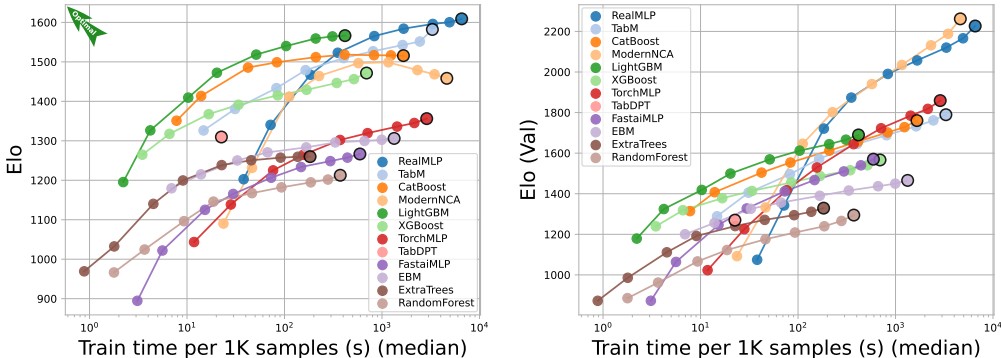

Figure A.15: The tuning trajectories of model ensembles for **(Left) Elo when evaluated on the test data. (Right) Elo when evaluated on the validation data.** Time is shown as the tuning time with points from left to right marking ensembles of increasing numbers of random configurations (1, 2, 5, 10, 25, 50, 100, 150, 201). The trajectories are sampled 20 times from all trials and averaged. The right-most highlighted points use all configurations. The left figure calibrates 1000 Elo to the test performance of the default random forest configuration. The right figure calibrates 1000 Elo to the validation performance of the default random forest configuration.

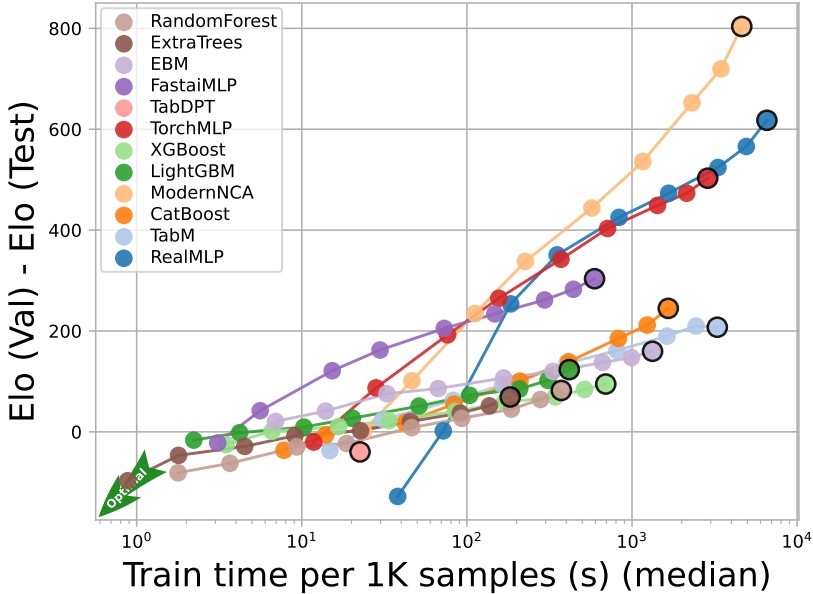

Figure A.16: The validation overfitting tuning trajectories of model ensembles for **Elo when evaluated on the test data** vs. **Elo when evaluated on the validation data**. The setup is identical to Figure A.15. A value of 800 on the y-axis means that the configuration has 800 higher elo based on its validation scores than on its test scores relative to default random forest. Higher values lead to less reliable estimates of a model's performance on test data given only its validation scores, and make effective model selection and post-hoc ensembling more challenging.

## B    Data Curation

For initializing the `TabArena` benchmark, we surveyed all the datasets used in 14 previous benchmarking studies: 450 from PMLB(Mini) [46, 47, 48], 72 from OpenML-CC18 [29], 45 from Grinsztajn et al. [32], 11 from Gorishniy et al. [30], 11 from Shwartz-Ziv and Armon [31], 176 from TabZilla [33], 35 from OpenML-CTR23 [34], 104 from AMLB [35], 187 from Kohli et al. [7], 8 from TabRed [10], 10 from Tschalzev et al. [8], 279 from TabRepo [37], 118 from PyTabKit [20], and 300 from TALENT [36, 50, 60].

These studies were selected with the goal of covering a wide range of datasets used in tabular benchmarking so that we can clean up the field from problematic or unsuitable datasets. Therefore, each of the studies represents a frequently used benchmark or a general milestone study in the field of tabular machine learning. Note that PMLB(Mini) [46, 47, 48] is not included in table 2 as the reference publication did not include an evaluation of methods. Combining the dataset collections results in 1053 uniquely named datasets.

For `TabArena`-v0.1, we aimed at using only those datasets representing realistic, predictive tabular data tasks that practitioners would be interested in solving. Therefore, we define a set of selection criteria described in Appendix B.1. Two of the coauthors manually investigated each of the datasets and applied our selection criteria. We publicly share their notes and curated metadata: `tabarena.ai/dataset-curation`. Furthermore, we share insights from our curation process in Appendix B.3 Importantly, we did not exhaustively test each dataset for each of our curation criteria, but proceeded with the next dataset whenever a dataset clearly met at least one of our criteria for exclusion. Therefore, Figure 2 represents the first reason for exclusion that we noticed in a dataset.

Surprisingly, only 51 datasets satisfying all criteria remained. Appendix B.2 provides additional information on the selected datasets. Moreover, we share all tasks and datasets as an OpenML suite [29, 61, 62] (ID 457, alias "tabarena-v0.1"). We consider our data curation a clean-up for tabular data benchmarking that is necessary, but imperfect. Therefore, we aim to continuously improve the data selection and invite researchers to challenge our documented decisions. Appendix E.3 details protocols to contribute new datasets to `TabArena` by applying our criteria.

**B.1 Dataset Selection Criteria**

The datasets for Tabarena-v0.1 were curated by applying the following criteria:

**Unique datasets:** We want the `TabArena` benchmark to be representative of a wide range of tasks without overrepresenting particular tasks. Therefore, we conduct a four-stage deduplication procedure: (1) Automatically filter data sets by name if they match after transforming to the lower case and removing filling characters ' '|'_'|'-'. (2) Manually remove datasets where different names were used for the same data set in different studies. (3) Manually remove alternative versions of the same dataset, i.e., temporal data sampled at different rates, or dataset versions with alternative targets. (4) Remove different datasets representing the same task from the same source (i.e., a collection of ML for software tasks named kc1-3).

**IID Tabular Data:** We exclude datasets that are non-IID. More specifically, we exclude datasets whose tasks require a non-random split, such as a temporal or group-based split. We leave a non-IID realization of `TabArena` with temporal and time-series data for future work.

**Tabular Domain Tasks:** We exclude datasets from non-tabular modalities transformed into a tabular format. Thus, we exclude featureized image, text, audio, or time series forecasting data. Likewise, we exclude problems that would no longer be solved with tabular machine learning, such as tabular data of control problems solved nowadays by reinforcement learning. While some tasks from other modalities may still be solved using feature extraction and tabular learning methods, it is impossible to assess that without domain experts. Instead of making uninformed decisions, we exclude all datasets from other domains for `TabArena`-v0.1. In future versions, we consider adding datasets from other domains if there is evidence that tabular learning methods are still a reasonable solution for the task. Therefore, we actively invite researchers from other domains to share datasets for which they apply tabular learning methods.

**Real Random Distribution:** We exclude purely artificial data, or any subset thereof, generated by a deterministic function, by sampling from a seeded random process, or by simulating a random distribution. We note that such datasets are still interesting toy functions that help analyze the theoretical capabilities of models qualitatively. Yet, they do not represent distributions from real-world predictive machine learning tasks. While some simulated datasets (i.e., higgs, or MiniBooNE) were conceptualized as machine learning tasks, we decided to exclude them for `TabArena`-v0.1 for consistency.

**Predictive Machine Learning Task:** We exclude tabular data that does not originally stem from a predictive machine learning task for classification or regression. Thus, we exclude tabular data intended for *scientific discovery* tasks such as anomaly detection, subgroup discovery, data visualization, or causal inference. In particular, this includes survey data never intended for use in a predictive machine learning task. While data from scientific discovery applications can be used for predictive machine learning tasks, we only include it if the original data source intended its use for predictive machine learning, or if a follow-up work re-used the data in a real-world application.

Moreover, we exclude non-predictive tables, where the target label is not predictable based on statistical information from other columns, such as those commonly found in collections like WikiTables [63] or GitTables [64].

We exclude datasets that are trivial to solve and therefore do not represent challenging ML tasks allowing to investigate model differences. We define trivial datasets as datasets where one of the following criteria applies: (1) at least one of the models in our scope is consistently able to achieve perfect performance; (2) multiple models achieve exactly the same highest performance. Note that after applying our set of criteria, none of the considered datasets was found to be trivial.

**Size Limit:** We exclude datasets that are tiny or large because they tend to require fundamentally different methods. Tiny datasets require a methodological focus on avoiding overfitting, while methods for large datasets must be very efficient during training. We aim to include tiny and large datasets with dedicated evaluation protocols in future versions of `TabArena`. For `TabArena`-v0.1, we exclude datasets with fewer than 500 or more than 250,000 samples, measured as the number of training samples after applying our train-test splits. Note that after applying our whole set of criteria, none of the datasets was excluded solely due to being too large, while many datasets were excluded due to a small sample size.

**Data Quality:** We exclude datasets that suffer from one of the following data quality issues: (1) heavily preprocessed datasets, such as those where the whole dataset was already used for preprocessing in a way that leaks the target or any information about the feature distribution of the test set (e.g., PCA features computed over all data points); (2) datasets for which we could not find sufficient information to judge their source and preprocessing; (3) datasets with an irreversible target leak. In general, we try to find the original state of the dataset and include it, if applicable. We do not generally exclude preprocessed datasets, as datasets are rarely published without any preprocessing, e.g., due to anonymization. We leave a benchmark with model-specific, domain-specific pre-processing per dataset for future work.

**No License Issues:** We exclude any dataset whose license does not allow sharing or using it for an academic benchmark. By doing so, we guard the future of `TabArena` as a living benchmark, its maintainers, and, most importantly, its users from legal threat.
As a result, we exclude several promising datasets, e.g., due to the default license of Kaggle competitions. Thus, progress towards a less data-restrictive license on Kaggle could greatly benefit the academic community. Likewise, any progress towards sharing more public domain datasets for tabular predictive machine learning would be highly beneficial.

**Open-access Structured Data API:** We exclude datasets that cannot be made to be automatically downloaded from a tabular data repository. Eligible data repositories must be open-access, i.e., users do not need an account to download data. Furthermore, the repositories require a structured data and task representation, including metadata information such as feature types, the target column, and outer splits. To the best of our knowledge, only OpenML [62] fulfills these requirements so far. If applicable due to licensing, we manually upload datasets to OpenML to include them in `TabArena`. This criterion is necessary to enable automated benchmarking and a straightforward user experience.

**Ethically Unambiguous Tasks:** We exclude datasets with tasks that pose ethical concerns, such as the Boston Housing dataset[2]. While curating our datasets, we flagged such datasets and excluded them. We implore the community to investigate our curated datasets for ethical concerns further and immediately notify the maintainers of `TabArena` about potential problems.

### B.2 Included Datasets Details

Table B.1 presents the domain coverage of all datasets included in `TabArena-v0.1`.
Table B.2 presents a detailed overview for all datasets included in `TabArena-v0.1`. We further share all tasks and datasets as an OpenML suite [29, 61, 62] (ID 457, alias "tabarena-v0.1"). We share BibTex and LaTeX code to easily reference datasets from `TabArena` in our data curation repository: tabarena.ai/data-tabular-ml-iid-study.

### B.3 Noteworthy Observations from Curation

We observed several trends while curating the datasets for `TabArena-v0.1`. To improve the discussion related to datasets in our community, we share some noteworthy trends below.

- For various datasets, it was not possible to automate the selection process, because the metadata that would be required is not available. Therefore, given the current state of data repositories, we consider that automated curation procedures produce more biased results than careful manual curation. Finding out which splits are appropriate for a task, or whether the targets were created using deterministic functions, requires substantial effort and oftentimes, reading and understanding the papers where datasets were introduced. To still make the inclusion of datasets as objective as possible, we introduce a checklist for new datasets in Appendix E.3.

- Most of the datasets excluded due to license issues were Kaggle datasets with restrictive licenses, which otherwise would have been well-suited for inclusion. In the future, we hope that more high-quality datasets with less restrictive licenses will become available, also on Kaggle.

---

[2]`https://fairlearn.org/main/user_guide/datasets/boston_housing_data.html`

Table B.1: **Domain Coverage of TabArena-v0.1.** We present the domain name, along with the count and percentage of datasets from each domain, for all datasets in `TabArena`. We follow the domain names and categorization of the TALENT benchmark [36]. Note that `TabArena` does not have datasets for all categories defined by Ye et al. [36], as these categories conflict with our selection criteria.

| Domain | Count | % |
|---|---|---|
| business & marketing | 16 | 31.37% |
| finance | 8 | 15.69% |
| chemistry & material science | 6 | 11.76% |
| medical & healthcare | 6 | 11.76% |
| biology & life sciences | 5 | 9.80% |
| technology & internet | 4 | 7.84% |
| physics & astronomy | 3 | 5.88% |
| education | 1 | 1.96% |
| environmental science & climate | 1 | 1.96% |
| industry & manufacturing | 1 | 1.96% |

- The large amount of datasets from other modalities seems to be an artifact from times before the development of high-performing modality-specific approaches. At least 16 datasets were images for handwritten digit or letter recognition. As those tasks are clearly outdated, we excluded them. To be consistent, we also excluded datasets consisting of features from image data for which we were not able to assess whether the tasks are outdated. Features extracted from image data are not an exclusion criterion for datasets in `TabArena`, as long as they represent a meaningful task and tabular models are a reasonable approach to solve those tasks.

- The huge amount of tiny (fewer than 500 training samples) datasets is likely an artifact of a time when data collection was done at a much smaller scale than nowadays. Only four of the 142 tiny datasets for which we found a publication date were published later than 2010. Moreover, many of the tiny datasets seem to have originated in educational content, such as books or toy examples in tutorials.

- Several datasets used in previous benchmarking studies were originally introduced as part of a series of AutoML Challenges. Datasets in these challenges were often released (and shared) with obscured, non-meaningful names. Most of the datasets are ablated versions of other datasets, and therefore have led to unintended duplicates in existing benchmarks. Furthermore, many of those datasets were from other domains, like images or text.

- Of the 254 datasets with alternative versions listed in Figure 2, most are from the PMLB benchmark [47] and represent differently parameterized versions of artificially created datasets: 118 are Feynman equations and 62 are Friedman data generation functions.

- Throughout the benchmarks, inconsistent versions of the same datasets were used: tasks were binarized, features were removed, and sometimes even targets were changed. This can be partially attributed to the misleading versioning system of OpenML. Subsequent versions of the same datasets correspond to a different upload with the same name, not necessarily an improved version of the same datasets. Therefore, some studies reused the alternative versions of the dataset uploaded under the same name for specific studies. In gathering the datasets, we disregarded which version of a dataset was used and solely focused on names. Therefore, some alternative versions were already filtered for the set of 1053 datasets with unique names. In our benchmark, we always searched for the raw version and used the dataset with minimal preprocessing.

- After applying all other criteria, only 51 datasets were found to satisfy the IID criterion, while 68 did not. This underscores the findings of Rubachev et al. [10] that all previous benchmarks used random splits inappropriately. `TabArena` aims to end this malpractice.

- A large number of datasets are tabular but were not intended to be used for predictive tasks. Most of these datasets were filtered due to being 'scientific discovery' tasks, some due to quality issues. In general, some of these datasets might still be useful for benchmarking if they represent realistic distributions and target functions. However, most of the datasets

Table B.2: **Datasets included in TabArena-v0.1.** 'Dataset (Task) ID' represents the OpenML dataset and task IDs, 'name' the dataset name, and Ref. the reference corresponding to the dataset. 'N' represents the no. of samples, 'd' the no. of features, 'C' the no. of classes (- for regression tasks), and '% cat' represents the percentage of features that are categorical. 'Subset' indicates whether the dataset has been used for the sub-benchmarks focusing on TabPFNv2 (left) and TabICL (right).

| Dataset (Task) ID | Name | Ref. | N | d | C | % cat | Subset |
|---|---|---|---|---|---|---|---|
| 46913 (363621) | blood-transfusion-service-center | [65] | 748 | 5 | 2 | 20.0 | ✓ \| ✓ |
| 46921 (363629) | diabetes | [66] | 768 | 9 | 2 | 11.11 | ✓ \| ✓ |
| 46906 (363614) | anneal | [67] | 898 | 39 | 5 | 84.62 | ✓ \| ✓ |
| 46954 (363698) | QSAR_fish_toxicity | [68] | 907 | 7 | - | 0.0 | ✓ \| ✗ |
| 46918 (363626) | credit-g | [69] | 1000 | 21 | 2 | 66.67 | ✓ \| ✓ |
| 46941 (363685) | maternal_health_risk | [70] | 1014 | 7 | 3 | 14.29 | ✓ \| ✓ |
| 46917 (363625) | concrete_compressive_strength | [71] | 1030 | 9 | - | 0.0 | ✓ \| ✗ |
| 46952 (363696) | qsar-biodeg | [72] | 1054 | 42 | 2 | 14.29 | ✓ \| ✓ |
| 46931 (363675) | healthcare_insurance_expenses | [73] | 1338 | 7 | - | 42.86 | ✓ \| ✗ |
| 46963 (363707) | website_phishing | [74] | 1353 | 10 | 3 | 100.0 | ✓ \| ✓ |
| 46927 (363671) | Fitness_Club | [75] | 1500 | 7 | 2 | 57.14 | ✓ \| ✓ |
| 46904 (363612) | airfoil_self_noise | [76] | 1503 | 6 | - | 16.67 | ✓ \| ✗ |
| 46907 (363615) | Another-Dataset-on-used-Fiat-500 | [77] | 1538 | 8 | - | 12.5 | ✓ \| ✗ |
| 46980 (363711) | MIC | [78] | 1699 | 112 | 8 | 84.82 | ✓ \| ✓ |
| 46938 (363682) | Is-this-a-good-customer | [79] | 1723 | 14 | 2 | 64.29 | ✓ \| ✓ |
| 46940 (363684) | Marketing_Campaign | [80] | 2240 | 26 | 2 | 34.62 | ✓ \| ✓ |
| 46930 (363674) | hazelnut-spread-contaminant-detection | [81] | 2400 | 31 | 2 | 3.23 | ✓ \| ✓ |
| 46956 (363700) | seismic-bumps | [82] | 2584 | 16 | 2 | 25.0 | ✓ \| ✓ |
| 46958 (363702) | splice | [83] | 3190 | 61 | 3 | 100.0 | ✓ \| ✓ |
| 46912 (363620) | Bioresponse | [84] | 3751 | 1777 | 2 | 0.06 | ✗ \| ✗ |
| 46933 (363677) | hiva_agnostic | [85] | 3845 | 1618 | 3 | 100.0 | ✗ \| ✗ |
| 46960 (363704) | students_dropout_and_academic_success | [86] | 4424 | 37 | 3 | 48.65 | ✓ \| ✓ |
| 46915 (363623) | churn | [87] | 5000 | 20 | 2 | 25.0 | ✓ \| ✓ |
| 46953 (363697) | QSAR-TID-11 | [88] | 5742 | 1025 | - | 0.0 | ✗ \| ✗ |
| 46950 (363694) | polish_companies_bankruptcy | [89] | 5910 | 65 | 2 | 1.54 | ✓ \| ✓ |
| 46964 (363708) | wine_quality | [90] | 6497 | 13 | - | 7.69 | ✓ \| ✗ |
| 46962 (363706) | taiwanese_bankruptcy_prediction | [91] | 6819 | 95 | 2 | 1.05 | ✓ \| ✓ |
| 46969 (363689) | NATICUSdroid | [92] | 7491 | 87 | 2 | 100.0 | ✓ \| ✓ |
| 46916 (363624) | coil2000_insurance_policies | [93] | 9822 | 86 | 2 | 4.65 | ✓ \| ✓ |
| 46911 (363619) | Bank_Customer_Churn | [94] | 10000 | 11 | 2 | 45.45 | ✓ \| ✓ |
| 46932 (363676) | heloc | [95] | 10459 | 24 | 2 | 4.17 | ✓ \| ✓ |
| 46979 (363712) | jm1 | [96] | 10885 | 22 | 2 | 4.55 | ✓ \| ✓ |
| 46924 (363632) | E-CommereShippingData | [97] | 10999 | 11 | 2 | 45.45 | ✓ \| ✓ |
| 46947 (363691) | online_shoppers_intention | [98] | 12330 | 18 | 2 | 44.44 | ✓ \| ✓ |
| 46937 (363681) | in_vehicle_coupon_recommendation | [99] | 12684 | 25 | 2 | 88.0 | ✓ \| ✓ |
| 46942 (363686) | miami_housing | [100] | 13776 | 16 | - | 6.25 | ✓ \| ✗ |
| 46935 (363679) | HR_Analytics_Job_Change_ of_Data_Scientists | [101] | 19158 | 13 | 2 | 76.92 | ✗ \| ✓ |
| 46934 (363678) | houses | [102] | 20640 | 9 | - | 0.0 | ✗ \| ✗ |
| 46961 (363705) | superconductivity | [103] | 21263 | 82 | - | 0.0 | ✗ \| ✗ |
| 46919 (363627) | credit_card_clients_default | [104] | 30000 | 24 | 2 | 16.67 | ✗ \| ✓ |
| 46905 (363613) | Amazon_employee_access | [105] | 32769 | 10 | 2 | 100.0 | ✗ \| ✓ |
| 46910 (363618) | bank-marketing | [106, 107] | 45211 | 14 | 2 | 57.14 | ✗ \| ✓ |
| 46928 (363672) | Food_Delivery_Time | [108] | 45451 | 10 | - | 30.0 | ✗ \| ✗ |
| 46949 (363693) | physiochemical_protein | [109] | 45730 | 10 | - | 0.0 | ✗ \| ✗ |
| 46939 (363683) | kddcup09_appetency | [110] | 50000 | 213 | 2 | 18.31 | ✗ \| ✓ |
| 46923 (363631) | diamonds | [111] | 53940 | 10 | - | 30.0 | ✗ \| ✗ |
| 46922 (363630) | Diabetes130US | [112] | 71518 | 48 | 2 | 83.33 | ✗ \| ✓ |
| 46908 (363616) | APSFailure | [113] | 76000 | 171 | 2 | 0.58 | ✗ \| ✓ |
| 46955 (363699) | SDSS17 | [114] | 78053 | 12 | 3 | 25.0 | ✗ \| ✓ |
| 46920 (363628) | customer_satisfaction_in_airline | [115] | 129880 | 22 | 2 | 77.27 | ✗ \| ✓ |
| 46929 (363673) | GiveMeSomeCredit | [116] | 150000 | 11 | 2 | 9.09 | ✗ \| ✓ |

filtered due to this criterion appeared to be relatively simple tasks. That is, some were already found to be trivially solvable in other studies, and some contained only a few features. In the future, we are open to considering including datasets not initially intended for predictive tasks, if no other issues are found, and if one can argue for potential predictive machine learning applications.

# C  Model Curation

## C.1  Implementation Framework Details

We implement models (and their unit tests) based on the `AbstractModel` framework[3] from Auto-Gluon [19]. In particular, we implement model-specific preprocessing, training, and inference within the `AbstractModel` framework for all models. The framework allows us to use all functionalities from AutoGluon, TabRepo, and in extension scikit-learn [24] to run models in a standardized way. Moreover, the pipeline logic encompassing models within `TabArena` is implemented in a tested, sophisticated framework that is regularly used in real-world applications.

To integrate models in `AbstractModel` framework, we require two properties of a model implementation: (**I**) Iteratively trained models (e.g., GBDTs or MLPs) must support early stopping based on a time limit. Moreover, they must support the use of externally provided validation data. (**II**) We require a default model-specific preprocessing pipeline that handles, if needed, data anomalies such as NaN values, categorical features, or feature scaling. The model-agnostic preprocessing of the `AbstractModel` framework detects categorical features, transforms text or image features, and cleans common data problems.
The original implementations of some models do not fulfill these requirements; thus, we added support ourselves or together with the model authors. Our requirements aim to get the most out of models in a proper benchmark and in real-world pipelines. Early stopping based on a time limit avoids model failures due to time constraints in benchmarks and is quintessential for integration into time-constrained, real-world pipelines. Likewise, only models that support externally provided validation data can be properly used in pipelines with pre-defined validation splits. Finally, different models require different preprocessing, and relying only on one shared model-agnostic preprocessing pipeline is inappropriate. We detail the model-agnostic and model-specific below.

**Foundation Model Implementation Details.**  For all foundation models, we refit on training and validation data instead of using cross-validation ensembles, following recommendations from the authors of TabPFN and TabICL. We hypothesize that the foundation models do not gain much from cross-validation ensembles because, unlike other models, they do not utilize the validation data per fold for early stopping during training. Thus, their in-context learning might benefit more from using the training and validation data as context for inference on test data.
The foundation models TabPFNv2 and TabICL have been released with restrictions in terms of dataset size. In particular, TabPFNv2 is restricted to datasets with up to $10,000$ training samples, $500$ features, and $10$ classes for classification tasks. TabICL is constrained to classification tasks with up to $100,000$ training samples and $500$ features. TabDPT has no size restrictions because it natively relies on context retrieval, dimensionality reduction, and class codes during inference [23]. For context retrieval, we use the default context size of $1024$ described in the paper. Thereby, we override the implementation's default of $128$, which we found to perform poorly in preliminary experiments. We use the newest available checkpoints for all foundation models.  For TabDPT, we use `tabdpt1_1.pth`.  For TabICL, we use `tabicl-classifier-v1.1-0506.ckpt`.  For TabPFN, we use the defaults for classification `tabpfn-v2-classifier.ckpt` and regression `tabpfn-v2-regressor.ckpt`, as well as all other checkpoints during HPO: `tabpfn-v2-classifier-gn2p4bpt.ckpt`, `tabpfn-v2-classifier-llderlii.ckpt`, `tabpfn-v2-classifier-od3j1g5m.ckpt`, `tabpfn-v2-classifier-vutqq28w.ckpt`, `tabpfn-v2-classifier-znskzxi4.ckpt`, `tabpfn-v2-regressor-09gpqh39.ckpt`, `tabpfn-v2-regressor-2noar4o2.ckpt`, `tabpfn-v2-regressor-5wof9ojf.ckpt`, `tabpfn-v2-regressor-wyl4o83o.ckpt`.

**Model-agnostic Preprocessing.**  Our model-agnostic preprocessing relies on AutoGluon's `AutoMLPipelineFeatureGenerator`[4]. The model-agnostic preprocessing can handle boolean, numerical, categorical, datetime, and text columns. Importantly, the implementation of a model can control how the model-agnostic preprocessing treats the input data. As a result, a model could obtain raw text and datetime columns as input, such that its model-specific preprocessing can handle them. For `TabArena`, we let the model-agnostic preprocessing handle text and datetime columns. Text

---

[3]`https://auto.gluon.ai/stable/tutorials/tabular/advanced/tabular-custom-model.html`

[4]`https://auto.gluon.ai/stable/tutorials/tabular/tabular-feature-engineering.html`

columns are transformed to n-hot encoded n-grams. Datetime columns are converted into a Pandas datetime and into multiple columns representing the year, month, day, and day of the week. Numerical columns are left untouched. Categorical columns are replaced with categorical codes to save memory space. The columns are, nevertheless, further treated as categorical. Finally, constant or duplicated columns are dropped. Importantly, we always keep missing values and delegate handling them to the model-specific preprocessing.

**Model-specific Preprocessing.**    We perform minimal invasive model-specific preprocessing and otherwise rely on the preprocessing already implemented within the model's code. Specifically, we use the following model-specific preprocessing before passing the data to the model's code:

- **CatBoost**, **LightGBM**, **XGBoost**, **EBM**, **TabICL**, **TabPFNv2**, **FastaiMLP**, and **TorchMLP** do not use any custom model-specific preprocessing and rely entirely on the model's code.

- **RandomForest** and **ExtraTrees** use ordinal encoding for categorical variables. Missing values are imputed to $0$.

- **TabDPT** uses ordinal encoding for categorical variables.

- **RealMLP** handles missing numericals by mean imputation with a missingness indicator.

- **TabM** and **ModernNCA** use the numerical quantile-based preprocessing from TabM and then use mean imputation with an indicator for numerical features.

- **Linear** uses one-hot-encoding, mean or median imputation (hyperparameter), and standard scaling or quantile transformation (hyperparameter).

- **KNN** uses different numerical and categorical feature encoding techniques as part of the search space. For numerical features, either z-standardization or quantile transformation is used. For categorical features, a 'cat_threshold' parameter is defined determining whether categorical features are dropped, ordinal-encoded, or one-hot-encoded. If the no. of unique values is below the threshold, a feature is one-hot-encoded and otherwise ordinal-encoded. A value of $0$ means that all categorical features are dropped. Missing numerical values are filled with $0$. Moreover, KNN uses leave-one-out cross-validation instead of 8-fold cross-validation. The leave-one-out cross-validation is natively implemented into the KNN model logic and allows for obtaining the validation predictions per sample very efficiently.

## C.2   Hyperparameter search spaces

In the following, we list some details and hyperparameter search spaces for different models:

- The search spaces for **CatBoost** (Table C.1), **LightGBM** (Table C.2), **XGBoost** (Table C.3), **RandomForest** (Table C.4), and **ExtraTrees** (Table C.5) were determined based on experiments. We assessed a large set of hyperparameters inspired by the respective documentations as well as several papers [e.g., 20, 37, 117] and experimentally determined good ranges for them for tuning. We verified that the new search spaces outperform the original search spaces on TabRepo [37].

  For gradient-boosted trees, we use the implementation from AutoGluon with its early stopping logic and `n_estimators=10_000`. For Random Forest and ExtraTrees, TabRepo fit 300 trees on training+validation data and used out-of-bag predictions for validation. Since we want to allow `bootstrap=False`, which does not support out-of-bag predictions, we fit these models using 8-fold CV with 50 estimators per model, resulting in 400 estimators overall.

- For **EBM**(Table C.6), we use a search space provided by the authors.

- For **RealMLP** (Table C.7), we also use a search space provided by the authors. For the default parameters, we turn off label smoothing since we are not using accuracy as our evaluation metric, as recommended by Holzmüller et al. [20].

- For **TabM** (Table C.8) and **ModernNCA** (Table C.9), we use search spaces coordinated with the authors. For the batch size, we choose a training-set-size dependent logic following

the TabM paper [9]:

$$\text{batch\_size} = \begin{cases} 32 & , N < 2800 \\ 64 & , N \in [2800, 4500) \\ 128 & , N \in [4500, 6400) \\ 256 & , N \in [6400, 32000) \\ 512 & , N \in [32000, 108000) \\ 1024 & , N \geq 108000 \, . \end{cases}$$

- **FastaiMLP** (Table C.10) and **TorchMLP** (Table C.11) were taken from AutoGluon, in dialogue with the maintainers/authors.

- **Linear** (Table C.12) and **KNN** (Table C.13) were taken from TabRepo. For KNN additional preprocessing was added to deal with varying numerical feature distributions and categorical features. For Linear, we used a log-uniform instead of uniform search space for the regularization parameter `C`.

- For **TabPFNv2**, we use the search space from the original paper and the official repository, in coordination with the authors.

- For **TabICL** and **TabDPT**, we only use their default configurations. For TabICL and TabDPT, we use the newest checkpoint (see Appendix C.1), unlike the original paper.

Table C.1: Hyperparameter search space for CatBoost.

| Hyperparameter | Space |
| --- | --- |
| learning_rate | LogUniform([0.005, 0.1]) |
| bootstrap_type | Bernoulli |
| subsample | Uniform([0.7, 1.0]) |
| grow_policy | Choice(["SymmetricTree", "Depthwise"]) |
| depth | UniformInt([4, 8]) |
| colsample_bylevel | Uniform([0.85, 1.0]) |
| l2_leaf_reg | LogUniform([1e-4, 5]) |
| leaf_estimation_iterations | LogUniformInt([1, 20]) |
| one_hot_max_size | LogUniformInt([8, 100]) |
| model_size_reg | LogUniform([0.1, 1.5]) |
| max_ctr_complexity | UniformInt([2, 5]) |
| boosting_type | Plain |
| max_bin | 254 |

Table C.2: Hyperparameter search space for LightGBM.

| Hyperparameter | Space |
| --- | --- |
| learning_rate | LogUniform([0.005, 0.1]) |
| feature_fraction | Uniform([0.4, 1.0]) |
| bagging_fraction | Uniform([0.7, 1.0]) |
| bagging_freq | 1 |
| num_leaves | LogUniformInt([2, 200]) |
| min_data_in_leaf | LogUniformInt([1, 64]) |
| extra_trees | Choice([False, True]) |
| min_data_per_group | LogUniformInt([2, 100]) |
| cat_l2 | LogUniform([0.005, 2]) |
| cat_smooth | LogUniform([0.001, 100]) |
| max_cat_to_onehot | LogUniformInt([8, 100]) |
| lambda_l1 | Uniform([1e-4, 1.0]) |
| lambda_l2 | Uniform([1e-4, 2.0]) |

Table C.3: Hyperparameter search space for XGBoost.

| Hyperparameter | Space |
|---|---|
| learning_rate | LogUniform([0.005, 0.1]) |
| max_depth | LogUniformInt([4, 10]) |
| min_child_weight | LogUniform([0.001, 5.0]) |
| subsample | Uniform([0.6, 1.0]) |
| colsample_bylevel | Uniform([0.6, 1.0]) |
| colsample_bynode | Uniform([0.6, 1.0]) |
| reg_alpha | Uniform([1e-4, 5.0]) |
| reg_lambda | Uniform([1e-4, 5.0]) |
| grow_policy | Choice(["depthwise", "lossguide"]) |
| max_cat_to_onehot | LogUniformInt([8, 100]) |
| max_leaves | LogUniformInt([8, 1024]) |

Table C.4: Hyperparameter search space for Random Forest.

| Hyperparameter | Space |
|---|---|
| max_features | Uniform([0.4, 1.0]) |
| max_samples | Uniform([0.5, 1.0]) |
| min_samples_split | LogUniformInt([2, 4]) |
| bootstrap | Choice([False, True]) |
| n_estimators | 50 |
| min_impurity_decrease | LogUniform([1e-5, 1e-3]) |

Table C.5: Hyperparameter search space for ExtraTrees.

| Hyperparameter | Space |
|---|---|
| max_features | Choice(["sqrt", 0.5, 0.75, 1.0]) |
| min_samples_split | LogUniformInt([2, 32]) |
| bootstrap | False |
| n_estimators | 50 |
| min_impurity_decrease | Choice([0.0, 1e-5, 3e-5, 1e-4, 3e-4, 1e-3], p=[0.5, 0.1, 0.1, 0.1, 0.1, 0.1]) |

Table C.6: Hyperparameter search space for EBM.

| Hyperparameter | Space |
|---|---|
| max_leaves | UniformInt([2, 3]) |
| smoothing_rounds | UniformInt([0, 1000]) |
| learning_rate | LogUniform([0.0025, 0.2]) |
| interactions | Uniform([0.95, 0.999]) |
| interaction_smoothing_rounds | UniformInt([0, 200]) |
| min_hessian | LogUniform([1e-10, 1e-2]) |
| min_samples_leaf | UniformInt([2, 20]) |
| validation_size | Uniform([0.05, 0.25]) |
| early_stopping_tolerance | LogUniform([1e-10, 1e-5]) |
| gain_scale | LogUniform([0.5, 5.0]) |

Table C.7: Hyperparameter search space for RealMLP. With probability 0.5, either the "Default" or the "Large" option is chosen for each configuration.

| Hyperparameter | Space | |
|---|---|---|
| n_hidden_layers | UniformInt([2, 4]) | |
| hidden_sizes | rectangular | |
| hidden_width | Choice([256, 384, 512]) | |
| p_drop | Uniform([0.0, 0.5]) | |
| act | mish | |
| plr_sigma | LogUniform([1e-2, 50]) | |
| sq_mom | $1 - \text{LogUniform}([0.005, 0.05])$ | |
| plr_lr_factor | LogUniform([0.05, 0.3]) | |
| scale_lr_factor | LogUniform([2.0, 10.0]) | |
| first_layer_lr_factor | LogUniform([0.3, 1.5]) | |
| ls_eps_sched | coslog4 | |
| ls_eps | LogUniform([0.005, 0.1]) | |
| lr | LogUniform([0.02, 0.3]) | |
| wd | LogUniform([0.001, 0.05]) | |
| use_ls | Choice([False, True]) | |
| early_stopping_additive_patience | 40 | |
| early_stopping_multiplicative_patience | 3 | |
| | Default (prob=0.5) | Large (prob=0.5) |
| plr_hidden_1 | 16 | Choice([8, 16, 32, 64]) |
| plr_hidden_2 | 4 | Choice([8, 16, 32, 64]) |
| n_epochs | 256 | Choice([256, 512]) |
| use_early_stopping | False | True |

Table C.8: Hyperparameter search space for TabM.

| Hyperparameter | Space |
|---|---|
| batch_size | auto |
| patience | 16 |
| amp | False |
| arch_type | tabm-mini |
| tabm_k | 32 |
| gradient_clipping_norm | 1.0 |
| share_training_batches | False |
| lr | LogUniform([1e-4, 3e-3]) |
| weight_decay | Choice([0.0, LogUniform([1e-4, 1e-1])]) |
| n_blocks | UniformInt([2, 5]) |
| d_block | Choice([128, 144, 160, ..., 1008, 1024]) |
| dropout | Choice([0.0, Uniform([0.0, 0.5])]) |
| num_emb_type | pwl |
| d_embedding | Choice([8, 12, 16, 20, 24, 28, 32]) |
| num_emb_n_bins | UniformInt([2, 128]) |

Table C.9: Hyperparameter search space for ModernNCA.

| Hyperparameter | Space |
|---|---|
| dropout | Uniform([0.0, 0.5]) |
| d_block | UniformInt([64, 1024]) |
| n_blocks | Choice([0, UniformInt([0, 2])]) |
| dim | UniformInt([64, 1024]) |
| num_emb_n_frequencies | UniformInt([16, 96]) |
| num_emb_frequency_scale | LogUniform([0.005, 10.0]) |
| num_emb_d_embedding | UniformInt([16, 64]) |
| sample_rate | Uniform([0.05, 0.6]) |
| lr | LogUniform([1e-5, 1e-1]) |
| weight_decay | Choice([0.0, LogUniform([1e-6, 1e-3])]) |
| temperature | 1.0 |
| num_emb_type | plr |
| num_emb_lite | True |
| batch_size | auto |

Table C.10: Hyperparameter search space for FastaiMLP.

| Hyperparameter | Space |
|---|---|
| layers | Choice([[200], [400], [200, 100], [400, 200], [800, 400], [200, 100, 50], [400, 200, 100]]) |
| emb_drop | Uniform([0.0, 0.7]) |
| ps | Uniform([0.0, 0.7]) |
| bs | Choice([128, 256, 512, 1024, 2048]) |
| lr | LogUniform([5e-4, 1e-1]) |
| epochs | UniformInt([20, 50]) |

Table C.11: Hyperparameter search space for TorchMLP.

| Hyperparameter | Space |
|---|---|
| learning_rate | LogUniform([1e-4, 3e-2]) |
| weight_decay | LogUniform([1e-12, 0.1]) |
| dropout_prob | Uniform([0.0, 0.4]) |
| use_batchnorm | Choice([False, True]) |
| num_layers | UniformInt([1, 5]) |
| hidden_size | UniformInt([8, 256]) |
| activation | Choice(["relu", "elu"]) |

Table C.12: Hyperparameter search space for LinearModel.

| Hyperparameter | Space |
|---|---|
| C | LogUniform([0.1, 1000]) |
| proc.skew_threshold | Choice([0.9, 0.99, 0.999, None]) |
| proc.impute_strategy | Choice(["median", "mean"]) |
| penalty | Choice(["L2", "L1"]) |

Table C.13: Hyperparameter search space for KNN.

| Hyperparameter | Space |
|---|---|
| n_neighbors | Choice([1, 2, 3, 4, 5, 6, 7, 8, 9, 10, 11, 13, 15, 20, 30, 40, 50, 100, 200, 300, 400, 500]) |
| weights | Choice(["uniform", "distance"]) |
| p | Choice([2, 1, 1.5]) |
| scaler | Choice(["standard", "quantile"]) |
| cat_threshold | Choice([0, 1, 5, 10, 20, 30, 50, 100, 1000000]) |

## C.3 Cross-validation Ensembles

We employ 8-fold cross-validation ensembles [27] within `TabArena`. Cross-validation ensembles function by ensembling models trained on different folds of the cross-validation process. In `TabArena`, we obtain 8 fold models. We then ensemble the 8 fold models by averaging their predictions, mirroring a bagging ensemble. The best configuration in the tuned regime is selected using the average out-of-fold performance over the 8 folds.

Cross-validation ensembles are a powerful alternative to refitting to obtain a final model for deployment. The diversity from bagging the fold models can lead to better predictions. Furthermore, cross-validation ensembles are more efficient during training because there is no need to spend time refitting; however, they increase inference time.

## C.4 Post-hoc Ensembling

Post-hoc ensembling (PHE) aims to combine a set of models previously evaluated on holdout data or by cross-validation during model selection (e.g., HPO) to improve performance over any single model [43, 118]. In particular, PHE relies on using data collected while evaluating models to build its ensemble, such as predictions on the validation data.

In practice, predictive machine learning systems most often [43] combine a set of models by aggregating their predictions with a *weighted* arithmetic mean whereby the weights of the models are commonly obtained using greedy ensemble selection (GES) [1, 119]. Likewise, multiple hyperparameter configurations of an individual tabular model can be ensembled with GES, as done in TabRepo [37] or by TabPFNv2 [5].

*Post hoc* ensembling with GES has four key advantages: 1) GES is very efficient due to reusing predictions on validation data previously collected while evaluating models [43, 120]; 2) GES optimizes a *user-defined* target metric using an *anytime* algorithm; 3) the final ensemble is usually small since GES produces sparse weight vectors [43, 44]; and 4) the predictive performance of post-hoc ensembling with GES is superior to the best individual model under mild assumptions [43, 44, 120, 121].

We build an ensemble of models using GES. In detail, we create an ensemble using the 200 hyperparameter configurations that were evaluated during the tuning process. To train the ensemble, we reuse the predictions on validation data that were computed during (inner) cross-validation. Then, we obtain a weight vector using GES [1, 119] for all configurations. Finally, we return the *weighted* average predictions of all non-zero-weighted configurations.

GES learns a weight vector $W = (w_1, ..., w_m)$ to combine multiple models $f_i \in F$ from a pool of $m$-many models $F = (f_1, ..., f_m)$ as $\sum_i w_i f_i$. GES ensures that $\forall i, 0 \leq w_i \leq 1$ and $\sum_i w_i = 1$. The vector $W$ is learned via a greedy algorithm that runs for a fixed number of iterations $N$ ($N = 40$ for TabArena). In each step $n \leq N$, GES finds $i$ such that increasing $w_i$ by $\frac{1-w_i}{n+1}$ and decreasing all other weights by $\frac{w_j}{n+1}, \forall j \neq i$ most reduces the validation error.

## C.5 TabArena Ensemble

The `TabArena` ensemble highlighted in Figure 7 was created by ensembling a portfolio, a set of hyperparameter configurations across models. Given a portfolio, we evaluate each of its models in sequence until a time limit is reached or all models have been evaluated. Then, we post-hoc ensemble [1] all evaluated hyperparameter configurations. For the sake of Figure 7, we simulated the `TabArena` ensemble using the result artifacts.

We created a portfolio following the learning procedure introduced by Salinas and Erickson [37] using leave-one-dataset-out cross-validation with a portfolio of size 200 and 40 ensemble selection steps. We leave further discussion and investigation of portfolio learning with the results of `TabArena` to future work.

## D Evaluation Design Details

### D.1 Elo Confidence Intervals

Suppose that the benchmark datasets $\mathcal{B} = (D_1, \ldots, D_{51})$ are i.i.d. samples from an unknown dataset-generating distribution $P_D$. We want to compute a confidence interval for the "infinite-datasets Elo score" $\mathrm{Elo}(P_D, A)$ of a method $A$, but can only compute finite-dataset Elo scores like $\mathrm{Elo}(\mathcal{B}, A)$.

Similar to energies in physics, only relative differences between Elo scores are meaningful, as they predict win-rates between pairs of methods. We need to choose a reference point to obtain absolute Elo scores. We consider two variants:

- $\mathrm{Elo_{mean}}$: Center the mean Elo of all methods to 0.
- $\mathrm{Elo_{RF}}$: Center the Elo of default random forest to 1000.

For a ML method $A$, we compute 200 bootstrap subsamples $\tilde{\mathcal{B}}$ of $\mathcal{B}$. The 2.5% and 97.5% quantiles of $\mathrm{Elo_{mean}}(\tilde{\mathcal{B}}, A)$ yield an approximate 95% confidence interval $[L_{\mathrm{mean}}(A), U_{\mathrm{mean}}(A)]$ for $\mathrm{Elo_{mean}}(P_D, A)$. Because of

$$\mathrm{Elo_{RF}}(\mathcal{B}, A) = \mathrm{Elo_{mean}}(\mathcal{B}, A) + \Delta(\mathcal{B}), \quad \Delta(\mathcal{B}) := 1000 - \mathrm{Elo_{mean}}(\mathcal{B}, \mathrm{RF \ (default)}),$$

we report the shifted intervals $[L_A + \Delta(\mathcal{B}), U_A + \Delta(\mathcal{B})]$, which are approximate confidence intervals for

$$\mathrm{Elo_{mean}}(P_D, A) + \Delta(\mathcal{B}) \ .$$

They are not good approximate confidence intervals for $\mathrm{Elo_{RF}}(P_D, A)$, because they do not factor in the variance in the difference $\Delta(\mathcal{B}) - \Delta(P_D) = \mathrm{Elo_{mean}}(P_D, \mathrm{RF \ (default)}) - \mathrm{Elo_{mean}}(\mathcal{B}, \mathrm{RF \ (default)})$. However, this term does not depend on $A$ and therefore does not affect the relative differences of results.

**Discussion.** The use of $\mathrm{Elo_{mean}}$-based confidence intervals explains why the confidence interval for RF(default) does not have length zero: We use a shift that shifts RF(default) to 1000 on this specific benchmark $\mathcal{B}$, but would not shift it to 1000 on other benchmark datasets. We use $\mathrm{Elo_{mean}}$ instead of $\mathrm{Elo_{RF}}$ for confidence intervals because centering the mean instead of a weak method produces smaller confidence intervals for strong methods. As a consequence, plots showing confidence intervals for absolute Elo values allow stronger conclusions about significance of relative differences in Elo values.

### D.2 Sources of Randomness

The comparison and evaluation of models in `TabArena` is affected by various sources of randomness. The results in `TabArena` are affected by the following sources of randomness.

- **Model Randomness**, resulting from: initialization, training, non-deterministic computations (on GPU or due to precision), inner validation splits (e.g., for early stopping), hyperparameter configurations, and the sampling of the hyperparameter configurations.
- **Data Randomness**, resulting from: the selected datasets, the inherent sampling bias of each data set, and the partitions used for repeated outer cross-validation.
- **Evaluation Randomness**, resulting from: metric calculation, and the precision of calculating metrics when the metric is used for ranking or normalization.

We guard against data randomness affecting our results by repeating our experiments several times per dataset. We partially guard against model randomness through repeating experiments and using many random configurations. Nevertheless, we do not fully guard against it, as we use a fixed random seed for models[5] and a static set of random configurations. We guard against evaluation randomness by using 100 bootstrapping rounds and a stable Bradley-Terry Elo implementation.

---

[5]Future versions of `TabArena` will no longer use a fixed random seed.

### D.3 Environmental Impact of TabArena

We are regrettably aware of the negative environmental impact of `TabArena`. We had several discussions about trading off environmental impact and research, but we have not converged on an official conclusion to this (philosophical) topic. To share some insights from our discussions, we provide several thoughts about the positive environmental impact of `TabArena` below. We argue that some of our key contributions will, over time, offset the negative environmental impact of running a large-scale benchmark.

- We save and share the predictions (along with other metadata) of all models, allowing others to avoid wasting energy by rerunning our experiments to obtain the predictions for future studies.
- We only simulate post-hoc ensembling on the saved predictions and thus save the computation overhead that could come from post-hoc ensembling, cf. [43, 44, 121].
- We impose time limits on the training time of a model (per split). Thus, we avoid running configurations of models that would potentially take a very long time to converge, without improving predictive performance. A tighter time limit would save more energy.
- The large (one-time) cost of running `TabArena` enables us to find more efficient and better models. Moreover, `TabArena` enables us to identify portfolios that improve the Pareto frontier in terms of both quality and efficiency. These portfolio configurations can be (and already are) implemented in predictive machine learning systems widely used in industry (e.g., AutoGluon), with applications that result in compute usage far exceeding that of the `TabArena` benchmark.

As a result of these contributions, the negative environmental impact of `TabArena` may be offset through its future applications, resulting in a net reduction of compute usage and a positive environmental impact.

## E  Using and Contributing to the Living Benchmark

### E.1 Benchmarking with TabArena

To benchmark a model, a user must (1) implement their model in the `AbstractModel` framework; (2) create a search space; (3) run the implementation on `TabArena` or `TabArena-Lite`; (4) and analyze the results. We provide code and more detailed documentation for these three steps in our code repositories with examples: tabarena.ai/code-examples. Below, we provide a snapshot[6] of code snippets for each step: model implementation (Listing 1), search space (Listing 2), benchmarking (Listing 3), and analysis of the results (Listing 4).

Listing 1: Implementing a custom RandomForest model for TabArena.

```python
import numpy as np
import pandas as pd
from autogluon.core.models import AbstractModel
from autogluon.features import LabelEncoderFeatureGenerator

class CustomRandomForestModel(AbstractModel):
    ag_key = "CRF"
    ag_name = "CustomRF"

    def __init__(self, **kwargs):
        super().__init__(**kwargs)
        self._feature_generator = None

    def _preprocess(self, X: pd.DataFrame, is_train=False, **kwargs)
            -> np.ndarray:
        """Model-specific preprocessing of the input data."""
        X = super()._preprocess(X, **kwargs)
        if is_train:
```

---

[6]Parts of this snapshot may become outdated due to the benchmarking system being updated.

```
18              self._feature_generator = LabelEncoderFeatureGenerator(
                    verbosity=0)
19              self._feature_generator.fit(X=X)
20          if self._feature_generator.features_in:
21              X = X.copy()
22              X[self._feature_generator.features_in] = self.
                    _feature_generator.transform(
23                  X=X
24              )
25          return X.fillna(0).to_numpy(dtype=np.float32)
26
27      def _fit(self, X, y, **kwargs):
28          from sklearn.ensemble import RandomForestRegressor,
                RandomForestClassifier
29          if self.problem_type in ["regression"]:
30              model_cls = RandomForestRegressor
31          else:
32              model_cls = RandomForestClassifier
33
34          X = self.preprocess(X, is_train=True)
35          self.model = model_cls(**self._get_model_params())
36          self.model.fit(X, y)
```

Listing 2: Creating a search space for the custom RandomForest model.

```
1  def get_configs_for_custom_rf(num_random_configs):
2      from autogluon.common.space import Int
3      from tabarena.utils.config_utils import ConfigGenerator
4
5      gen_custom_rf = ConfigGenerator(
6          model_cls=CustomRandomForestModel,
7          manual_configs=[{}],
8          search_space= {
9              "n_estimators": Int(4, 50),
10         },
11     )
12     return gen_custom_rf.generate_all_bag_experiments(
13         num_random_configs=num_random_configs
14     )
```

Listing 3: Benchmarking the custom RandomForest model.

```
1  import openml
2  from tabarena.benchmark.experiment import run_experiments_new
3
4  task_ids = openml.study.get_suite("tabarena-v0.1").tasks
5  methods = get_configs_for_custom_rf(num_random_configs=1)
6
7  run_experiments_new(
8      output_dir="/path/to/output/dir",
9      model_experiments=methods,
10     tasks=task_ids,
11     repetitions_mode="TabArena-Lite",
12 )
```

Listing 4: Comparing the custom RandomForest model to the leaderboard.

```
1  import pandas as pd
2  from tabarena.paper.paper_runner_tabarena import PaperRunTabArena
3
4  from . import post_process_local_results
5  from . import load_local_results, load_paper_reuslts
6
7  repo = post_process_local_results()
8  plotter = PaperRunTabArena(repo=repo, output_dir=EVAL_DIR)
```

```
 9
10  df_results = load_local_results(plotter)
11  df_results = load_paper_reuslts(df_results)
12
13  # Create and save the leaderboard figure and table
14  plotter.eval(
15      df_results=df_results,
16      framework_types_extra=list(df_results["config_type"].unique()),
17  )
```

## E.2   Contributing Models

To include a new model in `TabArena`, we ask users to open an issue on the `TabArena` benchmarking code repository (tabarena.ai/code) to start the process of adding a model. We envision this process not as a static request but as an ongoing interaction between the contributors and maintainers. During this process, the goal is to populate the issue over time with the information necessary to integrate a model. We require the following information to include a new model:

1. **Public Model Implementation.** The model must be implemented in the `AbstractModel` framework (see Appendix C.1), the code for this implementation must be publicly shared, and it must pass the default unit test for `TabArena` models. The implementation should represent a standalone model and not, for example, an ensembling pipeline of several existing models or sub-calls to other machine learning systems. We leave benchmarking for such pipelines, or in general, machine learning systems, to future iterations of `TabArena`. Finally, note that the model can also first be implemented in a scikit-learn API-like interface and then wrapped with the `AbstractModel` framework. This would be the recommended workflow in many cases.

2. **Preprocessing and Hyperparameters.** The implementation should specify model-specific preprocessing (see Appendix C.1). Moreover, the contributor must recommend default hyperparameters and a search space for hyperparameter optimization.

3. **Model Verification.** The maintainers of `TabArena` must have reviewed the source code of the model. In an ideal process, this review could also help the user to improve their model and implementation. In addition, the model should (at least) demonstrate promising results on `TabArena-Lite`. Moreover, if the contributor is not among the original authors of the model, the contributor (potentially in coordination with the maintainers of `TabArena`) shall reach out to the original authors to verify the implementation and its optimal intended usage. This may involve including the original author in GitHub issues, reviewing the pull request, or validating the results.

4. **Maintenance Commitment.** While the `TabArena` team generally maintains model implementations, we might need help from the original contributors to resolve future version conflicts or outdated functionality. Therefore, contributors must share their preferred way of being contacted. Note that the `TabArena` team may deprecate models that are no longer maintainable, consistently outperformed by newer models, or have bugs that cannot be reasonably resolved.

Once the issue is deemed finalized, two maintainers of `TabArena` need to review and approve the issue to complete the model integration.

## E.3   Contributing Data: New Datasets and Curation Feedback

We envision `TabArena` to be a platform for discussing benchmarking practices. Therefore, we invite users, researchers, and practitioners to challenge our curation decisions or provide curation feedback using GitHub issues in the TabArena curation repository: tabarena.ai/data-tabular-ml-iid-study. Moreover, we also invite the community to add new datasets and welcome any suggestions for datasets that could be included in future versions of `TabArena`. For a new dataset to be added to `TabArena`, there are two alternative processes: A maintainer-driven process and a user-driven process.

In the *maintainer-driven process*, we welcome GitHub issues with the 'Dataset Suggestion' template, which includes: (1) a link to the raw data, and (2) the dataset license. The `TabArena` maintainers will review the suggested dataset by applying the protocol outlined below and, if the criteria are met, include it in the next version of `TabArena`.

The *user-driven process* targets users with a high level of knowledge about the suggested dataset and requires users to follow our dataset inclusion template. We outline the current template below:

1. Reference to pull request with a .yaml file including a dataset description following the template in the repository.

2. Reference to a .py file containing a preprocessing pipeline to transform data from the raw data source into a format suitable for benchmarking.

3. A checklist answering the following questions

   (a) Is the data available through an API for automatic downloading, or does the license allow for reuploading the data?

   (b) What is the sample size?

   (c) Was the data extracted from another modality (i.e., text, image, time-series)

      If yes: Are tabular learning methods a reasonable solution compared to domain-specific methods? (If possible, provide a reference).

   (d) Is there a deterministic function for optimally mapping the features to the target?

   (e) Was the data generated artificially or from a parameterized simulation?

   (f) Can you provide a one-sentence user story detailing the benefits of better predictive performance in this task?

   (g) Were the samples collected over time?

      If yes: Is the task about predicting future data, and, if yes, are there distribution shifts for samples collected later?

   (h) Were the samples collected in different groups (i.e., transactions from different customers, patients from multiple hospitals, repeated experimental results from different batches)?

      If yes: Is the task about predicting samples from unseen groups, and if yes, are distribution shifts of samples from different groups expected?

   (i) Are there known preprocessing techniques already applied to the 'rawest' available data version?

   (j) What preprocessing steps are recommended to conceptualize the task in the preprocessing Python file?

   (k) Do you have any other recommendations for how to use the dataset for benchmarking?

The maintainers will verify the provided information and engage in discussions if required. After verifying that the task is reasonable, the dataset will be included in the next benchmark version.

The checklist results from our learnings during data curation and covers the essential aspects where we had to look closely at the data in our curation process. However, we want to emphasize that we do not generally exclude datasets using this checklist. On the contrary, for future versions of `TabArena`, we aim to explicitly extend the benchmark with tasks that are not covered sufficiently so far, either due to a lack of high-quality data or due to a lack of domain knowledge to judge the task quality on our end. Therefore, **we encourage users to propose datasets from other domains, non-IID data, and for any supervised learning task consisting of tabular features where strong performance is a desired property**.

### E.3.1 Checklist Examples

In the following, we provide examples of the application of our checklist to one included and one excluded dataset.

Example for the APSFailure dataset, which represents one of the borderline cases that were included in `TabArena-v0.1`:

a) Is the data available through an API for automatic downloading, or does the license allow for reuploading the data? **Yes**

b) What is the sample size? **76,000**

c) Was the data extracted from another modality (i.e., text, image, time-series)? **Unclear, as the data was anonymized. Some features represent histograms, so some of the features possibly were extracted from time-series.**

> If yes: Are tabular learning methods a reasonable solution compared to domain-specific methods? (If possible, provide reference) **The data is from a 2016 challenge and was provided by a well-known company. Given that the dataset is comparably recent and the source is legitimate, we conclude that it still represents a meaningful tabular data task.**

d) Is there a deterministic function for optimally mapping the features to the target? **No**

e) Was the data generated artificially or from a parameterized simulation? **No**

f) Can you provide a one-sentence user story detailing the benefits of a better predictive performance in this task? **By automatically detecting component failures in trucks, the company can save costly manual effort and prevent accidents from releasing trucks with faulty components.**

g) Were the samples collected over time? **Probably yes.**

> If yes: Is the task about predicting future data, and, if yes, are there distribution shifts for samples collected later? **In a real application, future data would be predicted, however, the provided test dataset revealed that no distribution shifts between train and test data can be expected as the features are time-invariant.**

h) Were the samples collected in different groups (i.e. transactions from different customers, patients from multiple hospitals, repeated experimental results from different batches)? **No**

> If yes: Is the task about predicting samples from unseen groups, and if yes, are distribution shifts of samples from different groups expected? **N/A**

i) Are there known preprocessing techniques that have already been applied to the 'rawest' available data version? **The feature names were anonymized. Some features were preprocessed.**

j) What preprocessing steps are recommended to conceptualize the task in the preprocessing Python file? **Combine the original training and test files. Convert "na" strings to real NaN/missing values for numeric features.**

k) Do you have any other recommendations for how to use the dataset for benchmarking? **The data originally comes with a cost-matrix, which could be considered in future benchmark versions.**

Example for the socmob dataset, which was excluded for `TabArena`-v0.1 as it represents a scientific discovery task where higher predictive performance is not relevant:

a) Is the data available through an API for automatic downloading, or does the license allow for reuploading the data? **Yes**

b) What is the sample size? **1156**

c) Was the data extracted from another modality (i.e., text, image, time-series) **No**

> If yes: Are tabular learning methods a reasonable solution compared to domain-specific methods? (If possible, provide reference) **N/A**

d) Is there a deterministic function for optimally mapping the features to the target? **No**

e) Was the data generated artificially or from a parameterized simulation? **No**

f) Can you provide a one-sentence user story detailing the benefits of a better predictive performance in this task? **No. The data was collected to empirically test the hypothesis that associations between socioeconomic and occupational attributes of fathers and sons among sons from intact families are stronger than associations between attributes of fathers and sons among sons from any kind of disrupted or reconstituted families. The dataset has one target and five predictive features, including the investigated family**

**structure. Although supervised (linear) models are applied to the data, the goal is not to maximize performance, but to empirically quantify the relationship of one feature to the target while controlling for confounding factors (other features).**

g) Were the samples collected over time? **No, the study was cross-sectional and collected data in 1973.**

> If yes: Is the task about predicting future data, and, if yes, are there distribution shifts for samples collected later? **N/A**

h) Were the samples collected in different groups (i.e. transactions from different customers, patients from multiple hospitals, repeated experimental results from different batches)? **No**

> If yes: Is the task about predicting samples from unseen groups, and if yes, are distribution shifts of samples from different groups expected? **N/A**

i) Are there known preprocessing techniques that have already been applied to the 'rawest' available data version? **No noteworthy steps.**

j) What preprocessing steps are recommended to conceptualize the task in the preprocessing Python file? **None.**

k) Do you have any other recommendations for how to use the dataset for benchmarking? **Do not use the data for benchmarking the capabilities of predictive modeling approaches, but maybe for a scientific discovery benchmark in the future.**

### E.4 Contributing Results: Leaderboard Submissions

We seek to define a process for `TabArena` to submit to the leaderboard that satisfies the following principles: (**1**) Equality: Submitting to the leaderboard is accessible in the same way to everyone. (**2**) Transparency: All attempts to submit to the leaderboard are transparent to the public. (**3**) Reproducibility: Submitted results are reproducible. (**4**) Fairness: Cheated results, i.e., by utilizing the test data in an inappropriate way or simply by submitting manually altered results, are rejected. (**5**) : Feasibility: The submission process, in particular the validation, must be manageable for the maintainers in a reasonable amount of time.

Using these guiding principles, we define our submission process:

1. To submit results to the leaderboard, users can write a pull request to tabarena.ai/community-results that contains:

   (a) An update to the results dataset collection with new data for their model.

   (b) Reproducible and documented code to obtain the results. We require users to start the process to add their new model to `TabArena` (as described in Appendix E.2) and to train and evaluate their approach using the provided `TabArena` benchmarking code.

   (c) A description or link to a description, e.g., a paper, for the new model.

   (d) The following statement: "I confirm that these results were produced using the attached modeling pipeline and to the best of my knowledge, I have used the test data appropriately and have not manipulated the results."

   (e) Indicate whether verification of the submitted results by the maintainers of `TabArena` is requested.

2. The maintainers will verify that all the required information is present and will proceed depending on whether verification was requested:

   (a) Non-verified submission (fast): The request will be merged without recomputing the results. Non-verified submissions will not appear on the landing page and will be presented as a separate leaderboard on tabarena.ai[7].

   (b) Verified submission: The maintainers will manually review the code and reproduce the results for a random sample of outer folds from different datasets. If the results can be reproduced successfully and no further issues are found, the request will be merged and the results will appear in the main `TabArena` leaderboard.

---

[7]Note that for `TabArena`-v0.1 no non-validated leaderboard exists on the website. This will change with the first submission from the community using this protocol.

We aim to continuously improve our submission process and welcome any feedback or suggestions for future versions of `TabArena`.

## E.5 Running TabArena Models in Practice

Models integrated into `TabArena` can be easily used to solve predictive machine learning tasks on new datasets, independent of the `TabArena` benchmark. Listing 5 shows how to run RealMLP on a toy dataset from scikit-learn. For more details on this code, please see our code repositories with examples: tabarena.ai/code-examples.

Listing 5: Running RealMLP from TabArena on a new dataset.

```python
from autogluon.core.data import LabelCleaner
from autogluon.features.generators import (
    AutoMLPipelineFeatureGenerator)
from sklearn.datasets import load_breast_cancer
from sklearn.metrics import roc_auc_score
from sklearn.model_selection import train_test_split

# Import a TabArena model
from tabrepo.benchmark.models.ag.realmlp.realmlp_model import (
    RealMLPModel)

# Get Data
X, y = load_breast_cancer(return_X_y=True, as_frame=True)
X_train, X_test, y_train, y_test = train_test_split(X, y, test_size
    =0.5, random_state=42)

# Model-agnostic Preprocessing
feature_generator, label_cleaner = AutoMLPipelineFeatureGenerator(),
    LabelCleaner.construct(problem_type="binary", y=y)
X_train, y_train = feature_generator.fit_transform(X_train),
    label_cleaner.transform(y_train)
X_test, y_test = feature_generator.transform(X_test), label_cleaner.
    transform(y_test)

# Train TabArena Model
clf = RealMLPModel()
clf.fit(X=X_train, y=y_train)

# Predict and score
prediction_probabilities = clf.predict_proba(X=X_test)
print("ROC AUC:", roc_auc_score(y_test, prediction_probabilities))
```

## E.6 Handling Foul Play and Dataset Contamination

A fundamental limitation of open-source benchmarking is that foul play and dataset contamination could compromise the leaderboard of `TabArena`. Model developers could overfit the hyperparameters of their model on the `TabArena` datasets, or use them for pretraining a foundation model.
Below, we provide a discussion on handling foul play and benchmarking foundation models with dataset contamination. In the future, we aim to explore various solutions to these challenges with `TabArena` to keep our leaderboard representative for practitioners.

**Avoiding Foul Play.** Foul play will inevitably affect TabArena. Thus, we, as maintainers, have considered future guards against foul play in four ways:

1. A simple mitigation is to provide leaderboards excluding models with potential contamination, and, in addition, a leaderboard that includes all models.

2. Another solution is a healthy dose of suspension by the maintainers. We will generally investigate reasons for better (or worse) performance per dataset, given model outliers. Given that model providers can train on `TabArena`'s datasets, we expect that for LLM-based approaches, model providers perform memorization tests [122]. For tabular foundation

models, we currently have no way of detecting contamination and do not know if such models can "remember" the data in a significant way, as is the case for text- or vision-based models. Here, future research and tools, akin to the work by Bordt et al. [122], are needed.

3. One significant difference between fighting foul play and data contamination in LLM benchmarks, such as ChatbotArena [39], to `TabArena` is that our official leaderboard requires an open mode (code, data, and result artifacts), which makes potential abuse easier to spot by the community. We do not want to rule out benchmarking API-based closed-source models in the future, so this might also not be a silver bullet.

4. Lastly, we believe that the living benchmark itself will detect foul play across future iterations as new datasets, potentially changing (the seed of) the dataset splits, or new tools to detect foul play will inevitably be added to the TabArena ecosystem.

A shared characteristic among the various future guards is the need to update and maintain the benchmark. Thus, we posit that the strongest protection against foul play is to keep `TabArena` alive.

**Possible Data Contamination in TabArena.** It is likely that many benchmarked models were developed or validated on datasets included in TabArena. Since most publications do not disclose this information, the extent of this issue cannot be estimated. Moreover, we do not wish to penalize methods for being transparent about their training and evaluation data. Therefore, we only discuss data contamination and not benchmark overfitting. In TabArena-v0.1, TabDPT is the only model directly pretrained on benchmark datasets, covering seven tasks (KDDCup09_appetency, QSAR-TID-11, Amazon_employee_access, APSFailure, wine_quality, Diabetes130US, heloc). Notably, the model is outperformed by others on most of these datasets, showing a clear advantage only on wine_quality. Although contamination cannot be ruled out entirely, it appears unlikely that the performance of TabDPT is unfairly overestimated compared to others. Therefore, we decided against taking immediate countermeasures. Nevertheless, as contamination could become a serious issue in the future, we will remain cautious when adding models that may have been developed using TabArena datasets.

**Benchmarking Foundation Models with Dataset Contamination.** Foundation models for tabular data are increasingly trained on real-world data, and often the pretraining data overlaps with the dataset in prior benchmarks; cf. [23, 123, 124, 125]. As a result, we inevitably expect that `TabArena` will be used to evaluate foundation models that have been trained on some or all of the datasets from `TabArena`. While we could exclude such models and ignore the problem, this would contradict our goal of benchmarking the state-of-the-art in machine learning on tabular data. Thus, we must consider a future in which we incorporate such models into `TabArena`. Therefore, we provide a brief discussion on approaches for benchmarking foundation models with dataset contamination below.

The most straightforward approach is to inform practitioners that dataset contamination might exist. For instance, `TabArena` could follow the lead of GIFT-Eval [56], which recently introduced a boolean "TestData Leakage" column to the leaderboard. Likewise, one could provide leaderboards with and without models suspected of dataset contamination.

An alternative, more aggressive approach could be to require models that are submitted not to be trained on the datasets from `TabArena`. The main problem with requiring model developers to provide a "leak-free" version of their foundation models is that we would not benchmark the model used in practice. Thus, our benchmark would not be helpful to practitioners who need to decide between specific models. To explain, consider that we would use a leak-free version as a proxy; then, we need to guarantee that the checkpoint of the leak-free version performs similarly to the original checkpoint, as if one were to change only the pretraining data. However, in deep learning, we generally lack a robust method for pretraining models. In most cases, pretraining requires considerable attention for each training run, such as manually setting learning hyperparameters. Due to the complexity of deep learning, there is no efficient or straightforward way to guarantee that different checkpoints trained on different datasets are representative of each other or achieve maximum performance given their respective datasets. Thus, using a leak-free version as a proxy would most likely not result in accurately benchmarking the best version of the model that practitioners would use.

We conclude that it is debatable how best to benchmark foundation models with dataset contamination. As a minimal measure, `TabArena` could communicate the presence of dataset contamination. More effective measures require more research. Consequently, we must keep updating `TabArena` to incorporate more effective measures in the future, assuming future work can identify such measures.

# F  Performance Results Per Dataset

This section presents the performance per dataset for all methods in `TabArena`-v0.1.

Table F.1: **Performance Per Dataset.** We show the average predictive performance per dataset with the standard deviation over folds. We show the performance for the default hyperparameter configuration (`Default`), for the model after tuning (`Tuned`), and for the ensemble after tuning (`Tuned + Ens.`). We highlight the best-performing methods with significance on three levels: (1) Green: The best performing method on average; (2) **Bold**: Methods that are not significantly worse than the best method on average, based on a Wilcoxon Signed-Rank test for paired samples with Holm-Bonferroni correction and $\alpha = 0.05$. (3) Underlined: Methods that are not significantly worse than the best method in the same pipeline regime (`Default`, `Tuned`, or `Tuned + Ens.`), based on a Wilcoxon Signed-Rank test for paired samples with Holm-Bonferroni correction and $\alpha = 0.05$. We exclude AutoGluon for significance tests in the `Tuned + Ens.` regime.

### APSFailure (AUC ↑)

| | Default | Tuned | Tuned + Ens. |
|---|---|---|---|
| RF | $0.990 \pm 0.002$ | $0.990 \pm 0.002$ | $0.990 \pm 0.002$ |
| ExtraTrees | $0.990 \pm 0.002$ | $0.990 \pm 0.003$ | $0.991 \pm 0.002$ |
| XGBoost | $\mathbf{0.992 \pm 0.002}$ | $\mathbf{0.992 \pm 0.002}$ | $\mathbf{0.992 \pm 0.002}$ |
| LightGBM | $0.992 \pm 0.002$ | $0.992 \pm 0.002$ | $0.992 \pm 0.002$ |
| CatBoost | $\mathbf{0.992 \pm 0.003}$ | $0.992 \pm 0.002$ | $\mathbf{0.992 \pm 0.003}$ |
| EBM | $0.991 \pm 0.002$ | $0.991 \pm 0.002$ | $0.991 \pm 0.002$ |
| FastAIMLP | $0.988 \pm 0.003$ | $0.989 \pm 0.002$ | $0.991 \pm 0.002$ |
| TorchMLP | $0.990 \pm 0.002$ | $0.991 \pm 0.002$ | $\mathbf{0.992 \pm 0.001}$ |
| RealMLP | $0.991 \pm 0.002$ | $0.991 \pm 0.002$ | $0.992 \pm 0.002$ |
| TabM | $0.992 \pm 0.002$ | $0.992 \pm 0.002$ | $\mathbf{0.993 \pm 0.002}$ |
| MNCA | $0.991 \pm 0.003$ | $0.991 \pm 0.002$ | $\mathbf{0.993 \pm 0.002}$ |
| TabPFNv2 | - | - | - |
| TabDPT | $0.990 \pm 0.003$ | - | - |
| TabICL | $\color{green}{0.993 \pm 0.002}$ | - | - |
| Linear | $0.988 \pm 0.002$ | $0.988 \pm 0.002$ | $0.990 \pm 0.001$ |
| KNN | $0.960 \pm 0.005$ | $0.985 \pm 0.002$ | $0.990 \pm 0.002$ |
| AutoGluon | - | - | $\mathbf{0.993 \pm 0.002}$ |

### Amazon_employee_access (AUC ↑)

| | Default | Tuned | Tuned + Ens. |
|---|---|---|---|
| RF | $0.839 \pm 0.005$ | $0.841 \pm 0.005$ | $0.849 \pm 0.005$ |
| ExtraTrees | $0.833 \pm 0.006$ | $0.841 \pm 0.007$ | $0.845 \pm 0.006$ |
| XGBoost | $0.834 \pm 0.007$ | $0.859 \pm 0.008$ | $0.862 \pm 0.008$ |
| LightGBM | $0.843 \pm 0.009$ | $0.850 \pm 0.007$ | $0.858 \pm 0.009$ |
| CatBoost | $0.882 \pm 0.008$ | $\color{green}{0.883 \pm 0.008}$ | $\mathbf{0.883 \pm 0.007}$ |
| EBM | $0.839 \pm 0.006$ | $0.841 \pm 0.007$ | $0.842 \pm 0.006$ |
| FastAIMLP | $0.854 \pm 0.007$ | $0.853 \pm 0.008$ | $0.866 \pm 0.007$ |
| TorchMLP | $0.835 \pm 0.007$ | $0.838 \pm 0.006$ | $0.849 \pm 0.007$ |
| RealMLP | $0.844 \pm 0.007$ | $0.846 \pm 0.008$ | $0.864 \pm 0.008$ |
| TabM | $0.833 \pm 0.009$ | $0.842 \pm 0.009$ | $0.849 \pm 0.009$ |
| MNCA | $0.846 \pm 0.008$ | $0.860 \pm 0.008$ | $0.869 \pm 0.007$ |
| TabPFNv2 | - | - | - |
| TabDPT | $0.841 \pm 0.006$ | - | - |
| TabICL | $0.854 \pm 0.006$ | - | - |
| Linear | $0.848 \pm 0.009$ | $0.850 \pm 0.008$ | $0.851 \pm 0.008$ |
| KNN | $0.750 \pm 0.006$ | $0.817 \pm 0.006$ | $0.852 \pm 0.006$ |
| AutoGluon | - | - | $\mathbf{0.882 \pm 0.005}$ |

### Another-Dataset-on-used-Fiat-500 (rmse ↓)

| | Default | Tuned | Tuned + Ens. |
|---|---|---|---|
| RF | $750.8 \pm 28.4$ | $736.5 \pm 24.8$ | $735.4 \pm 25.3$ |
| ExtraTrees | $744.2 \pm 29.5$ | $735.8 \pm 26.5$ | $735.1 \pm 26.7$ |
| XGBoost | $754.6 \pm 23.0$ | $741.4 \pm 22.2$ | $737.1 \pm 22.6$ |
| LightGBM | $746.0 \pm 22.4$ | $740.4 \pm 24.6$ | $729.4 \pm 22.7$ |
| CatBoost | $738.1 \pm 20.8$ | $736.3 \pm 21.8$ | $732.9 \pm 21.9$ |
| EBM | $749.9 \pm 22.9$ | $750.0 \pm 24.1$ | $745.7 \pm 23.6$ |
| FastAIMLP | $760.5 \pm 17.6$ | $761.2 \pm 21.6$ | $756.3 \pm 21.4$ |
| TorchMLP | $775.0 \pm 26.3$ | $769.8 \pm 25.0$ | $765.0 \pm 24.7$ |
| RealMLP | $757.4 \pm 23.8$ | $756.0 \pm 22.4$ | $\mathbf{726.6 \pm 24.0}$ |
| TabM | $752.1 \pm 23.3$ | $755.0 \pm 25.0$ | $748.0 \pm 22.8$ |
| MNCA | $753.7 \pm 27.0$ | $748.2 \pm 27.6$ | $\mathbf{731.1 \pm 28.2}$ |
| TabPFNv2 | $727.7 \pm 23.8$ | $733.2 \pm 27.2$ | $\mathbf{727.4 \pm 26.0}$ |
| TabDPT | $\mathbf{724.0 \pm 22.1}$ | - | - |
| TabICL | - | - | - |
| Linear | $793.8 \pm 25.4$ | $764.2 \pm 19.2$ | $764.4 \pm 19.3$ |
| KNN | $823.7 \pm 28.9$ | $801.5 \pm 29.5$ | $782.2 \pm 27.7$ |
| AutoGluon | - | - | $729.6 \pm 24.6$ |

### Bank_Customer_Churn (AUC ↑)

| | Default | Tuned | Tuned + Ens. |
|---|---|---|---|
| RF | $0.851 \pm 0.008$ | $0.857 \pm 0.008$ | $0.858 \pm 0.008$ |
| ExtraTrees | $0.851 \pm 0.006$ | $0.860 \pm 0.007$ | $0.861 \pm 0.007$ |
| XGBoost | $0.864 \pm 0.009$ | $0.866 \pm 0.010$ | $0.866 \pm 0.010$ |
| LightGBM | $0.864 \pm 0.009$ | $0.866 \pm 0.009$ | $0.867 \pm 0.009$ |
| CatBoost | $0.870 \pm 0.009$ | $0.870 \pm 0.009$ | $0.870 \pm 0.009$ |
| EBM | $0.862 \pm 0.010$ | $0.864 \pm 0.010$ | $0.864 \pm 0.010$ |
| FastAIMLP | $0.859 \pm 0.009$ | $0.863 \pm 0.007$ | $0.864 \pm 0.008$ |
| TorchMLP | $0.860 \pm 0.008$ | $0.866 \pm 0.008$ | $0.866 \pm 0.008$ |
| RealMLP | $0.866 \pm 0.009$ | $0.870 \pm 0.009$ | $0.871 \pm 0.009$ |
| TabM | $0.869 \pm 0.010$ | $0.871 \pm 0.010$ | $0.871 \pm 0.010$ |
| MNCA | $0.864 \pm 0.008$ | $0.869 \pm 0.008$ | $0.869 \pm 0.008$ |
| TabPFNv2 | $0.872 \pm 0.009$ | $\color{green}{0.874 \pm 0.009}$ | $\mathbf{0.874 \pm 0.008}$ |
| TabDPT | $0.859 \pm 0.008$ | - | - |
| TabICL | $0.868 \pm 0.009$ | - | - |
| Linear | $0.772 \pm 0.010$ | $0.773 \pm 0.010$ | $0.773 \pm 0.010$ |
| KNN | $0.814 \pm 0.010$ | $0.827 \pm 0.009$ | $0.839 \pm 0.007$ |
| AutoGluon | - | - | $0.869 \pm 0.009$ |

### Bioresponse (AUC ↑)

| | Default | Tuned | Tuned + Ens. |
|---|---|---|---|
| RF | $0.873 \pm 0.007$ | $0.873 \pm 0.007$ | $0.876 \pm 0.006$ |
| ExtraTrees | $0.867 \pm 0.008$ | $0.868 \pm 0.009$ | $0.871 \pm 0.008$ |
| XGBoost | $0.873 \pm 0.008$ | $0.875 \pm 0.008$ | $\mathbf{0.876 \pm 0.008}$ |
| LightGBM | $0.872 \pm 0.008$ | $0.874 \pm 0.007$ | $0.875 \pm 0.008$ |
| CatBoost | $0.872 \pm 0.009$ | $0.875 \pm 0.008$ | $0.874 \pm 0.008$ |
| EBM | $0.852 \pm 0.008$ | $0.863 \pm 0.008$ | $0.866 \pm 0.008$ |
| FastAIMLP | $0.850 \pm 0.011$ | $0.857 \pm 0.010$ | $0.860 \pm 0.010$ |
| TorchMLP | $0.846 \pm 0.008$ | $0.856 \pm 0.009$ | $0.863 \pm 0.008$ |
| RealMLP | $0.858 \pm 0.009$ | $0.864 \pm 0.008$ | $\mathbf{0.875 \pm 0.008}$ |
| TabM | $0.863 \pm 0.005$ | $0.871 \pm 0.007$ | $0.873 \pm 0.007$ |
| MNCA | $0.860 \pm 0.010$ | $0.865 \pm 0.007$ | $0.874 \pm 0.008$ |
| TabPFNv2 | - | - | - |
| TabDPT | $0.862 \pm 0.010$ | - | - |
| TabICL | - | - | - |
| Linear | $0.789 \pm 0.011$ | $0.810 \pm 0.009$ | $0.817 \pm 0.009$ |
| KNN | $0.818 \pm 0.015$ | $0.837 \pm 0.011$ | $0.846 \pm 0.010$ |
| AutoGluon | - | - | $\color{green}{0.878 \pm 0.007}$ |

### Diabetes130US (AUC ↑)

| | Default | Tuned | Tuned + Ens. |
|---|---|---|---|
| RF | $0.631 \pm 0.006$ | $0.656 \pm 0.008$ | $0.657 \pm 0.008$ |
| ExtraTrees | $0.623 \pm 0.009$ | $0.651 \pm 0.007$ | $0.653 \pm 0.008$ |
| XGBoost | $0.662 \pm 0.008$ | $0.668 \pm 0.008$ | $0.670 \pm 0.008$ |
| LightGBM | $0.648 \pm 0.008$ | $0.668 \pm 0.007$ | $\mathbf{0.672 \pm 0.008}$ |
| CatBoost | $0.671 \pm 0.008$ | $0.671 \pm 0.008$ | $0.672 \pm 0.008$ |
| EBM | $0.659 \pm 0.008$ | $0.662 \pm 0.007$ | $0.665 \pm 0.008$ |
| FastAIMLP | $0.647 \pm 0.007$ | $0.656 \pm 0.008$ | $0.662 \pm 0.008$ |
| TorchMLP | $0.655 \pm 0.007$ | $0.663 \pm 0.009$ | $0.667 \pm 0.009$ |
| RealMLP | $0.659 \pm 0.005$ | $0.662 \pm 0.008$ | $0.669 \pm 0.008$ |
| TabM | $0.660 \pm 0.007$ | $0.662 \pm 0.008$ | $0.663 \pm 0.008$ |
| MNCA | $0.658 \pm 0.008$ | $0.662 \pm 0.008$ | $0.666 \pm 0.008$ |
| TabPFNv2 | - | - | - |
| TabDPT | $0.609 \pm 0.008$ | - | - |
| TabICL | $0.647 \pm 0.008$ | - | - |
| Linear | $0.648 \pm 0.008$ | $0.658 \pm 0.008$ | $0.659 \pm 0.008$ |
| KNN | $0.542 \pm 0.006$ | $0.633 \pm 0.005$ | $0.651 \pm 0.006$ |
| AutoGluon | - | - | $\color{green}{0.673 \pm 0.008}$ |

## E-CommereShippingData (AUC ↑)

| | Default | Tuned | Tuned + Ens. |
|---|---|---|---|
| RF | 0.739 ± 0.005 | **0.740 ± 0.006** | **0.741 ± 0.005** |
| ExtraTrees | 0.737 ± 0.006 | 0.741 ± 0.006 | 0.740 ± 0.005 |
| XGBoost | 0.740 ± 0.006 | **0.742 ± 0.007** | **0.742 ± 0.006** |
| LightGBM | 0.739 ± 0.006 | **0.740 ± 0.005** | **0.741 ± 0.006** |
| CatBoost | **0.744 ± 0.006** | 0.742 ± 0.007 | 0.741 ± 0.007 |
| EBM | **0.744 ± 0.004** | 0.743 ± 0.005 | **0.743 ± 0.004** |
| FastAIMLP | 0.737 ± 0.005 | 0.741 ± 0.007 | **0.741 ± 0.007** |
| TorchMLP | 0.737 ± 0.008 | 0.740 ± 0.007 | **0.741 ± 0.007** |
| RealMLP | **0.741 ± 0.007** | **0.742 ± 0.007** | **0.742 ± 0.007** |
| TabM | **0.744 ± 0.008** | **0.743 ± 0.005** | **0.743 ± 0.006** |
| MNCA | **0.741 ± 0.010** | 0.741 ± 0.006 | 0.741 ± 0.004 |
| TabPFNv2 | **0.744 ± 0.007** | **0.744 ± 0.007** | **0.744 ± 0.006** |
| TabDPT | 0.735 ± 0.007 | - | - |
| TabICL | **0.743 ± 0.006** | - | - |
| Linear | 0.704 ± 0.006 | 0.722 ± 0.008 | 0.722 ± 0.008 |
| KNN | 0.703 ± 0.006 | 0.736 ± 0.008 | 0.733 ± 0.005 |
| AutoGluon | - | - | **0.738 ± 0.005** |

## Fitness_Club (AUC ↑)

| | Default | Tuned | Tuned + Ens. |
|---|---|---|---|
| RF | 0.775 ± 0.018 | 0.801 ± 0.015 | 0.800 ± 0.015 |
| ExtraTrees | 0.769 ± 0.018 | 0.816 ± 0.013 | 0.815 ± 0.014 |
| XGBoost | 0.798 ± 0.015 | 0.808 ± 0.015 | 0.808 ± 0.015 |
| LightGBM | 0.795 ± 0.015 | 0.815 ± 0.015 | 0.814 ± 0.015 |
| CatBoost | 0.814 ± 0.014 | 0.812 ± 0.015 | 0.811 ± 0.015 |
| EBM | 0.813 ± 0.015 | 0.810 ± 0.015 | 0.812 ± 0.015 |
| FastAIMLP | 0.806 ± 0.013 | 0.814 ± 0.013 | 0.814 ± 0.014 |
| TorchMLP | 0.813 ± 0.016 | 0.813 ± 0.017 | 0.814 ± 0.016 |
| RealMLP | 0.812 ± 0.015 | 0.814 ± 0.015 | 0.816 ± 0.014 |
| TabM | 0.818 ± 0.014 | 0.818 ± 0.014 | 0.818 ± 0.014 |
| MNCA | 0.817 ± 0.014 | 0.814 ± 0.015 | 0.801 ± 0.017 |
| TabPFNv2 | **0.822 ± 0.012** | 0.820 ± 0.013 | 0.817 ± 0.013 |
| TabDPT | 0.818 ± 0.014 | - | - |
| TabICL | 0.819 ± 0.013 | - | - |
| Linear | 0.819 ± 0.014 | 0.819 ± 0.015 | 0.819 ± 0.015 |
| KNN | 0.722 ± 0.026 | 0.786 ± 0.016 | 0.798 ± 0.016 |
| AutoGluon | - | - | 0.811 ± 0.012 |

## Food_Delivery_Time (rmse ↓)

| | Default | Tuned | Tuned + Ens. |
|---|---|---|---|
| RF | 7.855 ± 0.041 | 7.587 ± 0.046 | 7.588 ± 0.046 |
| ExtraTrees | 8.179 ± 0.045 | 7.753 ± 0.053 | 7.749 ± 0.053 |
| XGBoost | 7.397 ± 0.055 | 7.397 ± 0.055 | 7.400 ± 0.055 |
| LightGBM | 7.616 ± 0.053 | 7.378 ± 0.054 | 7.374 ± 0.053 |
| CatBoost | 7.379 ± 0.051 | **7.367 ± 0.051** | **7.368 ± 0.051** |
| EBM | 7.433 ± 0.040 | 7.424 ± 0.051 | 7.412 ± 0.044 |
| FastAIMLP | 8.188 ± 0.053 | 8.085 ± 0.056 | 8.060 ± 0.052 |
| TorchMLP | 7.735 ± 0.059 | 7.579 ± 0.049 | 7.559 ± 0.052 |
| RealMLP | 7.926 ± 0.053 | 7.453 ± 0.051 | 7.414 ± 0.046 |
| TabM | 7.760 ± 0.050 | 7.651 ± 0.053 | 7.651 ± 0.052 |
| MNCA | 7.486 ± 0.044 | 7.421 ± 0.045 | 7.379 ± 0.047 |
| TabPFNv2 | - | - | - |
| TabDPT | 7.551 ± 0.042 | - | - |
| TabICL | - | - | - |
| Linear | 8.564 ± 0.047 | 8.340 ± 0.063 | 8.277 ± 0.055 |
| KNN | 8.563 ± 0.042 | 8.042 ± 0.048 | 7.909 ± 0.049 |
| AutoGluon | - | - | **7.362 ± 0.051** |

## GiveMeSomeCredit (AUC ↑)

| | Default | Tuned | Tuned + Ens. |
|---|---|---|---|
| RF | 0.846 ± 0.003 | 0.862 ± 0.003 | 0.863 ± 0.003 |
| ExtraTrees | 0.840 ± 0.003 | 0.857 ± 0.002 | 0.857 ± 0.003 |
| XGBoost | 0.865 ± 0.002 | 0.866 ± 0.002 | 0.866 ± 0.002 |
| LightGBM | 0.865 ± 0.002 | 0.866 ± 0.002 | 0.867 ± 0.002 |
| CatBoost | 0.866 ± 0.002 | 0.867 ± 0.002 | 0.867 ± 0.002 |
| EBM | 0.864 ± 0.002 | 0.865 ± 0.002 | 0.865 ± 0.002 |
| FastAIMLP | 0.829 ± 0.004 | 0.843 ± 0.005 | 0.847 ± 0.003 |
| TorchMLP | 0.863 ± 0.002 | 0.864 ± 0.002 | 0.865 ± 0.002 |
| RealMLP | 0.865 ± 0.002 | 0.866 ± 0.002 | 0.866 ± 0.002 |
| TabM | 0.866 ± 0.002 | **0.867 ± 0.002** | **0.867 ± 0.002** |
| MNCA | 0.866 ± 0.002 | 0.866 ± 0.002 | 0.867 ± 0.002 |
| TabPFNv2 | - | - | - |
| TabDPT | 0.842 ± 0.003 | - | - |
| TabICL | 0.866 ± 0.002 | - | - |
| Linear | 0.841 ± 0.002 | 0.841 ± 0.002 | 0.841 ± 0.002 |
| KNN | 0.747 ± 0.004 | 0.858 ± 0.002 | 0.859 ± 0.002 |
| AutoGluon | - | - | 0.867 ± 0.002 |

## HR_Analytics_Job_Change (AUC ↑)

| | Default | Tuned | Tuned + Ens. |
|---|---|---|---|
| RF | 0.789 ± 0.006 | 0.802 ± 0.006 | 0.802 ± 0.006 |
| ExtraTrees | 0.784 ± 0.007 | 0.800 ± 0.007 | 0.801 ± 0.007 |
| XGBoost | **0.805 ± 0.006** | 0.803 ± 0.005 | **0.805 ± 0.006** |
| LightGBM | 0.802 ± 0.007 | 0.803 ± 0.007 | 0.804 ± 0.006 |
| CatBoost | **0.804 ± 0.006** | **0.804 ± 0.006** | 0.804 ± 0.006 |
| EBM | 0.800 ± 0.006 | 0.800 ± 0.006 | 0.801 ± 0.006 |
| FastAIMLP | 0.801 ± 0.005 | 0.801 ± 0.007 | 0.803 ± 0.007 |
| TorchMLP | 0.801 ± 0.006 | 0.801 ± 0.006 | 0.803 ± 0.006 |
| RealMLP | 0.801 ± 0.007 | 0.801 ± 0.006 | 0.803 ± 0.006 |
| TabM | 0.801 ± 0.007 | 0.803 ± 0.006 | 0.803 ± 0.006 |
| MNCA | 0.801 ± 0.006 | 0.802 ± 0.007 | 0.803 ± 0.007 |
| TabPFNv2 | - | - | - |
| TabDPT | 0.801 ± 0.006 | - | - |
| TabICL | **0.805 ± 0.006** | - | - |
| Linear | 0.796 ± 0.006 | 0.798 ± 0.005 | 0.797 ± 0.005 |
| KNN | 0.745 ± 0.007 | 0.789 ± 0.007 | 0.798 ± 0.007 |
| AutoGluon | - | - | **0.805 ± 0.007** |

## Is-this-a-good-customer (AUC ↑)

| | Default | Tuned | Tuned + Ens. |
|---|---|---|---|
| RF | 0.721 ± 0.020 | 0.727 ± 0.018 | 0.729 ± 0.020 |
| ExtraTrees | 0.695 ± 0.021 | 0.713 ± 0.022 | 0.718 ± 0.024 |
| XGBoost | 0.723 ± 0.021 | 0.742 ± 0.023 | 0.744 ± 0.022 |
| LightGBM | 0.724 ± 0.020 | 0.741 ± 0.022 | **0.746 ± 0.020** |
| CatBoost | **0.748 ± 0.020** | 0.743 ± 0.019 | 0.744 ± 0.019 |
| EBM | **0.751 ± 0.019** | 0.745 ± 0.018 | 0.748 ± 0.018 |
| FastAIMLP | 0.711 ± 0.018 | **0.742 ± 0.025** | 0.745 ± 0.017 |
| TorchMLP | 0.728 ± 0.020 | 0.727 ± 0.023 | 0.733 ± 0.018 |
| RealMLP | 0.732 ± 0.023 | 0.731 ± 0.025 | 0.742 ± 0.020 |
| TabM | 0.744 ± 0.022 | 0.743 ± 0.019 | 0.744 ± 0.019 |
| MNCA | 0.738 ± 0.024 | 0.732 ± 0.020 | 0.705 ± 0.024 |
| TabPFNv2 | **0.746 ± 0.019** | 0.735 ± 0.022 | 0.743 ± 0.018 |
| TabDPT | 0.742 ± 0.016 | - | - |
| TabICL | 0.744 ± 0.019 | - | - |
| Linear | 0.738 ± 0.021 | 0.735 ± 0.020 | 0.738 ± 0.021 |
| KNN | 0.652 ± 0.037 | 0.741 ± 0.024 | 0.730 ± 0.023 |
| AutoGluon | - | - | 0.745 ± 0.019 |

## MIC (logloss ↓)

| | Default | Tuned | Tuned + Ens. |
|---|---|---|---|
| RF | 0.513 ± 0.031 | 0.485 ± 0.023 | 0.474 ± 0.019 |
| ExtraTrees | 0.521 ± 0.030 | 0.482 ± 0.020 | 0.470 ± 0.018 |
| XGBoost | 0.470 ± 0.020 | 0.440 ± 0.019 | 0.440 ± 0.019 |
| LightGBM | 0.503 ± 0.019 | 0.453 ± 0.020 | 0.453 ± 0.019 |
| CatBoost | 0.455 ± 0.020 | 0.453 ± 0.019 | 0.451 ± 0.018 |
| EBM | 0.475 ± 0.018 | 0.446 ± 0.016 | 0.445 ± 0.016 |
| FastAIMLP | 0.506 ± 0.024 | 0.462 ± 0.023 | 0.450 ± 0.020 |
| TorchMLP | 0.473 ± 0.019 | 0.465 ± 0.024 | 0.453 ± 0.017 |
| RealMLP | 0.492 ± 0.027 | 0.439 ± 0.021 | 0.434 ± 0.017 |
| TabM | 0.432 ± 0.017 | **0.431 ± 0.017** | **0.430 ± 0.016** |
| MNCA | 0.465 ± 0.019 | 0.455 ± 0.020 | 0.451 ± 0.018 |
| TabPFNv2 | 0.468 ± 0.043 | 0.440 ± 0.022 | **0.433 ± 0.022** |
| TabDPT | 0.481 ± 0.021 | - | - |
| TabICL | 0.465 ± 0.022 | - | - |
| Linear | 0.589 ± 0.035 | 0.469 ± 0.021 | 0.468 ± 0.021 |
| KNN | 1.477 ± 0.096 | 0.657 ± 0.025 | 0.583 ± 0.019 |
| AutoGluon | - | - | 0.445 ± 0.018 |

## Marketing_Campaign (AUC ↑)

| | Default | Tuned | Tuned + Ens. |
|---|---|---|---|
| RF | 0.883 ± 0.015 | 0.881 ± 0.016 | 0.882 ± 0.015 |
| ExtraTrees | 0.884 ± 0.015 | 0.886 ± 0.015 | 0.888 ± 0.015 |
| XGBoost | 0.897 ± 0.015 | 0.903 ± 0.016 | 0.904 ± 0.015 |
| LightGBM | 0.901 ± 0.014 | 0.911 ± 0.015 | 0.911 ± 0.014 |
| CatBoost | 0.907 ± 0.015 | 0.904 ± 0.014 | 0.903 ± 0.015 |
| EBM | 0.903 ± 0.015 | 0.905 ± 0.015 | 0.906 ± 0.015 |
| FastAIMLP | 0.890 ± 0.017 | 0.905 ± 0.015 | 0.909 ± 0.014 |
| TorchMLP | 0.898 ± 0.015 | 0.910 ± 0.014 | 0.915 ± 0.013 |
| RealMLP | 0.906 ± 0.015 | 0.907 ± 0.014 | 0.911 ± 0.014 |
| TabM | 0.901 ± 0.016 | 0.916 ± 0.013 | 0.916 ± 0.014 |
| MNCA | 0.909 ± 0.016 | 0.912 ± 0.016 | 0.909 ± 0.015 |
| TabPFNv2 | 0.915 ± 0.015 | 0.915 ± 0.013 | **0.919 ± 0.013** |
| TabDPT | 0.896 ± 0.016 | - | - |
| TabICL | 0.911 ± 0.013 | - | - |
| Linear | 0.906 ± 0.013 | 0.905 ± 0.013 | 0.905 ± 0.013 |
| KNN | 0.834 ± 0.018 | 0.854 ± 0.018 | 0.875 ± 0.014 |
| AutoGluon | - | - | 0.915 ± 0.013 |

## NATICUSdroid (AUC ↑)

| | Default | Tuned | Tuned + Ens. |
|---|---|---|---|
| RF | 0.977 ± 0.002 | 0.981 ± 0.003 | 0.981 ± 0.003 |
| ExtraTrees | 0.977 ± 0.002 | 0.982 ± 0.002 | 0.982 ± 0.002 |
| XGBoost | 0.985 ± 0.002 | 0.985 ± 0.002 | 0.985 ± 0.002 |
| LightGBM | 0.985 ± 0.002 | 0.986 ± 0.002 | 0.986 ± 0.002 |
| CatBoost | 0.986 ± 0.002 | 0.986 ± 0.002 | 0.986 ± 0.002 |
| EBM | 0.984 ± 0.002 | 0.985 ± 0.001 | 0.985 ± 0.001 |
| FastAIMLP | 0.985 ± 0.002 | 0.985 ± 0.001 | 0.986 ± 0.001 |
| TorchMLP | 0.985 ± 0.002 | 0.985 ± 0.001 | 0.986 ± 0.002 |
| RealMLP | 0.985 ± 0.002 | 0.986 ± 0.001 | 0.986 ± 0.001 |
| TabM | 0.986 ± 0.001 | 0.986 ± 0.001 | 0.986 ± 0.001 |
| MNCA | 0.983 ± 0.002 | 0.984 ± 0.002 | 0.985 ± 0.002 |
| TabPFNv2 | 0.983 ± 0.002 | 0.984 ± 0.003 | 0.985 ± 0.002 |
| TabDPT | 0.985 ± 0.002 | - | - |
| TabICL | **0.987 ± 0.001** | - | - |
| Linear | 0.981 ± 0.002 | 0.981 ± 0.002 | 0.981 ± 0.002 |
| KNN | 0.948 ± 0.004 | 0.978 ± 0.003 | 0.980 ± 0.003 |
| AutoGluon | - | - | 0.987 ± 0.002 |

## QSAR-TID-11 (rmse ↓)

| | Default | Tuned | Tuned + Ens. |
|---|---|---|---|
| RF | 0.806 ± 0.047 | 0.805 ± 0.048 | 0.796 ± 0.046 |
| ExtraTrees | 0.806 ± 0.046 | 0.802 ± 0.046 | 0.791 ± 0.046 |
| XGBoost | 0.786 ± 0.049 | 0.761 ± 0.047 | 0.760 ± 0.048 |
| LightGBM | 0.772 ± 0.050 | 0.758 ± 0.049 | 0.756 ± 0.048 |
| CatBoost | 0.774 ± 0.049 | 0.773 ± 0.048 | 0.771 ± 0.049 |
| EBM | 0.872 ± 0.039 | 0.859 ± 0.042 | 0.853 ± 0.042 |
| FastAIMLP | 0.776 ± 0.045 | 0.766 ± 0.049 | 0.761 ± 0.050 |
| TorchMLP | 0.774 ± 0.052 | 0.762 ± 0.049 | 0.748 ± 0.054 |
| RealMLP | 0.763 ± 0.047 | 0.764 ± 0.049 | 0.754 ± 0.050 |
| TabM | 0.761 ± 0.050 | 0.758 ± 0.050 | 0.755 ± 0.049 |
| MNCA | 0.771 ± 0.045 | 0.744 ± 0.044 | **0.734 ± 0.043** |
| TabPFNv2 | - | - | - |
| TabDPT | 0.773 ± 0.046 | - | - |
| TabICL | - | - | - |
| Linear | 1.020 ± 0.032 | 0.940 ± 0.040 | 0.938 ± 0.039 |
| KNN | 1.003 ± 0.043 | 0.888 ± 0.046 | 0.859 ± 0.047 |
| AutoGluon | - | - | 0.747 ± 0.049 |

## QSAR_fish_toxicity (rmse ↓)

| | Default | Tuned | Tuned + Ens. |
|---|---|---|---|
| RF | 0.907 ± 0.047 | 0.885 ± 0.050 | 0.884 ± 0.049 |
| ExtraTrees | 0.880 ± 0.052 | 0.873 ± 0.055 | 0.870 ± 0.052 |
| XGBoost | 0.905 ± 0.050 | 0.881 ± 0.043 | 0.879 ± 0.042 |
| LightGBM | 0.894 ± 0.043 | 0.889 ± 0.045 | 0.883 ± 0.044 |
| CatBoost | 0.877 ± 0.049 | 0.875 ± 0.045 | 0.874 ± 0.047 |
| EBM | 0.905 ± 0.050 | 0.904 ± 0.048 | 0.898 ± 0.047 |
| FastAIMLP | 0.908 ± 0.048 | 0.909 ± 0.046 | 0.897 ± 0.048 |
| TorchMLP | 0.906 ± 0.055 | 0.897 ± 0.055 | 0.890 ± 0.055 |
| RealMLP | 0.878 ± 0.054 | 0.884 ± 0.058 | 0.865 ± 0.052 |
| TabM | 0.910 ± 0.047 | 0.898 ± 0.051 | 0.887 ± 0.048 |
| MNCA | 0.882 ± 0.053 | 0.882 ± 0.053 | 0.873 ± 0.053 |
| TabPFNv2 | 0.868 ± 0.047 | 0.873 ± 0.051 | **0.860 ± 0.049** |
| TabDPT | **0.859 ± 0.049** | - | - |
| TabICL | - | - | - |
| Linear | 0.950 ± 0.056 | 0.949 ± 0.056 | 0.950 ± 0.056 |
| KNN | 0.896 ± 0.057 | 0.890 ± 0.052 | 0.880 ± 0.054 |
| AutoGluon | - | - | 0.880 ± 0.054 |

## SDSS17 (logloss ↓)

| | Default | Tuned | Tuned + Ens. |
|---|---|---|---|
| RF | 0.085 ± 0.002 | 0.073 ± 0.002 | 0.072 ± 0.002 |
| ExtraTrees | 0.131 ± 0.002 | 0.080 ± 0.002 | 0.078 ± 0.002 |
| XGBoost | 0.074 ± 0.002 | 0.074 ± 0.002 | 0.074 ± 0.002 |
| LightGBM | 0.087 ± 0.002 | 0.073 ± 0.003 | 0.073 ± 0.002 |
| CatBoost | 0.075 ± 0.002 | 0.074 ± 0.003 | 0.074 ± 0.003 |
| EBM | 0.087 ± 0.002 | 0.081 ± 0.002 | 0.081 ± 0.002 |
| FastAIMLP | 0.134 ± 0.004 | 0.111 ± 0.004 | 0.112 ± 0.003 |
| TorchMLP | 0.094 ± 0.003 | 0.081 ± 0.002 | 0.080 ± 0.002 |
| RealMLP | 0.103 ± 0.002 | 0.089 ± 0.002 | 0.087 ± 0.002 |
| TabM | 0.097 ± 0.002 | 0.083 ± 0.002 | 0.083 ± 0.002 |
| MNCA | 0.082 ± 0.002 | 0.076 ± 0.002 | 0.075 ± 0.002 |
| TabPFNv2 | - | - | - |
| TabDPT | 0.088 ± 0.001 | - | - |
| TabICL | 0.076 ± 0.002 | - | - |
| Linear | 0.146 ± 0.003 | 0.145 ± 0.002 | 0.135 ± 0.002 |
| KNN | 0.380 ± 0.030 | 0.224 ± 0.003 | 0.153 ± 0.003 |
| AutoGluon | - | - | **0.067 ± 0.002** |

## airfoil_self_noise (rmse ↓)

| | Default | Tuned | Tuned + Ens. |
|---|---|---|---|
| RF | 1.898 ± 0.095 | 1.891 ± 0.101 | 1.851 ± 0.096 |
| ExtraTrees | 1.822 ± 0.094 | 1.676 ± 0.106 | 1.683 ± 0.105 |
| XGBoost | 1.549 ± 0.104 | 1.439 ± 0.104 | 1.443 ± 0.104 |
| LightGBM | 1.554 ± 0.093 | 1.480 ± 0.108 | 1.451 ± 0.108 |
| CatBoost | 1.583 ± 0.096 | 1.327 ± 0.101 | 1.330 ± 0.105 |
| EBM | 2.010 ± 0.117 | 1.955 ± 0.104 | 1.935 ± 0.113 |
| FastAIMLP | 2.327 ± 0.141 | 1.624 ± 0.100 | 1.646 ± 0.105 |
| TorchMLP | 1.493 ± 0.113 | 1.372 ± 0.105 | 1.374 ± 0.098 |
| RealMLP | 1.179 ± 0.085 | 1.146 ± 0.078 | 1.109 ± 0.081 |
| TabM | 1.249 ± 0.099 | 1.149 ± 0.092 | 1.140 ± 0.088 |
| MNCA | 1.539 ± 0.095 | 1.523 ± 0.137 | 1.452 ± 0.099 |
| TabPFNv2 | 1.119 ± 0.088 | 1.112 ± 0.102 | **1.074 ± 0.094** |
| TabDPT | 1.203 ± 0.084 | - | - |
| TabICL | - | - | - |
| Linear | 5.344 ± 0.199 | 4.752 ± 0.141 | 4.732 ± 0.144 |
| KNN | 3.369 ± 0.113 | 2.599 ± 0.154 | 2.221 ± 0.122 |
| AutoGluon | - | - | 1.269 ± 0.090 |

## anneal (logloss ↓)

| | Default | Tuned | Tuned + Ens. |
|---|---|---|---|
| RF | 0.046 ± 0.010 | 0.028 ± 0.025 | 0.023 ± 0.013 |
| ExtraTrees | 0.064 ± 0.012 | 0.028 ± 0.022 | 0.026 ± 0.022 |
| XGBoost | 0.039 ± 0.025 | 0.031 ± 0.025 | 0.031 ± 0.025 |
| LightGBM | 0.055 ± 0.026 | 0.033 ± 0.019 | 0.034 ± 0.019 |
| CatBoost | 0.040 ± 0.022 | 0.022 ± 0.021 | 0.021 ± 0.021 |
| EBM | 0.043 ± 0.025 | 0.036 ± 0.033 | 0.034 ± 0.032 |
| FastAIMLP | 0.085 ± 0.029 | 0.056 ± 0.022 | 0.054 ± 0.021 |
| TorchMLP | 0.040 ± 0.034 | 0.052 ± 0.044 | 0.040 ± 0.038 |
| RealMLP | 0.039 ± 0.031 | 0.029 ± 0.032 | 0.024 ± 0.026 |
| TabM | 0.036 ± 0.026 | 0.029 ± 0.027 | 0.028 ± 0.025 |
| MNCA | 0.043 ± 0.031 | 0.032 ± 0.038 | 0.033 ± 0.039 |
| TabPFNv2 | **0.016 ± 0.014** | 0.023 ± 0.019 | 0.019 ± 0.014 |
| TabDPT | 0.058 ± 0.022 | - | - |
| TabICL | 0.028 ± 0.014 | - | - |
| Linear | 0.090 ± 0.028 | 0.046 ± 0.032 | 0.037 ± 0.024 |
| KNN | 0.151 ± 0.043 | 0.080 ± 0.046 | 0.062 ± 0.040 |
| AutoGluon | - | - | 0.037 ± 0.059 |

## bank-marketing (AUC ↑)

| | Default | Tuned | Tuned + Ens. |
|---|---|---|---|
| RF | 0.726 ± 0.006 | 0.761 ± 0.005 | 0.761 ± 0.005 |
| ExtraTrees | 0.721 ± 0.004 | 0.758 ± 0.005 | 0.758 ± 0.005 |
| XGBoost | 0.763 ± 0.005 | **0.765 ± 0.006** | **0.765 ± 0.005** |
| LightGBM | 0.763 ± 0.005 | **0.765 ± 0.005** | **0.766 ± 0.005** |
| CatBoost | **0.766 ± 0.005** | **0.766 ± 0.005** | **0.765 ± 0.005** |
| EBM | 0.762 ± 0.005 | 0.763 ± 0.005 | 0.763 ± 0.005 |
| FastAIMLP | 0.759 ± 0.005 | 0.760 ± 0.006 | 0.761 ± 0.005 |
| TorchMLP | 0.757 ± 0.006 | 0.758 ± 0.006 | 0.759 ± 0.006 |
| RealMLP | 0.761 ± 0.005 | 0.763 ± 0.005 | 0.765 ± 0.005 |
| TabM | 0.764 ± 0.006 | 0.764 ± 0.005 | **0.765 ± 0.005** |
| MNCA | 0.762 ± 0.005 | 0.764 ± 0.006 | 0.764 ± 0.005 |
| TabPFNv2 | - | - | - |
| TabDPT | 0.761 ± 0.005 | - | - |
| TabICL | 0.764 ± 0.005 | - | - |
| Linear | 0.748 ± 0.004 | 0.748 ± 0.004 | 0.748 ± 0.004 |
| KNN | 0.712 ± 0.005 | 0.749 ± 0.006 | 0.756 ± 0.005 |
| AutoGluon | - | - | **0.765 ± 0.006** |

## blood-transfusion-service-center (AUC ↑)

| | Default | Tuned | Tuned + Ens. |
|---|---|---|---|
| RF | 0.682 ± 0.026 | 0.714 ± 0.029 | 0.713 ± 0.029 |
| ExtraTrees | 0.689 ± 0.025 | 0.728 ± 0.033 | 0.727 ± 0.031 |
| XGBoost | 0.708 ± 0.030 | 0.733 ± 0.031 | 0.731 ± 0.031 |
| LightGBM | 0.726 ± 0.033 | 0.743 ± 0.033 | 0.743 ± 0.031 |
| CatBoost | 0.738 ± 0.029 | 0.737 ± 0.031 | 0.736 ± 0.031 |
| EBM | 0.742 ± 0.032 | 0.743 ± 0.033 | 0.743 ± 0.033 |
| FastAIMLP | 0.743 ± 0.030 | **0.754 ± 0.030** | **0.756 ± 0.030** |
| TorchMLP | 0.749 ± 0.032 | 0.747 ± 0.030 | 0.748 ± 0.030 |
| RealMLP | 0.750 ± 0.031 | 0.737 ± 0.029 | 0.742 ± 0.027 |
| TabM | 0.741 ± 0.029 | 0.737 ± 0.031 | 0.741 ± 0.029 |
| MNCA | **0.756 ± 0.030** | 0.737 ± 0.032 | 0.715 ± 0.030 |
| TabPFNv2 | **0.755 ± 0.029** | 0.748 ± 0.034 | 0.746 ± 0.030 |
| TabDPT | 0.751 ± 0.030 | - | - |
| TabICL | 0.737 ± 0.031 | - | - |
| Linear | 0.731 ± 0.032 | 0.749 ± 0.029 | **0.754 ± 0.026** |
| KNN | 0.688 ± 0.023 | 0.727 ± 0.033 | 0.736 ± 0.034 |
| AutoGluon | - | - | 0.748 ± 0.032 |

## churn (AUC ↑)

| | Default | Tuned | Tuned + Ens. |
|---|---|---|---|
| RF | 0.915 ± 0.009 | 0.913 ± 0.012 | 0.913 ± 0.011 |
| ExtraTrees | 0.917 ± 0.011 | 0.919 ± 0.011 | 0.921 ± 0.011 |
| XGBoost | 0.923 ± 0.009 | 0.921 ± 0.011 | 0.920 ± 0.011 |
| LightGBM | 0.916 ± 0.011 | 0.920 ± 0.011 | 0.920 ± 0.011 |
| CatBoost | 0.924 ± 0.011 | 0.920 ± 0.012 | 0.922 ± 0.011 |
| EBM | 0.922 ± 0.014 | 0.924 ± 0.011 | 0.924 ± 0.011 |
| FastAIMLP | 0.918 ± 0.011 | 0.921 ± 0.009 | 0.921 ± 0.010 |
| TorchMLP | 0.888 ± 0.009 | 0.918 ± 0.010 | 0.918 ± 0.011 |
| RealMLP | 0.920 ± 0.010 | 0.924 ± 0.012 | 0.927 ± 0.011 |
| TabM | 0.925 ± 0.011 | 0.923 ± 0.010 | 0.923 ± 0.010 |
| MNCA | 0.903 ± 0.013 | **0.932 ± 0.013** | **0.931 ± 0.013** |
| TabPFNv2 | **0.928 ± 0.011** | **0.925 ± 0.008** | **0.924 ± 0.010** |
| TabDPT | 0.923 ± 0.009 | - | - |
| TabICL | 0.924 ± 0.011 | - | - |
| Linear | 0.777 ± 0.018 | 0.825 ± 0.008 | 0.825 ± 0.008 |
| KNN | 0.866 ± 0.004 | 0.892 ± 0.010 | 0.899 ± 0.010 |
| AutoGluon | - | - | 0.922 ± 0.011 |

## coil2000_insurance_policies (AUC ↑)

| | Default | Tuned | Tuned + Ens. |
|---|---|---|---|
| RF | 0.697 ± 0.014 | 0.741 ± 0.017 | 0.742 ± 0.016 |
| ExtraTrees | 0.696 ± 0.016 | 0.744 ± 0.016 | 0.748 ± 0.017 |
| XGBoost | 0.757 ± 0.015 | 0.757 ± 0.016 | 0.758 ± 0.014 |
| LightGBM | 0.752 ± 0.014 | 0.759 ± 0.015 | 0.761 ± 0.015 |
| CatBoost | 0.757 ± 0.014 | 0.758 ± 0.013 | 0.759 ± 0.012 |
| EBM | 0.754 ± 0.014 | 0.757 ± 0.013 | 0.761 ± 0.013 |
| FastAIMLP | 0.719 ± 0.010 | 0.749 ± 0.013 | 0.747 ± 0.013 |
| TorchMLP | 0.740 ± 0.011 | 0.747 ± 0.015 | 0.752 ± 0.014 |
| RealMLP | 0.742 ± 0.012 | 0.755 ± 0.011 | 0.763 ± 0.013 |
| TabM | 0.761 ± 0.012 | 0.763 ± 0.012 | 0.766 ± 0.012 |
| MNCA | 0.753 ± 0.013 | 0.757 ± 0.015 | **0.767 ± 0.012** |
| TabPFNv2 | 0.753 ± 0.015 | **0.773 ± 0.015** | **0.773 ± 0.014** |
| TabDPT | 0.725 ± 0.012 | - | - |
| TabICL | 0.756 ± 0.012 | - | - |
| Linear | 0.737 ± 0.012 | 0.739 ± 0.011 | 0.740 ± 0.012 |
| KNN | 0.657 ± 0.020 | 0.718 ± 0.015 | 0.732 ± 0.014 |
| AutoGluon | - | - | 0.759 ± 0.016 |

## concrete_compressive_strength (rmse ↓)

|  | Default | Tuned | Tuned + Ens. |
|---|---|---|---|
| RF | 5.261 ± 0.336 | 5.189 ± 0.366 | 5.106 ± 0.352 |
| ExtraTrees | 5.139 ± 0.341 | 5.073 ± 0.333 | 5.048 ± 0.348 |
| XGBoost | 4.755 ± 0.387 | 4.236 ± 0.373 | 4.222 ± 0.384 |
| LightGBM | 4.484 ± 0.388 | 4.235 ± 0.395 | 4.212 ± 0.396 |
| CatBoost | 4.214 ± 0.413 | 4.231 ± 0.415 | 4.209 ± 0.411 |
| EBM | 4.442 ± 0.295 | 4.429 ± 0.331 | 4.371 ± 0.308 |
| FastAIMLP | 6.369 ± 0.379 | 5.187 ± 0.355 | 5.272 ± 0.360 |
| TorchMLP | 4.817 ± 0.354 | 4.715 ± 0.350 | 4.654 ± 0.334 |
| RealMLP | 4.688 ± 0.364 | 4.344 ± 0.289 | **4.133 ± 0.329** |
| TabM | 4.271 ± 0.385 | 4.269 ± 0.496 | **4.146 ± 0.409** |
| MNCA | 4.940 ± 0.309 | 4.749 ± 0.424 | 4.487 ± 0.388 |
| TabPFNv2 | 4.259 ± 0.379 | 4.171 ± 0.439 | **4.118 ± 0.409** |
| TabDPT | 4.267 ± 0.422 | - | - |
| TabICL | - | - | - |
| Linear | 8.228 ± 0.359 | 8.160 ± 0.362 | 8.153 ± 0.354 |
| KNN | 8.972 ± 0.464 | 6.762 ± 0.565 | 6.548 ± 0.555 |
| AutoGluon | - | - | **4.165 ± 0.389** |

## credit-g (AUC ↑)

|  | Default | Tuned | Tuned + Ens. |
|---|---|---|---|
| RF | 0.783 ± 0.017 | 0.781 ± 0.019 | 0.782 ± 0.019 |
| ExtraTrees | 0.779 ± 0.019 | 0.781 ± 0.018 | 0.782 ± 0.018 |
| XGBoost | 0.783 ± 0.021 | 0.792 ± 0.021 | 0.793 ± 0.021 |
| LightGBM | 0.771 ± 0.019 | 0.792 ± 0.020 | **0.796 ± 0.020** |
| CatBoost | 0.789 ± 0.017 | 0.795 ± 0.020 | 0.795 ± 0.017 |
| EBM | 0.790 ± 0.021 | 0.782 ± 0.025 | 0.787 ± 0.023 |
| FastAIMLP | 0.783 ± 0.029 | 0.784 ± 0.025 | **0.793 ± 0.023** |
| TorchMLP | 0.772 ± 0.017 | 0.782 ± 0.020 | 0.788 ± 0.020 |
| RealMLP | 0.785 ± 0.022 | 0.784 ± 0.023 | 0.791 ± 0.020 |
| TabM | 0.793 ± 0.022 | 0.790 ± 0.020 | **0.795 ± 0.020** |
| MNCA | 0.783 ± 0.020 | 0.783 ± 0.020 | 0.775 ± 0.021 |
| TabPFNv2 | 0.776 ± 0.019 | 0.773 ± 0.020 | 0.792 ± 0.020 |
| TabDPT | 0.780 ± 0.019 | - | - |
| TabICL | 0.790 ± 0.017 | - | - |
| Linear | 0.781 ± 0.021 | 0.787 ± 0.021 | 0.787 ± 0.021 |
| KNN | 0.756 ± 0.031 | 0.778 ± 0.022 | 0.784 ± 0.022 |
| AutoGluon | - | - | **0.794 ± 0.020** |

## credit_card_clients_default (AUC ↑)

|  | Default | Tuned | Tuned + Ens. |
|---|---|---|---|
| RF | 0.765 ± 0.005 | 0.780 ± 0.004 | 0.780 ± 0.004 |
| ExtraTrees | 0.766 ± 0.004 | 0.781 ± 0.004 | 0.782 ± 0.004 |
| XGBoost | 0.783 ± 0.004 | 0.785 ± 0.004 | 0.785 ± 0.004 |
| LightGBM | 0.784 ± 0.004 | 0.785 ± 0.004 | 0.785 ± 0.004 |
| CatBoost | 0.784 ± 0.004 | 0.785 ± 0.004 | 0.785 ± 0.004 |
| EBM | 0.783 ± 0.004 | 0.783 ± 0.004 | 0.784 ± 0.004 |
| FastAIMLP | 0.781 ± 0.005 | 0.783 ± 0.005 | 0.783 ± 0.005 |
| TorchMLP | 0.779 ± 0.003 | 0.783 ± 0.003 | 0.785 ± 0.003 |
| RealMLP | 0.785 ± 0.005 | 0.785 ± 0.005 | 0.786 ± 0.004 |
| TabM | 0.784 ± 0.004 | **0.788 ± 0.004** | **0.788 ± 0.004** |
| MNCA | 0.781 ± 0.004 | 0.787 ± 0.003 | **0.787 ± 0.004** |
| TabPFNv2 | - | - | - |
| TabDPT | 0.780 ± 0.004 | - | - |
| TabICL | **0.788 ± 0.004** | - | - |
| Linear | 0.745 ± 0.004 | 0.745 ± 0.004 | 0.745 ± 0.004 |
| KNN | 0.736 ± 0.004 | 0.758 ± 0.004 | 0.769 ± 0.004 |
| AutoGluon | - | - | **0.787 ± 0.004** |

## customer_satisfaction_in_airline (AUC ↑)

|  | Default | Tuned | Tuned + Ens. |
|---|---|---|---|
| RF | 0.993 ± 0.000 | 0.993 ± 0.000 | 0.993 ± 0.000 |
| ExtraTrees | 0.992 ± 0.000 | 0.994 ± 0.000 | 0.994 ± 0.000 |
| XGBoost | 0.994 ± 0.000 | 0.994 ± 0.000 | 0.994 ± 0.000 |
| LightGBM | 0.994 ± 0.000 | 0.995 ± 0.000 | 0.995 ± 0.000 |
| CatBoost | 0.995 ± 0.000 | 0.995 ± 0.000 | 0.995 ± 0.000 |
| EBM | 0.985 ± 0.000 | 0.986 ± 0.001 | 0.986 ± 0.001 |
| FastAIMLP | 0.995 ± 0.000 | 0.995 ± 0.000 | 0.995 ± 0.000 |
| TorchMLP | 0.993 ± 0.000 | 0.995 ± 0.000 | 0.995 ± 0.000 |
| RealMLP | 0.995 ± 0.000 | 0.995 ± 0.000 | 0.995 ± 0.000 |
| TabM | 0.995 ± 0.000 | 0.995 ± 0.000 | 0.995 ± 0.000 |
| MNCA | 0.993 ± 0.000 | 0.995 ± 0.000 | 0.995 ± 0.000 |
| TabPFNv2 | - | - | - |
| TabDPT | 0.994 ± 0.000 | - | - |
| TabICL | 0.995 ± 0.000 | - | - |
| Linear | 0.964 ± 0.001 | 0.964 ± 0.001 | 0.965 ± 0.001 |
| KNN | 0.985 ± 0.000 | 0.985 ± 0.000 | 0.988 ± 0.000 |
| AutoGluon | - | - | **0.996 ± 0.000** |

## diabetes (AUC ↑)

|  | Default | Tuned | Tuned + Ens. |
|---|---|---|---|
| RF | 0.825 ± 0.023 | 0.830 ± 0.025 | 0.830 ± 0.024 |
| ExtraTrees | 0.826 ± 0.022 | 0.837 ± 0.021 | 0.837 ± 0.021 |
| XGBoost | 0.824 ± 0.024 | 0.830 ± 0.024 | 0.832 ± 0.024 |
| LightGBM | 0.829 ± 0.025 | 0.838 ± 0.026 | 0.837 ± 0.025 |
| CatBoost | 0.833 ± 0.025 | 0.834 ± 0.025 | 0.835 ± 0.024 |
| EBM | 0.840 ± 0.025 | 0.839 ± 0.023 | 0.840 ± 0.023 |
| FastAIMLP | 0.826 ± 0.024 | 0.832 ± 0.023 | 0.835 ± 0.023 |
| TorchMLP | 0.821 ± 0.024 | 0.825 ± 0.026 | 0.827 ± 0.025 |
| RealMLP | 0.833 ± 0.023 | 0.832 ± 0.022 | 0.836 ± 0.024 |
| TabM | 0.832 ± 0.024 | 0.830 ± 0.024 | 0.834 ± 0.024 |
| MNCA | 0.839 ± 0.024 | 0.834 ± 0.024 | 0.813 ± 0.022 |
| TabPFNv2 | **0.844 ± 0.023** | **0.842 ± 0.024** | 0.839 ± 0.024 |
| TabDPT | 0.840 ± 0.023 | - | - |
| TabICL | 0.837 ± 0.023 | - | - |
| Linear | 0.832 ± 0.024 | 0.830 ± 0.023 | 0.831 ± 0.023 |
| KNN | 0.811 ± 0.023 | 0.823 ± 0.025 | 0.823 ± 0.025 |
| AutoGluon | - | - | 0.835 ± 0.023 |

## diamonds (rmse ↓)

|  | Default | Tuned | Tuned + Ens. |
|---|---|---|---|
| RF | 549.9 ± 8.3 | 549.9 ± 8.3 | 547.2 ± 9.3 |
| ExtraTrees | 536.6 ± 8.7 | 536.3 ± 9.1 | 534.6 ± 8.9 |
| XGBoost | 539.0 ± 10.0 | 530.1 ± 10.0 | 528.2 ± 10.9 |
| LightGBM | 532.1 ± 9.1 | 524.9 ± 9.7 | 519.0 ± 9.4 |
| CatBoost | 520.7 ± 12.0 | 520.7 ± 12.0 | 520.8 ± 11.8 |
| EBM | 618.2 ± 14.5 | 613.7 ± 11.9 | 612.4 ± 13.9 |
| FastAIMLP | 563.1 ± 15.0 | 559.4 ± 9.4 | 550.0 ± 8.9 |
| TorchMLP | 627.3 ± 25.8 | 550.5 ± 16.6 | 542.6 ± 15.2 |
| RealMLP | 529.6 ± 8.1 | 521.5 ± 7.6 | 513.7 ± 7.3 |
| TabM | 522.5 ± 8.9 | 521.5 ± 8.1 | 518.9 ± 8.8 |
| MNCA | 524.7 ± 8.7 | 521.1 ± 7.5 | **510.3 ± 8.1** |
| TabPFNv2 | - | - | - |
| TabDPT | 535.4 ± 13.1 | - | - |
| TabICL | - | - | - |
| Linear | 1652.8 ± 179.7 | 1139.6 ± 27.1 | 1140.3 ± 28.0 |
| KNN | 957.3 ± 22.1 | 706.9 ± 15.5 | 668.3 ± 14.2 |
| AutoGluon | - | - | **510.5 ± 9.5** |

## hazelnut-spread-contaminant-detection (AUC ↑)

|            | Default             | Tuned               | Tuned + Ens.        |
|------------|---------------------|---------------------|---------------------|
| RF         | 0.958 ± 0.006       | 0.959 ± 0.006       | 0.960 ± 0.006       |
| ExtraTrees | 0.955 ± 0.006       | 0.964 ± 0.005       | 0.964 ± 0.005       |
| XGBoost    | 0.973 ± 0.005       | 0.975 ± 0.004       | 0.975 ± 0.004       |
| LightGBM   | 0.973 ± 0.005       | 0.978 ± 0.004       | 0.978 ± 0.004       |
| CatBoost   | 0.974 ± 0.004       | 0.975 ± 0.004       | 0.974 ± 0.004       |
| EBM        | 0.971 ± 0.005       | 0.975 ± 0.004       | 0.976 ± 0.004       |
| FastAIMLP  | 0.983 ± 0.003       | 0.986 ± 0.003       | 0.986 ± 0.003       |
| TorchMLP   | 0.983 ± 0.003       | 0.987 ± 0.003       | 0.987 ± 0.003       |
| RealMLP    | 0.984 ± 0.003       | 0.986 ± 0.003       | 0.986 ± 0.003       |
| TabM       | 0.967 ± 0.005       | 0.984 ± 0.003       | 0.985 ± 0.003       |
| MNCA       | 0.986 ± 0.003       | 0.988 ± 0.003       | 0.988 ± 0.003       |
| TabPFNv2   | 0.988 ± 0.003       | 0.989 ± 0.003       | 0.989 ± 0.003       |
| TabDPT     | **0.992 ± 0.002**   | -                   | -                   |
| TabICL     | **0.992 ± 0.002**   | -                   | -                   |
| Linear     | 0.948 ± 0.006       | 0.952 ± 0.006       | 0.953 ± 0.006       |
| KNN        | 0.914 ± 0.009       | 0.934 ± 0.009       | 0.943 ± 0.008       |
| AutoGluon  | -                   | -                   | 0.987 ± 0.003       |

## healthcare_insurance_expenses (rmse ↓)

|            | Default             | Tuned               | Tuned + Ens.        |
|------------|---------------------|---------------------|---------------------|
| RF         | 4888.5 ± 289.2      | 4641.1 ± 303.6      | 4629.6 ± 302.9      |
| ExtraTrees | 4844.9 ± 274.8      | 4607.0 ± 324.0      | 4609.9 ± 323.0      |
| XGBoost    | 4672.0 ± 306.3      | 4523.3 ± 319.7      | 4519.6 ± 320.2      |
| LightGBM   | 4610.4 ± 313.8      | 4525.1 ± 329.2      | 4511.9 ± 325.5      |
| CatBoost   | 4535.4 ± 328.8      | 4518.2 ± 321.8      | 4519.2 ± 321.1      |
| EBM        | 4549.2 ± 318.8      | **4499.7 ± 333.4**  | **4499.3 ± 325.7**  |
| FastAIMLP  | 4720.2 ± 313.1      | 4633.9 ± 317.7      | 4624.2 ± 311.0      |
| TorchMLP   | 4661.1 ± 342.8      | 4526.1 ± 328.6      | 4534.0 ± 333.3      |
| RealMLP    | 4579.2 ± 313.2      | 4570.6 ± 323.5      | 4534.8 ± 323.1      |
| TabM       | 4513.7 ± 323.4      | 4530.0 ± 323.1      | 4510.9 ± 325.7      |
| MNCA       | 4605.4 ± 332.0      | 4614.4 ± 330.3      | 4589.2 ± 331.3      |
| TabPFNv2   | 4694.7 ± 302.6      | 4650.3 ± 336.9      | 4567.7 ± 318.7      |
| TabDPT     | **4508.4 ± 295.4**  | -                   | -                   |
| TabICL     | -                   | -                   | -                   |
| Linear     | 6083.4 ± 275.9      | 6085.5 ± 275.8      | 6084.5 ± 276.6      |
| KNN        | 5797.5 ± 392.9      | 5058.7 ± 315.5      | 5024.4 ± 319.8      |
| AutoGluon  | -                   | -                   | **4490.4 ± 331.9**  |

## heloc (AUC ↑)

|            | Default             | Tuned               | Tuned + Ens.        |
|------------|---------------------|---------------------|---------------------|
| RF         | 0.791 ± 0.005       | 0.792 ± 0.006       | 0.793 ± 0.005       |
| ExtraTrees | 0.790 ± 0.005       | 0.793 ± 0.005       | 0.793 ± 0.005       |
| XGBoost    | 0.794 ± 0.005       | 0.797 ± 0.005       | 0.797 ± 0.005       |
| LightGBM   | 0.794 ± 0.005       | 0.799 ± 0.005       | 0.799 ± 0.005       |
| CatBoost   | 0.798 ± 0.004       | 0.798 ± 0.005       | 0.798 ± 0.004       |
| EBM        | 0.799 ± 0.005       | 0.799 ± 0.005       | 0.799 ± 0.005       |
| FastAIMLP  | 0.791 ± 0.004       | 0.794 ± 0.005       | 0.795 ± 0.005       |
| TorchMLP   | 0.791 ± 0.004       | 0.795 ± 0.004       | 0.796 ± 0.004       |
| RealMLP    | 0.798 ± 0.004       | 0.798 ± 0.004       | 0.800 ± 0.004       |
| TabM       | 0.797 ± 0.004       | 0.799 ± 0.004       | 0.799 ± 0.004       |
| MNCA       | 0.799 ± 0.004       | **0.801 ± 0.005**   | 0.799 ± 0.005       |
| TabPFNv2   | **0.801 ± 0.003**   | **0.801 ± 0.004**   | **0.801 ± 0.003**   |
| TabDPT     | 0.794 ± 0.004       | -                   | -                   |
| TabICL     | 0.800 ± 0.004       | -                   | -                   |
| Linear     | 0.786 ± 0.005       | 0.786 ± 0.005       | 0.786 ± 0.005       |
| KNN        | 0.743 ± 0.004       | 0.788 ± 0.005       | 0.788 ± 0.004       |
| AutoGluon  | -                   | -                   | 0.798 ± 0.005       |

## hiva_agnostic (logloss ↓)

|            | Default             | Tuned               | Tuned + Ens.        |
|------------|---------------------|---------------------|---------------------|
| RF         | 0.263 ± 0.025       | **0.174 ± 0.001**   | **0.174 ± 0.001**   |
| ExtraTrees | 0.268 ± 0.027       | **0.174 ± 0.000**   | **0.174 ± 0.000**   |
| XGBoost    | 0.182 ± 0.002       | 0.179 ± 0.002       | 0.179 ± 0.002       |
| LightGBM   | 0.175 ± 0.001       | 0.175 ± 0.001       | 0.175 ± 0.001       |
| CatBoost   | 0.176 ± 0.001       | 0.177 ± 0.002       | 0.177 ± 0.002       |
| EBM        | **0.174 ± 0.001**   | 0.176 ± 0.001       | 0.175 ± 0.001       |
| FastAIMLP  | 0.213 ± 0.010       | 0.183 ± 0.008       | 0.183 ± 0.004       |
| TorchMLP   | 0.183 ± 0.005       | 0.176 ± 0.001       | 0.178 ± 0.003       |
| RealMLP    | 0.196 ± 0.002       | 0.176 ± 0.002       | 0.179 ± 0.002       |
| TabM       | 0.177 ± 0.001       | 0.175 ± 0.001       | 0.175 ± 0.001       |
| MNCA       | 0.224 ± 0.012       | 0.176 ± 0.002       | 0.179 ± 0.002       |
| TabPFNv2   | -                   | -                   | -                   |
| TabDPT     | 0.181 ± 0.004       | -                   | -                   |
| TabICL     | -                   | -                   | -                   |
| Linear     | 0.448 ± 0.024       | 0.335 ± 0.019       | 0.335 ± 0.019       |
| KNN        | 0.468 ± 0.017       | 0.176 ± 0.003       | 0.176 ± 0.003       |
| AutoGluon  | -                   | -                   | 0.193 ± 0.027       |

## houses (rmse ↓)

|            | Default             | Tuned               | Tuned + Ens.        |
|------------|---------------------|---------------------|---------------------|
| RF         | 0.231 ± 0.002       | 0.231 ± 0.002       | 0.230 ± 0.002       |
| ExtraTrees | 0.243 ± 0.002       | 0.238 ± 0.002       | 0.238 ± 0.002       |
| XGBoost    | 0.215 ± 0.003       | 0.215 ± 0.002       | 0.215 ± 0.002       |
| LightGBM   | 0.217 ± 0.002       | 0.212 ± 0.002       | 0.211 ± 0.002       |
| CatBoost   | 0.211 ± 0.002       | 0.211 ± 0.002       | 0.211 ± 0.002       |
| EBM        | 0.231 ± 0.003       | 0.229 ± 0.003       | 0.228 ± 0.003       |
| FastAIMLP  | 0.244 ± 0.001       | 0.236 ± 0.002       | 0.235 ± 0.001       |
| TorchMLP   | 0.233 ± 0.003       | 0.228 ± 0.003       | 0.226 ± 0.002       |
| RealMLP    | 0.223 ± 0.002       | 0.211 ± 0.003       | 0.203 ± 0.003       |
| TabM       | 0.212 ± 0.002       | 0.208 ± 0.002       | 0.205 ± 0.002       |
| MNCA       | 0.204 ± 0.003       | 0.203 ± 0.003       | **0.199 ± 0.002**   |
| TabPFNv2   | -                   | -                   | -                   |
| TabDPT     | 0.209 ± 0.003       | -                   | -                   |
| TabICL     | -                   | -                   | -                   |
| Linear     | 0.325 ± 0.003       | 0.325 ± 0.003       | 0.323 ± 0.003       |
| KNN        | 0.290 ± 0.002       | 0.282 ± 0.002       | 0.273 ± 0.002       |
| AutoGluon  | -                   | -                   | 0.204 ± 0.002       |

## in_vehicle_coupon_recommendation (AUC ↑)

|            | Default             | Tuned               | Tuned + Ens.        |
|------------|---------------------|---------------------|---------------------|
| RF         | 0.812 ± 0.007       | 0.817 ± 0.008       | 0.822 ± 0.008       |
| ExtraTrees | 0.798 ± 0.007       | 0.804 ± 0.008       | 0.811 ± 0.008       |
| XGBoost    | 0.832 ± 0.004       | 0.842 ± 0.005       | 0.843 ± 0.005       |
| LightGBM   | 0.836 ± 0.008       | 0.844 ± 0.005       | 0.845 ± 0.005       |
| CatBoost   | 0.840 ± 0.006       | 0.843 ± 0.005       | 0.844 ± 0.006       |
| EBM        | 0.802 ± 0.006       | 0.807 ± 0.006       | 0.807 ± 0.006       |
| FastAIMLP  | 0.810 ± 0.007       | 0.823 ± 0.007       | 0.826 ± 0.006       |
| TorchMLP   | 0.825 ± 0.004       | 0.833 ± 0.008       | 0.841 ± 0.006       |
| RealMLP    | 0.837 ± 0.006       | 0.839 ± 0.006       | 0.849 ± 0.006       |
| TabM       | 0.848 ± 0.005       | 0.851 ± 0.006       | **0.852 ± 0.006**   |
| MNCA       | 0.812 ± 0.006       | 0.842 ± 0.006       | 0.849 ± 0.006       |
| TabPFNv2   | 0.789 ± 0.006       | 0.806 ± 0.008       | 0.837 ± 0.007       |
| TabDPT     | 0.798 ± 0.005       | -                   | -                   |
| TabICL     | 0.846 ± 0.006       | -                   | -                   |
| Linear     | 0.735 ± 0.007       | 0.735 ± 0.007       | 0.735 ± 0.007       |
| KNN        | 0.728 ± 0.009       | 0.778 ± 0.007       | 0.807 ± 0.006       |
| AutoGluon  | -                   | -                   | 0.847 ± 0.006       |

## jm1 (AUC ↑)

|            | Default           | Tuned             | Tuned + Ens.      |
|------------|-------------------|-------------------|-------------------|
| RF         | $0.752 \pm 0.008$ | $0.752 \pm 0.008$ | $0.761 \pm 0.007$ |
| ExtraTrees | $0.756 \pm 0.007$ | $\underline{0.758 \pm 0.008}$ | $0.765 \pm 0.006$ |
| XGBoost    | $0.748 \pm 0.007$ | $0.749 \pm 0.007$ | $0.752 \pm 0.006$ |
| LightGBM   | $0.748 \pm 0.006$ | $0.751 \pm 0.006$ | $0.753 \pm 0.006$ |
| CatBoost   | $0.744 \pm 0.005$ | $0.751 \pm 0.005$ | $0.749 \pm 0.005$ |
| EBM        | $0.735 \pm 0.006$ | $0.734 \pm 0.007$ | $0.736 \pm 0.007$ |
| FastAIMLP  | $0.728 \pm 0.007$ | $0.728 \pm 0.007$ | $0.733 \pm 0.006$ |
| TorchMLP   | $0.728 \pm 0.005$ | $0.734 \pm 0.005$ | $0.736 \pm 0.005$ |
| RealMLP    | $0.731 \pm 0.007$ | $0.735 \pm 0.007$ | $0.749 \pm 0.007$ |
| TabM       | $0.733 \pm 0.007$ | $0.738 \pm 0.004$ | $0.746 \pm 0.006$ |
| MNCA       | $0.762 \pm 0.003$ | $\underline{0.761 \pm 0.006}$ | $0.769 \pm 0.006$ |
| TabPFNv2   | $0.732 \pm 0.008$ | $\underline{0.755 \pm 0.007}$ | $\underline{0.773 \pm 0.006}$ |
| TabDPT     | $0.771 \pm 0.005$ | -                 | -                 |
| TabICL     | $\mathbf{\color{green}{0.776 \pm 0.005}}$ | -                 | -                 |
| Linear     | $\underline{0.724 \pm 0.006}$ | $0.723 \pm 0.006$ | $0.724 \pm 0.006$ |
| KNN        | $0.740 \pm 0.007$ | $0.750 \pm 0.007$ | $0.761 \pm 0.007$ |
| AutoGluon  | -                 | -                 | $0.760 \pm 0.007$ |

## kddcup09_appetency (AUC ↑)

|            | Default           | Tuned             | Tuned + Ens.      |
|------------|-------------------|-------------------|-------------------|
| RF         | $0.772 \pm 0.016$ | $0.822 \pm 0.011$ | $0.821 \pm 0.010$ |
| ExtraTrees | $0.771 \pm 0.012$ | $0.819 \pm 0.012$ | $0.821 \pm 0.009$ |
| XGBoost    | $0.830 \pm 0.012$ | $0.833 \pm 0.009$ | $0.837 \pm 0.010$ |
| LightGBM   | $0.798 \pm 0.009$ | $0.821 \pm 0.009$ | $0.829 \pm 0.010$ |
| CatBoost   | $\mathbf{0.846 \pm 0.008}$ | $\mathbf{0.845 \pm 0.008}$ | $\mathbf{0.845 \pm 0.008}$ |
| EBM        | $\underline{0.826 \pm 0.009}$ | $0.831 \pm 0.010$ | $\underline{0.833 \pm 0.010}$ |
| FastAIMLP  | $0.749 \pm 0.023$ | $0.795 \pm 0.013$ | $0.804 \pm 0.014$ |
| TorchMLP   | $0.819 \pm 0.012$ | $0.826 \pm 0.012$ | $0.831 \pm 0.013$ |
| RealMLP    | $0.820 \pm 0.011$ | $0.822 \pm 0.011$ | $0.832 \pm 0.011$ |
| TabM       | $0.806 \pm 0.011$ | $0.821 \pm 0.010$ | $0.821 \pm 0.010$ |
| MNCA       | $0.797 \pm 0.010$ | $0.814 \pm 0.012$ | $0.803 \pm 0.015$ |
| TabPFNv2   | -                 | -                 | -                 |
| TabDPT     | $0.742 \pm 0.009$ | -                 | -                 |
| TabICL     | $0.811 \pm 0.014$ | -                 | -                 |
| Linear     | $0.797 \pm 0.013$ | $0.820 \pm 0.014$ | $0.821 \pm 0.014$ |
| KNN        | $0.605 \pm 0.010$ | $0.760 \pm 0.012$ | $0.776 \pm 0.013$ |
| AutoGluon  | -                 | -                 | $\mathbf{\color{green}{0.846 \pm 0.009}}$ |

## maternal_health_risk (logloss ↓)

|            | Default           | Tuned             | Tuned + Ens.      |
|------------|-------------------|-------------------|-------------------|
| RF         | $0.479 \pm 0.071$ | $0.470 \pm 0.053$ | $0.448 \pm 0.054$ |
| ExtraTrees | $0.478 \pm 0.069$ | $0.453 \pm 0.055$ | $0.443 \pm 0.055$ |
| XGBoost    | $0.470 \pm 0.051$ | $0.459 \pm 0.051$ | $0.462 \pm 0.050$ |
| LightGBM   | $0.488 \pm 0.052$ | $0.462 \pm 0.049$ | $0.461 \pm 0.045$ |
| CatBoost   | $0.478 \pm 0.055$ | $0.463 \pm 0.053$ | $0.459 \pm 0.049$ |
| EBM        | $0.569 \pm 0.038$ | $0.562 \pm 0.042$ | $0.557 \pm 0.040$ |
| FastAIMLP  | $0.650 \pm 0.044$ | $0.617 \pm 0.040$ | $0.611 \pm 0.039$ |
| TorchMLP   | $0.606 \pm 0.054$ | $0.566 \pm 0.053$ | $0.554 \pm 0.046$ |
| RealMLP    | $0.558 \pm 0.056$ | $0.463 \pm 0.058$ | $0.436 \pm 0.049$ |
| TabM       | $0.513 \pm 0.045$ | $0.484 \pm 0.057$ | $0.469 \pm 0.051$ |
| MNCA       | $0.453 \pm 0.042$ | $\underline{0.442 \pm 0.047}$ | $\underline{0.428 \pm 0.050}$ |
| TabPFNv2   | $0.451 \pm 0.047$ | $\underline{0.439 \pm 0.057}$ | $\underline{0.437 \pm 0.057}$ |
| TabDPT     | $\mathbf{\color{green}{0.405 \pm 0.062}}$ | -                 | -                 |
| TabICL     | $\mathbf{0.410 \pm 0.058}$ | -                 | -                 |
| Linear     | $0.796 \pm 0.037$ | $0.793 \pm 0.038$ | $0.784 \pm 0.037$ |
| KNN        | $0.938 \pm 0.236$ | $0.810 \pm 0.126$ | $0.464 \pm 0.062$ |
| AutoGluon  | -                 | -                 | $0.462 \pm 0.061$ |

## miami_housing (rmse ↓)

|            | Default          | Tuned            | Tuned + Ens.     |
|------------|------------------|------------------|------------------|
| RF         | $9676 \pm 592$   | $9332 \pm 503$   | $9302 \pm 500$   |
| ExtraTrees | $9482 \pm 400$   | $9195 \pm 395$   | $9166 \pm 409$   |
| XGBoost    | $8650 \pm 447$   | $8062 \pm 361$   | $8042 \pm 357$   |
| LightGBM   | $8563 \pm 463$   | $8124 \pm 410$   | $7961 \pm 354$   |
| CatBoost   | $\underline{7985 \pm 315}$ | $7836 \pm 342$ | $\mathbf{7847 \pm 329}$ |
| EBM        | $10420 \pm 365$  | $9882 \pm 466$   | $9823 \pm 438$   |
| FastAIMLP  | $9034 \pm 512$   | $8855 \pm 535$   | $8664 \pm 483$   |
| TorchMLP   | $9265 \pm 455$   | $8631 \pm 511$   | $8528 \pm 466$   |
| RealMLP    | $8605 \pm 402$   | $8337 \pm 457$   | $8018 \pm 399$   |
| TabM       | $8307 \pm 486$   | $8115 \pm 449$   | $8023 \pm 428$   |
| MNCA       | $8813 \pm 434$   | $8307 \pm 380$   | $8015 \pm 409$   |
| TabPFNv2   | $8579 \pm 447$   | $\underline{7829 \pm 457}$ | $\mathbf{\color{green}{7711 \pm 442}}$ |
| TabDPT     | $8213 \pm 497$   | -                | -                |
| TabICL     | -                | -                | -                |
| Linear     | $19126 \pm 458$  | $17557 \pm 366$  | $17556 \pm 367$  |
| KNN        | $11957 \pm 565$  | $10441 \pm 503$  | $10134 \pm 548$  |
| AutoGluon  | -                | -                | $7873 \pm 429$   |

## online_shoppers_intention (AUC ↑)

|            | Default           | Tuned             | Tuned + Ens.      |
|------------|-------------------|-------------------|-------------------|
| RF         | $0.926 \pm 0.004$ | $0.931 \pm 0.004$ | $0.932 \pm 0.003$ |
| ExtraTrees | $0.918 \pm 0.005$ | $0.931 \pm 0.004$ | $0.931 \pm 0.004$ |
| XGBoost    | $0.935 \pm 0.004$ | $0.934 \pm 0.003$ | $0.936 \pm 0.003$ |
| LightGBM   | $0.934 \pm 0.003$ | $\underline{0.935 \pm 0.004}$ | $0.935 \pm 0.003$ |
| CatBoost   | $0.934 \pm 0.003$ | $0.933 \pm 0.004$ | $0.934 \pm 0.004$ |
| EBM        | $0.931 \pm 0.003$ | $0.931 \pm 0.003$ | $0.931 \pm 0.003$ |
| FastAIMLP  | $0.926 \pm 0.004$ | $0.931 \pm 0.005$ | $0.932 \pm 0.004$ |
| TorchMLP   | $0.930 \pm 0.004$ | $0.935 \pm 0.004$ | $0.936 \pm 0.003$ |
| RealMLP    | $0.929 \pm 0.004$ | $0.933 \pm 0.003$ | $0.934 \pm 0.004$ |
| TabM       | $0.935 \pm 0.003$ | $\underline{0.936 \pm 0.003}$ | $0.936 \pm 0.004$ |
| MNCA       | $0.934 \pm 0.003$ | $\underline{0.935 \pm 0.003}$ | $0.936 \pm 0.003$ |
| TabPFNv2   | $0.934 \pm 0.004$ | $\underline{0.937 \pm 0.003}$ | $\mathbf{\color{green}{0.937 \pm 0.003}}$ |
| TabDPT     | $0.926 \pm 0.005$ | -                 | -                 |
| TabICL     | $\mathbf{0.937 \pm 0.003}$ | -                 | -                 |
| Linear     | $0.913 \pm 0.007$ | $0.913 \pm 0.007$ | $0.918 \pm 0.006$ |
| KNN        | $0.795 \pm 0.005$ | $0.897 \pm 0.007$ | $0.922 \pm 0.004$ |
| AutoGluon  | -                 | -                 | $0.936 \pm 0.003$ |

## physiochemical_protein (rmse ↓)

|            | Default           | Tuned             | Tuned + Ens.      |
|------------|-------------------|-------------------|-------------------|
| RF         | $3.565 \pm 0.021$ | $3.463 \pm 0.021$ | $3.471 \pm 0.021$ |
| ExtraTrees | $3.548 \pm 0.022$ | $3.466 \pm 0.022$ | $3.474 \pm 0.021$ |
| XGBoost    | $3.513 \pm 0.024$ | $3.390 \pm 0.024$ | $3.390 \pm 0.024$ |
| LightGBM   | $3.477 \pm 0.026$ | $3.381 \pm 0.026$ | $3.384 \pm 0.027$ |
| CatBoost   | $3.522 \pm 0.026$ | $3.395 \pm 0.029$ | $3.383 \pm 0.027$ |
| EBM        | $4.241 \pm 0.020$ | $4.234 \pm 0.019$ | $4.226 \pm 0.020$ |
| FastAIMLP  | $4.014 \pm 0.027$ | $3.675 \pm 0.031$ | $3.683 \pm 0.034$ |
| TorchMLP   | $3.388 \pm 0.021$ | $3.289 \pm 0.028$ | $3.228 \pm 0.018$ |
| RealMLP    | $3.466 \pm 0.042$ | $3.284 \pm 0.029$ | $3.125 \pm 0.028$ |
| TabM       | $3.426 \pm 0.027$ | $3.273 \pm 0.031$ | $3.212 \pm 0.032$ |
| MNCA       | $3.170 \pm 0.040$ | $\underline{3.056 \pm 0.034}$ | $\underline{2.985 \pm 0.038}$ |
| TabPFNv2   | -                 | -                 | -                 |
| TabDPT     | $\mathbf{\color{green}{2.912 \pm 0.033}}$ | -                 | -                 |
| TabICL     | -                 | -                 | -                 |
| Linear     | $5.194 \pm 0.024$ | $5.188 \pm 0.022$ | $5.142 \pm 0.023$ |
| KNN        | $3.909 \pm 0.032$ | $3.688 \pm 0.042$ | $3.586 \pm 0.041$ |
| AutoGluon  | -                 | -                 | $3.107 \pm 0.026$ |

### polish_companies_bankruptcy (AUC ↑)

|  | Default | Tuned | Tuned + Ens. |
|---|---|---|---|
| RF | 0.927 ± 0.007 | 0.935 ± 0.008 | 0.938 ± 0.007 |
| ExtraTrees | 0.874 ± 0.013 | 0.891 ± 0.013 | 0.893 ± 0.011 |
| XGBoost | 0.958 ± 0.007 | 0.957 ± 0.007 | 0.957 ± 0.008 |
| LightGBM | 0.954 ± 0.007 | 0.955 ± 0.008 | 0.957 ± 0.007 |
| CatBoost | 0.961 ± 0.008 | 0.961 ± 0.008 | 0.960 ± 0.008 |
| EBM | 0.962 ± 0.009 | 0.962 ± 0.010 | 0.964 ± 0.009 |
| FastAIMLP | 0.841 ± 0.026 | 0.852 ± 0.034 | 0.862 ± 0.022 |
| TorchMLP | 0.903 ± 0.006 | 0.955 ± 0.006 | 0.957 ± 0.005 |
| RealMLP | 0.962 ± 0.004 | 0.957 ± 0.006 | 0.963 ± 0.006 |
| TabM | 0.951 ± 0.009 | 0.969 ± 0.005 | 0.970 ± 0.004 |
| MNCA | 0.962 ± 0.008 | 0.965 ± 0.009 | 0.968 ± 0.006 |
| TabPFNv2 | 0.959 ± 0.006 | 0.979 ± 0.003 | **0.981 ± 0.002** |
| TabDPT | 0.958 ± 0.009 | - | - |
| TabICL | 0.974 ± 0.002 | - | - |
| Linear | 0.867 ± 0.016 | 0.887 ± 0.011 | 0.896 ± 0.010 |
| KNN | 0.775 ± 0.029 | 0.840 ± 0.008 | 0.852 ± 0.011 |
| AutoGluon | - | - | 0.969 ± 0.005 |

### qsar-biodeg (AUC ↑)

|  | Default | Tuned | Tuned + Ens. |
|---|---|---|---|
| RF | 0.930 ± 0.013 | 0.929 ± 0.013 | 0.929 ± 0.013 |
| ExtraTrees | 0.932 ± 0.013 | 0.932 ± 0.013 | 0.934 ± 0.013 |
| XGBoost | 0.926 ± 0.013 | 0.931 ± 0.012 | 0.931 ± 0.012 |
| LightGBM | 0.927 ± 0.012 | 0.933 ± 0.012 | 0.933 ± 0.012 |
| CatBoost | 0.930 ± 0.013 | 0.931 ± 0.012 | 0.932 ± 0.012 |
| EBM | 0.931 ± 0.011 | 0.931 ± 0.011 | 0.933 ± 0.011 |
| FastAIMLP | 0.932 ± 0.013 | 0.932 ± 0.014 | 0.934 ± 0.013 |
| TorchMLP | 0.924 ± 0.014 | 0.923 ± 0.014 | 0.927 ± 0.014 |
| RealMLP | 0.927 ± 0.013 | 0.926 ± 0.015 | 0.934 ± 0.012 |
| TabM | 0.931 ± 0.011 | 0.934 ± 0.012 | 0.936 ± 0.012 |
| MNCA | 0.928 ± 0.012 | 0.928 ± 0.012 | 0.931 ± 0.012 |
| TabPFNv2 | 0.936 ± 0.011 | 0.932 ± 0.013 | 0.936 ± 0.012 |
| TabDPT | 0.934 ± 0.012 | - | - |
| TabICL | **0.938 ± 0.012** | - | - |
| Linear | 0.910 ± 0.016 | 0.923 ± 0.014 | 0.923 ± 0.014 |
| KNN | 0.907 ± 0.017 | 0.914 ± 0.015 | 0.923 ± 0.014 |
| AutoGluon | - | - | 0.934 ± 0.013 |

### seismic-bumps (AUC ↑)

|  | Default | Tuned | Tuned + Ens. |
|---|---|---|---|
| RF | 0.747 ± 0.021 | 0.767 ± 0.022 | 0.765 ± 0.026 |
| ExtraTrees | 0.734 ± 0.024 | 0.767 ± 0.029 | 0.771 ± 0.025 |
| XGBoost | 0.759 ± 0.022 | 0.768 ± 0.024 | 0.771 ± 0.025 |
| LightGBM | 0.752 ± 0.027 | 0.770 ± 0.027 | 0.771 ± 0.026 |
| CatBoost | **0.776 ± 0.027** | 0.772 ± 0.026 | 0.767 ± 0.028 |
| EBM | 0.770 ± 0.026 | 0.763 ± 0.026 | 0.767 ± 0.025 |
| FastAIMLP | 0.728 ± 0.034 | 0.759 ± 0.033 | 0.761 ± 0.028 |
| TorchMLP | 0.763 ± 0.026 | 0.758 ± 0.026 | 0.762 ± 0.025 |
| RealMLP | 0.760 ± 0.030 | 0.761 ± 0.027 | 0.766 ± 0.027 |
| TabM | 0.769 ± 0.024 | 0.768 ± 0.026 | 0.770 ± 0.026 |
| MNCA | **0.771 ± 0.023** | 0.763 ± 0.027 | 0.734 ± 0.026 |
| TabPFNv2 | 0.772 ± 0.025 | 0.766 ± 0.023 | 0.769 ± 0.024 |
| TabDPT | **0.774 ± 0.022** | - | - |
| TabICL | **0.783 ± 0.024** | - | - |
| Linear | 0.759 ± 0.024 | 0.761 ± 0.023 | 0.763 ± 0.025 |
| KNN | 0.685 ± 0.034 | 0.764 ± 0.028 | 0.760 ± 0.022 |
| AutoGluon | - | - | 0.758 ± 0.032 |

### splice (logloss ↓)

|  | Default | Tuned | Tuned + Ens. |
|---|---|---|---|
| RF | 0.317 ± 0.005 | 0.180 ± 0.015 | 0.178 ± 0.014 |
| ExtraTrees | 0.393 ± 0.007 | 0.175 ± 0.015 | 0.173 ± 0.013 |
| XGBoost | 0.119 ± 0.020 | 0.109 ± 0.018 | 0.109 ± 0.018 |
| LightGBM | 0.111 ± 0.016 | 0.103 ± 0.016 | 0.102 ± 0.015 |
| CatBoost | 0.110 ± 0.017 | 0.115 ± 0.018 | 0.111 ± 0.018 |
| EBM | 0.118 ± 0.013 | 0.119 ± 0.012 | 0.117 ± 0.012 |
| FastAIMLP | 0.118 ± 0.013 | **0.105 ± 0.013** | **0.103 ± 0.014** |
| TorchMLP | 0.157 ± 0.022 | 0.127 ± 0.017 | 0.116 ± 0.014 |
| RealMLP | 0.126 ± 0.014 | 0.110 ± 0.014 | 0.106 ± 0.013 |
| TabM | 0.110 ± 0.016 | 0.112 ± 0.019 | 0.110 ± 0.017 |
| MNCA | 0.144 ± 0.013 | 0.120 ± 0.011 | 0.122 ± 0.014 |
| TabPFNv2 | 0.107 ± 0.015 | 0.113 ± 0.019 | **0.099 ± 0.015** |
| TabDPT | 0.267 ± 0.010 | - | - |
| TabICL | 0.148 ± 0.020 | - | - |
| Linear | 0.167 ± 0.019 | 0.148 ± 0.012 | 0.148 ± 0.012 |
| KNN | 0.551 ± 0.017 | 0.485 ± 0.024 | 0.447 ± 0.025 |
| AutoGluon | - | - | **0.100 ± 0.017** |

### students_dropout_and_academic_success (logloss ↓)

|  | Default | Tuned | Tuned + Ens. |
|---|---|---|---|
| RF | 0.584 ± 0.011 | 0.576 ± 0.014 | 0.574 ± 0.013 |
| ExtraTrees | 0.591 ± 0.012 | 0.570 ± 0.011 | 0.566 ± 0.013 |
| XGBoost | 0.554 ± 0.016 | 0.545 ± 0.015 | 0.546 ± 0.014 |
| LightGBM | 0.555 ± 0.015 | 0.543 ± 0.016 | 0.542 ± 0.015 |
| CatBoost | 0.552 ± 0.016 | 0.543 ± 0.017 | 0.541 ± 0.016 |
| EBM | 0.565 ± 0.014 | 0.561 ± 0.013 | 0.560 ± 0.014 |
| FastAIMLP | 0.565 ± 0.021 | 0.549 ± 0.015 | 0.540 ± 0.015 |
| TorchMLP | 0.581 ± 0.018 | 0.559 ± 0.016 | 0.552 ± 0.014 |
| RealMLP | 0.556 ± 0.015 | 0.553 ± 0.013 | 0.542 ± 0.014 |
| TabM | 0.543 ± 0.013 | 0.541 ± 0.014 | 0.538 ± 0.014 |
| MNCA | 0.555 ± 0.015 | 0.554 ± 0.012 | 0.546 ± 0.013 |
| TabPFNv2 | 0.534 ± 0.012 | **0.529 ± 0.015** | **0.527 ± 0.015** |
| TabDPT | 0.561 ± 0.018 | - | - |
| TabICL | 0.550 ± 0.014 | - | - |
| Linear | 0.571 ± 0.017 | 0.562 ± 0.013 | 0.561 ± 0.013 |
| KNN | 0.829 ± 0.047 | 0.705 ± 0.018 | 0.663 ± 0.017 |
| AutoGluon | - | - | 0.536 ± 0.015 |

### superconductivity (rmse ↓)

|  | Default | Tuned | Tuned + Ens. |
|---|---|---|---|
| RF | 9.63 ± 0.19 | 9.62 ± 0.19 | 9.53 ± 0.19 |
| ExtraTrees | 9.43 ± 0.18 | 9.43 ± 0.18 | 9.39 ± 0.18 |
| XGBoost | 9.45 ± 0.18 | 9.35 ± 0.21 | 9.34 ± 0.20 |
| LightGBM | 9.41 ± 0.21 | 9.28 ± 0.22 | 9.26 ± 0.20 |
| CatBoost | 9.36 ± 0.19 | 9.34 ± 0.21 | 9.34 ± 0.20 |
| EBM | 10.53 ± 0.23 | 10.43 ± 0.22 | 10.21 ± 0.18 |
| FastAIMLP | 11.51 ± 0.10 | 10.59 ± 0.10 | 10.55 ± 0.12 |
| TorchMLP | 9.95 ± 0.21 | 9.66 ± 0.16 | 9.56 ± 0.16 |
| RealMLP | 9.57 ± 0.29 | 9.44 ± 0.23 | 9.22 ± 0.21 |
| TabM | 9.51 ± 0.21 | 9.31 ± 0.21 | 9.24 ± 0.20 |
| MNCA | 9.56 ± 0.16 | 9.50 ± 0.14 | 9.28 ± 0.19 |
| TabPFNv2 | - | - | - |
| TabDPT | **9.08 ± 0.20** | - | - |
| TabICL | - | - | - |
| Linear | 17.43 ± 0.10 | 17.42 ± 0.09 | 17.26 ± 0.10 |
| KNN | 11.10 ± 0.16 | 10.25 ± 0.26 | 9.95 ± 0.22 |
| AutoGluon | - | - | 9.22 ± 0.16 |

## taiwanese_bankruptcy_prediction (AUC ↑)

| | Default | Tuned | Tuned + Ens. |
|---|---|---|---|
| RF | $0.933 \pm 0.011$ | $0.932 \pm 0.011$ | $0.932 \pm 0.012$ |
| ExtraTrees | $0.937 \pm 0.011$ | $0.937 \pm 0.008$ | $0.938 \pm 0.008$ |
| XGBoost | $0.941 \pm 0.008$ | $\mathbf{0.943 \pm 0.008}$ | $\mathbf{0.944 \pm 0.008}$ |
| LightGBM | $0.938 \pm 0.008$ | $\mathbf{0.943 \pm 0.008}$ | $\mathbf{0.944 \pm 0.007}$ |
| CatBoost | $\mathbf{0.944 \pm 0.006}$ | $0.944 \pm 0.007$ | $0.943 \pm 0.006$ |
| EBM | $0.942 \pm 0.005$ | $0.940 \pm 0.005$ | $0.941 \pm 0.004$ |
| FastAIMLP | $0.914 \pm 0.022$ | $0.923 \pm 0.009$ | $0.927 \pm 0.010$ |
| TorchMLP | $0.925 \pm 0.009$ | $0.940 \pm 0.007$ | $\mathbf{0.943 \pm 0.007}$ |
| RealMLP | $0.941 \pm 0.006$ | $0.941 \pm 0.007$ | $\mathbf{0.945 \pm 0.006}$ |
| TabM | $0.941 \pm 0.004$ | $0.942 \pm 0.004$ | $0.943 \pm 0.004$ |
| MNCA | $0.940 \pm 0.006$ | $0.939 \pm 0.010$ | $0.932 \pm 0.013$ |
| TabPFNv2 | $0.942 \pm 0.005$ | $0.943 \pm 0.007$ | $\mathbf{0.945 \pm 0.008}$ |
| TabDPT | $0.937 \pm 0.008$ | - | - |
| TabICL | $0.944 \pm 0.006$ | - | - |
| Linear | $0.936 \pm 0.005$ | $0.938 \pm 0.003$ | $0.938 \pm 0.003$ |
| KNN | $0.870 \pm 0.017$ | $0.925 \pm 0.014$ | $0.936 \pm 0.007$ |
| AutoGluon | - | - | $\color{green}\mathbf{0.946 \pm 0.006}$ |

## website_phishing (logloss ↓)

| | Default | Tuned | Tuned + Ens. |
|---|---|---|---|
| RF | $0.312 \pm 0.037$ | $0.262 \pm 0.019$ | $0.257 \pm 0.019$ |
| ExtraTrees | $0.312 \pm 0.036$ | $0.258 \pm 0.020$ | $0.250 \pm 0.019$ |
| XGBoost | $0.260 \pm 0.027$ | $0.251 \pm 0.022$ | $0.251 \pm 0.023$ |
| LightGBM | $0.255 \pm 0.021$ | $0.249 \pm 0.021$ | $0.247 \pm 0.021$ |
| CatBoost | $0.252 \pm 0.022$ | $0.239 \pm 0.022$ | $0.239 \pm 0.021$ |
| EBM | $0.357 \pm 0.021$ | $0.357 \pm 0.020$ | $0.357 \pm 0.020$ |
| FastAIMLP | $0.334 \pm 0.023$ | $0.245 \pm 0.022$ | $0.241 \pm 0.021$ |
| TorchMLP | $0.289 \pm 0.035$ | $0.237 \pm 0.020$ | $0.230 \pm 0.019$ |
| RealMLP | $0.256 \pm 0.026$ | $0.236 \pm 0.026$ | $0.232 \pm 0.021$ |
| TabM | $0.246 \pm 0.026$ | $0.238 \pm 0.027$ | $0.236 \pm 0.025$ |
| MNCA | $0.261 \pm 0.023$ | $0.243 \pm 0.025$ | $0.239 \pm 0.022$ |
| TabPFNv2 | $0.229 \pm 0.025$ | $\mathbf{0.223 \pm 0.027}$ | $\color{green}\mathbf{0.222 \pm 0.025}$ |
| TabDPT | $0.228 \pm 0.027$ | - | - |
| TabICL | $0.228 \pm 0.026$ | - | - |
| Linear | $0.360 \pm 0.021$ | $0.359 \pm 0.023$ | $0.358 \pm 0.022$ |
| KNN | $0.938 \pm 0.142$ | $0.385 \pm 0.048$ | $0.299 \pm 0.033$ |
| AutoGluon | - | - | $0.233 \pm 0.021$ |

## wine_quality (rmse ↓)

| | Default | Tuned | Tuned + Ens. |
|---|---|---|---|
| RF | $0.620 \pm 0.021$ | $0.616 \pm 0.021$ | $0.611 \pm 0.021$ |
| ExtraTrees | $0.613 \pm 0.021$ | $0.606 \pm 0.025$ | $0.603 \pm 0.021$ |
| XGBoost | $0.621 \pm 0.021$ | $0.610 \pm 0.019$ | $0.610 \pm 0.020$ |
| LightGBM | $0.628 \pm 0.019$ | $0.607 \pm 0.019$ | $0.608 \pm 0.019$ |
| CatBoost | $0.621 \pm 0.019$ | $0.605 \pm 0.020$ | $0.605 \pm 0.020$ |
| EBM | $0.679 \pm 0.015$ | $0.678 \pm 0.016$ | $0.675 \pm 0.016$ |
| FastAIMLP | $0.669 \pm 0.019$ | $0.668 \pm 0.018$ | $0.657 \pm 0.019$ |
| TorchMLP | $0.691 \pm 0.019$ | $0.653 \pm 0.018$ | $0.650 \pm 0.015$ |
| RealMLP | $0.625 \pm 0.018$ | $0.618 \pm 0.019$ | $0.603 \pm 0.020$ |
| TabM | $0.637 \pm 0.020$ | $0.620 \pm 0.020$ | $0.612 \pm 0.020$ |
| MNCA | $0.610 \pm 0.020$ | $0.606 \pm 0.019$ | $0.601 \pm 0.020$ |
| TabPFNv2 | $0.692 \pm 0.012$ | $0.639 \pm 0.017$ | $0.610 \pm 0.020$ |
| TabDPT | $\color{green}\mathbf{0.590 \pm 0.018}$ | - | - |
| TabICL | - | - | - |
| Linear | $0.731 \pm 0.017$ | $0.731 \pm 0.017$ | $0.730 \pm 0.016$ |
| KNN | $0.627 \pm 0.023$ | $0.627 \pm 0.022$ | $0.617 \pm 0.021$ |
| AutoGluon | - | - | $0.599 \pm 0.020$ |

