## Appendix Contents.

# A Additional Experiments

## A.1 Alternative Leaderboard Versions

**Aggregation Methods.** Here, we provide more leaderboard variants, using different aggregation strategies. Specifically, we obtain errors $\text{err}_i$ for each dataset $i$ by averaging error metrics (1-AUROC for binary, logloss for multiclass, and RMSE for regression) over all outer folds. We then aggregate these errors as follows:

- **Elo**: As described in Section 2.3.

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

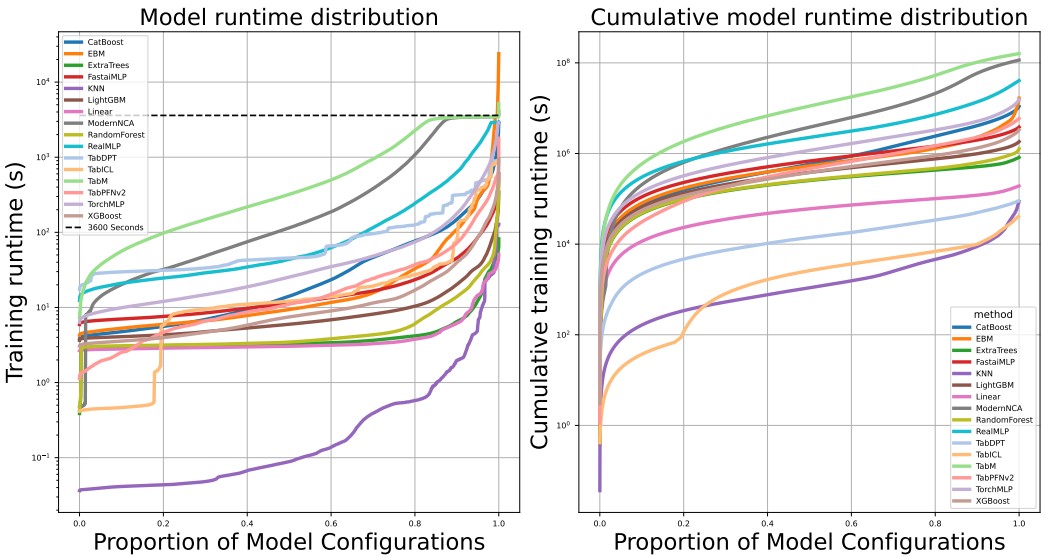

Figure A.8: **Training Runtime Analysis, Runtime Distribution (left) and Cumulative Total Runtime (right) Across Hyperparameter Configurations.** We show the training runtime in seconds for the hyperparameter configurations across models. Notably, TabM and ModernNCA on CPU are impacted by the 1-hour time limit in ∼16% of their configurations.

Table A.1: **TabArena-v0.1 Leaderboard.** We show default (D), tuned (T), and tuned + ensembled (T+E) performances. Results of TabPFNv2 and TabICL are imputed with RandomForest (default) for datasets on which they were not run. Times are median times per 1K samples across datasets, averaged over all outer folds per dataset. The best three values in columns are highlighted with gold, silver, and bronze colors. For Elo values, we also indicate their approximate 95% confidence intervals obtained through bootstrapping.

| Model | Elo (↑) | Norm. score (↑) | Avg. rank (↓) | Harm. mean rank (↓) | #wins (↑) | Improva-bility (↓) | Train time per 1K [s] | Predict time per 1K [s] |
|---|---|---|---|---|---|---|---|---|
| RealMLP (T+E) | $1574_{-30,+22}$ | **0.638** | **8.4** | **4.5** | 2 | **6.2%** | 6566.62 | 10.26 |
| LightGBM (T+E) | $1536_{-26,+29}$ | **0.583** | **9.7** | 5.3 | 2 | 8.1% | 417.05 | 2.64 |
| TabM (T+E) | $1534_{-29,+21}$ | **0.592** | **9.8** | 4.8 | 3 | **7.0%** | 38348.60 | 18.19 |
| CatBoost (T+E) | $1488_{-22,+23}$ | 0.555 | 11.5 | 7.2 | 0 | **7.5%** | 1658.43 | 0.65 |
| CatBoost (T) | $1475_{-27,+21}$ | 0.545 | 12.1 | 6.0 | 2 | 7.7% | 1658.43 | 0.08 |
| LightGBM (T) | $1453_{-24,+25}$ | 0.519 | 13.0 | 10.4 | 0 | 8.8% | 417.05 | 0.33 |
| XGBoost (T+E) | $1443_{-21,+24}$ | 0.502 | 13.4 | 7.8 | 1 | 8.9% | 693.49 | 1.69 |
| TabM (T) | $1434_{-32,+25}$ | 0.501 | 13.8 | 7.9 | 0 | 8.2% | 38348.60 | 2.04 |
| CatBoost (D) | $1432_{-24,+27}$ | 0.508 | 13.9 | 6.9 | 2 | 8.8% | 6.83 | 0.08 |
| ModernNCA (T+E) | $1428_{-28,+23}$ | 0.519 | 14.1 | 5.6 | 3 | 8.7% | 20604.60 | 62.20 |
| TabPFNv2 (T+E) | $1414_{-26,+25}$ | 0.509 | 14.8 | **3.1** | **11** | 8.1% | 3031.50 | 21.44 |
| XGBoost (T) | $1407_{-21,+24}$ | 0.467 | 15.0 | 11.9 | 0 | 9.2% | 693.49 | 0.31 |
| ModernNCA (T) | $1389_{-26,+30}$ | 0.429 | 15.9 | 10.0 | 0 | 9.3% | 20604.60 | 3.08 |
| TabICL (D) | $1388_{-23,+23}$ | 0.453 | 15.9 | 4.3 | 7 | 8.8% | 6.63 | 1.48 |
| RealMLP (T) | $1350_{-21,+26}$ | 0.385 | 17.6 | 14.7 | 0 | 9.7% | 6566.62 | 0.49 |
| TabPFNv2 (T) | $1345_{-23,+22}$ | 0.399 | 18.0 | 5.6 | 1 | 10.2% | 3031.50 | 0.46 |
| TabM (D) | $1339_{-20,+24}$ | 0.370 | 18.2 | 11.9 | 0 | 11.2% | 65.60 | 1.01 |
| TorchMLP (T+E) | $1333_{-19,+28}$ | 0.350 | 18.4 | 13.3 | 0 | 10.2% | 2875.52 | 1.95 |
| TabPFNv2 (D) | $1317_{-27,+28}$ | 0.370 | 19.3 | 5.5 | 4 | 11.2% | 3.36 | 0.31 |
| ModernNCA (D) | $1317_{-22,+23}$ | 0.314 | 19.3 | 11.1 | 1 | 12.8% | 43.53 | 1.45 |
| TabDPT (D) | $1297_{-23,+25}$ | 0.369 | 20.4 | 4.8 | 7 | 11.9% | 22.53 | 8.55 |
| EBM (T+E) | $1289_{-24,+26}$ | 0.271 | 20.7 | 11.7 | 1 | 14.3% | 1331.68 | 0.20 |
| FastaiMLP (T+E) | $1250_{-19,+24}$ | 0.243 | 22.6 | 11.7 | 1 | 13.6% | 593.24 | 4.47 |
| RealMLP (D) | $1246_{-22,+29}$ | 0.236 | 22.8 | 18.7 | 0 | 12.2% | 21.86 | 0.84 |
| ExtraTrees (T+E) | $1243_{-28,+25}$ | 0.230 | 22.9 | 14.9 | 0 | 13.9% | 183.02 | 0.76 |
| EBM (T) | $1234_{-23,+27}$ | 0.219 | 23.4 | 16.4 | 0 | 14.9% | 1331.68 | 0.02 |
| XGBoost (D) | $1227_{-25,+30}$ | 0.256 | 23.5 | 18.1 | 0 | 12.3% | 1.94 | 0.12 |
| TorchMLP (T) | $1221_{-29,+25}$ | 0.238 | 23.8 | 20.1 | 0 | 12.2% | 2875.52 | 0.13 |
| LightGBM (D) | $1206_{-28,+28}$ | 0.241 | 24.7 | 21.9 | 0 | 13.1% | 1.96 | 0.14 |
| RandomForest (T+E) | $1203_{-29,+22}$ | 0.187 | 24.8 | 12.8 | 1 | 14.7% | 373.24 | 0.77 |
| EBM (D) | $1202_{-30,+24}$ | 0.200 | 24.8 | 13.1 | 1 | 15.9% | 4.67 | 0.04 |
| ExtraTrees (T) | $1198_{-29,+22}$ | 0.188 | 25.1 | 16.5 | 0 | 15.0% | 183.02 | 0.09 |
| FastaiMLP (T) | $1158_{-20,+23}$ | 0.147 | 26.8 | 21.1 | 0 | 15.2% | 593.24 | 0.31 |
| RandomForest (T) | $1149_{-26,+26}$ | 0.150 | 27.2 | 16.5 | 0 | 15.7% | 373.24 | 0.09 |
| TorchMLP (D) | $1067_{-25,+27}$ | 0.076 | 30.6 | 27.8 | 0 | 17.1% | 9.99 | 0.13 |
| FastaiMLP (D) | $1008_{-24,+31}$ | 0.057 | 32.7 | 29.7 | 0 | 20.4% | 2.86 | 0.37 |
| RandomForest (D) | $1000_{-0,+0}$ | 0.052 | 33.0 | 31.2 | 0 | 20.9% | 0.43 | 0.05 |
| ExtraTrees (D) | $969_{-21,+36}$ | 0.073 | 34.0 | 30.3 | 0 | 22.7% | 0.25 | 0.05 |
| Linear (T+E) | $919_{-28,+29}$ | 0.044 | 35.6 | 25.4 | 0 | 30.6% | 47.50 | 0.17 |
| Linear (T) | $883_{-29,+22}$ | 0.036 | 36.5 | 31.9 | 0 | 31.3% | 47.50 | 0.07 |
| Linear (D) | $859_{-29,+33}$ | 0.031 | 37.1 | 29.8 | 0 | 32.7% | 1.52 | 0.09 |
| KNN (T+E) | $683_{-26,+24}$ | 0.005 | 40.5 | 40.2 | 0 | 45.2% | 3.26 | 0.18 |
| KNN (T) | $608_{-41,+33}$ | 0.000 | 41.4 | 41.3 | 0 | 46.8% | 3.26 | 0.04 |
| KNN (D) | $456_{-47,+46}$ | 0.000 | 42.8 | 42.6 | 0 | 54.1% | 0.05 | 0.02 |

## A.3 Tabular Deep Learning on GPU vs. CPU

In our main experiments, we ran TabM, ModernNCA, and RealMLP on CPU instead of GPU to make our experiments affordable. As observed in Appendix A.2, GPU-optimized models, especially TabM and ModernNCA, reach the per-configuration limit of 1 hour for $\sim15\%$ of their hyperparameter configurations. In these cases, their training is gracefully stopped, and a potentially non-converged checkpoint is used for inference.

To further investigate the impact of this early stopping on predictive performance, we ran the first 50 from the 200 configurations from TabM and ModernNCA on GPU. Figure A.9 shows that for TabM and ModernNCA, models trained on GPU outperform their CPU counterparts. Figure A.10 further

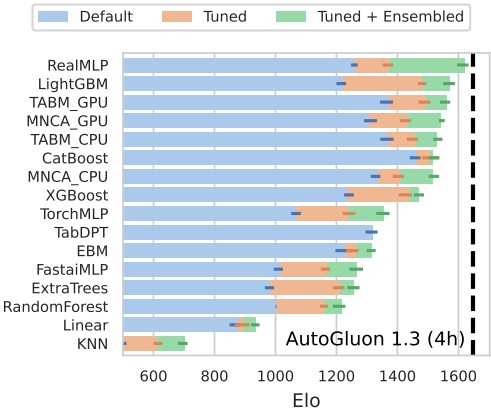

Figure A.9: **Predictive Performance of TabM and ModernNCA on CPU vs. GPU.** We show the predictive performance of TabM and ModernNCA when run on CPU (TabM_CPU, MNCA_CPU) and GPU (TabM_GPU, MNCA_GPU) across TabArena. For this study, the number of configurations for TabM and ModernNCA is limited to 50 for both CPU and GPU, while the other models use 200 configurations. Therefore, this figure should only be used to conclude that GPU improves results over CPU, rather than drawing conclusions on performance compared to other models.

demonstrates that the hyperparameter configurations of TabM and ModernNCA train much faster on GPU than on CPU. Moreover, we observe that only a tiny proportion ($\sim$0.1%) of configurations trained on GPU are affected by the time limit. We conclude from this ablation that training TabM, ModernNCA, and RealMLP on CPU with a time limit of 1 hour negatively influences their predictive performance. While the influence is marginal for RealMLP, it seems non-marginal for TabM and ModernNCA and could change the ranking on the full leaderboard.
As maintainers of `TabArena`, we aim to remedy this negative influence as a first proof-of-concept of the living benchmark. We are rerunning all 200 configurations for TabM, ModernNCA, and RealMLP on GPU and will update `TabArena` with the new results.

## A.4   TabArena-Lite

Benchmarking can quickly become very expensive, especially with a sophisticated protocol to guarantee robust results. To reduce the cost of benchmarking, we introduce `TabArena-Lite`. `TabArena-Lite` is a continually maintained subset of `TabArena` that consists, in its first version, of all datasets with one outer fold. Figure A.11 shows results on `TabArena-Lite`, using 200 hyperparameter configurations per model, but only a single outer fold for all datasets. The results are similar to the results on `TabArena` in Figure 1, showing that `TabArena-Lite` is a good indicator of model performance.
To further reduce the cost of benchmarking, we also recommend running new models on `TabArena-Lite` with one default hyperparameter configuration and optionally with a lower number of random hyperparameter configurations (e.g., 25). As all other models in `TabArena` are tuned, a less heavily tuned model that performs comparably could already show that a new model is promising. We designate `TabArena-Lite` to be used in academic studies and find any novel model that outperforms other models on at least one dataset, even if it is not among the best on average, a valuable publication. Furthermore, we as maintainers use the performance on `TabArena-Lite` to prioritize the integration of new models into `TabArena`. We envision `TabArena-Lite` also as a living, continuously updated subset. Ideally, future work could determine a method that finds the optimal and most representative subset of partitions and datasets in `TabArena` to populate `TabArena-Lite`.

## A.5   Investigating Statistical Significance

We investigate the statistical significance between models by using critical difference diagrams (CDDs) to represent the results of a Friedman test and then a Nemenyi post-hoc test ($\alpha = 0.05$) from

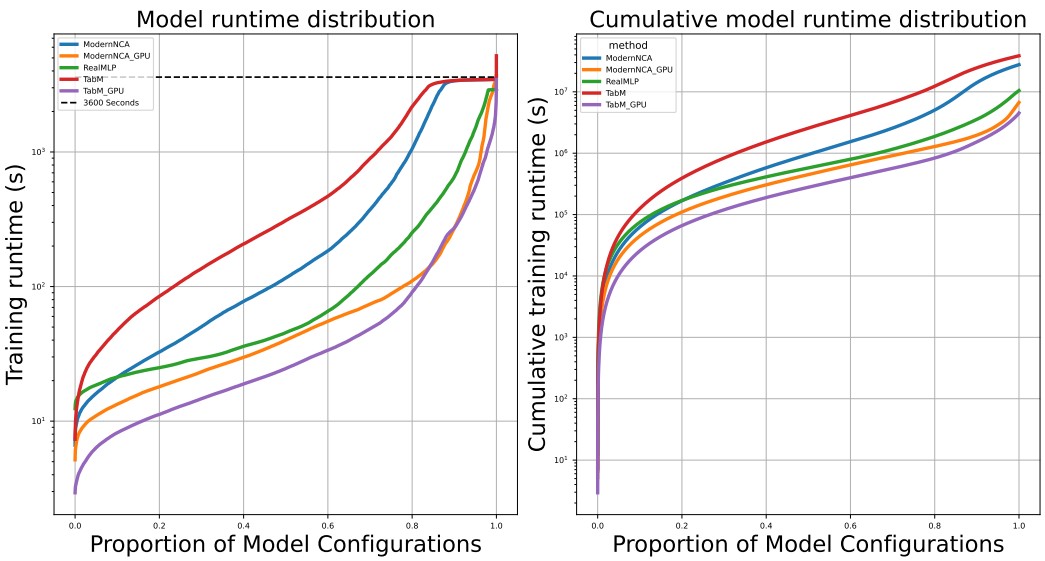

Figure A.10: **Runtime Analysis of TabM and ModernNCA on CPU vs. GPU.** We show the training runtime distribution of 50 hyperparameter configurations for TabM and ModernNCA trained on CPU and GPU across all TabArena datasets. For CPU, approximately $16\%$ of runs are early stopped due to the 1-hour time limit. For GPU, less than $0.1\%$ of runs are early stopped due to the time limit. We also include RealMLP trained on CPU for 50 configurations, for which less than $3\%$ of the configurations were early stopped based on time.

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

.1"). Namely, we included the following datasets: wine_quality [52], in_vehicle_coupon_recommendation [53], HR_Analytics_Job_Change_of_Data_Scientists [54], houses [55], hiva_agnostic [56], heloc [57], healthcare_insurance_expenses [58], hazelnut-spread-contaminant-detection [59], GiveMe-SomeCredit [60], Food_Delivery_Time [61], Fitness_Club [62], E-CommereShippingData [63], diamonds [64], Diabetes130US [65], diabetes [66], customer_satisfaction_in_airline [67], credit_card_clients_default [68], credit-g [69], concrete_compressive_strength [70], coil2000_insurance_policies [71], churn [72], blood-transfusion-service-center [73], Biore-sponse [74], Bank_Customer_Churn [75], bank-marketing [76, 77], APSFailure [78], Another-Dataset-on-used-Fiat-500 [79], anneal [80], Amazon_employee_access [81], air-foil_self_noise [82], Is-this-a-good-customer [83], jm1 [84], kddcup09_appetency [85], Market-ing_Campaign [86], maternal_health_risk [87], miami_housing [88], MIC [89], NATICUSdroid [90], online_shoppers_intention [91], physiochemical_protein [92], polish_companies_bankruptcy [93], qsar-biodeg [94], QSAR-TID-11 [95], QSAR_fish_toxicity [96], SDSS17 [97], seismic-bumps [98], splice [99], students_dropout_and_academic_success [100], superconductivity [101], taiwanese_bankruptcy_prediction [102], website_phishing [103].

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

1425 5  task_metadata = {
1426 6      task_id: openml.tasks.get_task(task_id).get_dataset().name
1427 7      for task_id in task_ids
1428 8  }
1429 9  methods = get_configs_for_custom_rf(num_random_configs=1)
1430 10
1431 11 run_experiments(
1432 12     expname="/path/to/output/dir",
1433 13     tids=task_ids,
1434 14     task_metadata=task_metadata,
1435 15     methods=methods,
1436 16     # TabArena-Lite - only run on the first split of each dataset.
1437 17     repeat_fold_pairs=[(0, 0)],
1438 18     folds=None,
1439 19     repeats=None,
1440 20 )
```

Listing 4: Comparing the custom RandomForest model to the leaderboard.

```
1441 1  import pandas as pd
1442 2  from tabrepo import EvaluationRepository
1443 3  from tabrepo.nips2025_utils.load_final_paper_results import
1444     load_paper_results
1445 4  from tabrepo.paper.paper_runner_tabarena import PaperRunTabArena
1446 5
1447 6  from . import post_process_local_results
1448 7  from . import rename_default
1449 8
1450 9  # Prepare local results (e.g., simulate HPO and ensembling)
1451 10 repo_dir = post_process_local_results(output_dir="/path/to/output/dir"
1452     )
1453 11 repo = EvaluationRepository.from_dir(repo_dir)
1454 12 repo.set_config_fallback(repo.configs()[0])
1455 13 plotter = PaperRunTabArena(repo=repo, output_dir="example_artifacts",
1456     backend="native")
1457 14 df_results = plotter.run_no_sim()
1458 15 is_default = df_results["framework"].str.contains("_c1_") & (
1459     df_results["method_type"] == "config")
```

```
1460 16 df_results.loc[is_default, "framework"] = df_results.loc[is_default]["
1461    config_type"].apply(rename_default)
1462 17 datasets = list(df_results["dataset"].unique())
1463 18 folds = list(df_results["fold"].unique())
1464 19 config_types = list(df_results["config_type"].unique())
1465 20
1466 21 # Load and prepare pre-computed results
1467 22 pre_df_results, _, _, _ = load_paper_results()
1468 23 pre_df_results = pre_df_results[pre_df_results["fold"].isin(folds) &
1469    pre_df_results["dataset"].isin(datasets)]
1470 24 df_results = PaperRunTabArena.compute_normalized_error_dynamic(
1471    df_results=pd.concat([df_results, pre_df_results], ignore_index=
1472    True))
1473 25