# OpenReview forum: "TabArena: A Living Benchmark for Machine Learning on Tabular Data"
_NeurIPS.cc/2025/Datasets_and_Benchmarks_Track — NeurIPS 2025 Datasets and Benchmarks Track spotlight_

### Official Review · Reviewer_NWc4 · 2025-06-18

**Rating:** 5
**Confidence:** 3

**Summary:**

The paper introduces TabArena, the first living benchmark for machine learning on tabular data.

- A curated collection of 51 real-world IID tabular datasets is distilled from 1053 candidates through a rigorous, largely manual vetting pipeline. Sixteen well-implemented models are benchmarked.
- A public leaderboard, reproducible code, pre-computed predictions, and explicit maintenance protocols turn TabArena into a continuously updated community resource.
- Empirical findings show that (i) post-hoc ensembling is vital for peak performance, (ii) modern neural networks catch up with—or surpass—GBDTs under generous time budgets, (iii) tabular foundation models dominate small datasets, and (iv) cross-model ensembles advance the SOTA.

**Dataset Code Accessibility:**

Yes

**Dataset Code Comments:**

The running leaderboard can be found in https://huggingface.co/spaces/TabArena/leaderboard.

**Ethical Considerations:**

No, there are no or only very minor ethics concerns

**Final Justification:**

The authors have in general resolved my questions regarding Elo. I hope the discussions could be added to the revised paper. Given my positive rating, I shall keep my score.

**Limitations Weaknesses:**

- While elegant, Elo aggregation is unfamiliar in tabular ML and may obscure absolute metric performance.
- The current version covers only small- to medium-sized IID classification and regression tasks; it does not address temporal, non-IID, very small or very large, or unsupervised problems. For example, recent work has shown that tabular LLMs excel at semi-supervised learning, which could potentially be benchmarked [1]. This is not a criticism, however.

[1] Zhang Q, Tan Y S, Tian Q, et al. TabPFN: One Model to Rule Them All?[J]. arXiv preprint arXiv:2505.20003, 2025.

**Strengths Contributions:**

- The benchmark is explicitly maintained as software, filling a long-standing gap in tabular ML research.
- Manual, criteria-driven filtering removes duplicates, leaks and licensing issues, producing a cleaner corpus than prior work.
- Clear take-aways on GBDT vs DL, ensemble utility, and foundation-model behaviour on small data.
- The writing and presentation are great.

---

> ### Author Rebuttal · Authors · 2025-07-30
>
> Dear Reviewer NWc4,
>
> We appreciate your straightforward feedback. We reply to your feedback regarding Elo below.
>
> ---
>
> ### Elo may obscure metric performance
>
> We want to give a clear ranking of models as part of the leaderboard. Thus, we decided on one “main” metric while allowing users to re-sort the leaderboard according to several other metrics. This was not a trivial decision as part of the evaluation design because every aggregation metric has its own pitfalls, and none is perfect. We slightly favor Elo, as we discuss below.
>
> Elo is based on pairwise comparison scoring, which only considers wins, ties, or losses, and neglects the magnitude of performance differences. This can be a disadvantage, since minor performance differences could be considered irrelevant by practitioners in specific applications. At the same time, this means that each dataset contributes equally to Elo, hence the aggregation is not biased towards certain domains or dataset properties (e.g., small or non-noisy datasets). For TabArena, this is a key advantage because we want to create a benchmark whose results are representative of all domains and datasets. In general, we expect Elo to be more robust towards choices in dataset selection, and it has more statistical power (as seen by the small confidence intervals).
>
> To add a complementary point of view to the main paper, we will add plots with improvability, a metric that measures the performance of methods relative to the best method, and is therefore sensitive to the magnitude of performance differences. Specifically, we plan to add two new figures in the main text: 1) showing the Pareto front of Improvability and inference time, and 2) showing the performance over (tuning) time related to Improvability.
>
> ---
>
> Thank you for the reference on tabular LLMs for semi-supervised learning. While we have focused on common supervised learning methods so far, future iterations of TabArena can also investigate semi-supervised learning.
>
> Thank you and kind regards,
>
> The Authors

---

> > ### Comment · Reviewer_NWc4 · 2025-08-02
> >
> > The authors have in general resolved my questions regarding Elo. I hope the discussions could be added to the revised paper. Given my positive rating, I shall keep my score.

---

### Official Review · Reviewer_txew · 2025-06-27

**Rating:** 5
**Confidence:** 4

**Summary:**

TabArena is a “living” benchmark platform for tabular data models – meaning it is continuously updated and maintained rather than a static, one-off benchmark. The authors identify that prior tabular ML benchmarks have become outdated or flawed (e.g. using old datasets, poor protocols, no updates), and propose TabArena as a solution. To launch TabArena, they curate 51 real-world tabular datasets (from an initial pool of 1053) and 16 state-of-the-art tabular models (including 3 recent “foundation” models for tabular data) with well-defined hyperparameter search spaces. They conduct an extensive experimental study (~25 million model training runs) to initialize a public leaderboard, and release all reproducible code, results, and a website (tabarena.ai) for the community. The benchmark currently focuses on standard supervised tabular classification and regression tasks on independent and identically distributed (IID) data in the “small-to-medium” data regime, reflecting the most common real-world use cases. Key findings from TabArena’s initial results include: (A) Ensembling tuned hyperparameter configurations yields the best performance for individual models; (B) Deep learning methods can match or surpass gradient-boosted tree models when given sufficient tuning and ensembling; (C) Tabular foundation models (with in-context learning) excel on small datasets even without tuning; and (D) Ensembles that combine different model types achieve the overall state-of-the-art, although not all models contribute equally to these ensembles. In summary, TabArena’s contribution is a continuously improving benchmark infrastructure – complete with curated datasets, models, evaluation protocols, and an active maintenance plan – aimed at providing a reliable, up-to-date assessment of tabular ML methods.

**Additional Feedback:**

The scope of TabArena’s initial study is very large (25M model training runs, ~15 years of compute in aggregate). This raises the concern that only big organizations could replicate or extend such a benchmark. However, the authors have mitigated this by releasing all results and a “TabArena-Lite” subset for cheaper reuse. This is a great idea. I encourage the maintainers to continue providing resource-friendly options (such as a smaller benchmark track or a way to test on a representative subset) so that researchers without extensive compute can still participate. Perhaps future work could explore efficient approximations of the full benchmark (e.g., using learning curve extrapolation to predict a model’s performance with less training) to reduce cost. Also, given the environmental impact of such large-scale experiments, it might be worth noting if any steps were taken to offset or minimize it (though this is not required, it’s increasingly on reviewers’ radar).

**Dataset Code Accessibility:**

Yes

**Dataset Code Comments:**

The dataset and code are very accessible and well-documented, which greatly facilitates reproducibility. The authors have released the entire codebase for TabArena (as reproducible pipelines) and even a public web interface (leaderboard) for viewing results and contributing models. The 51 curated datasets themselves are all from public sources (like UCI, OpenML, etc.) with permissive licenses, and the authors made sure that each dataset can be automatically downloaded via an API or is hosted for access. They also provide detailed documentation for each dataset (via the dataset-curation notes on the website) including any preprocessing or notable aspects. The combination of the released precomputed results, scripts, and open data means that one can reproduce the benchmark results or test new models without undue difficulty. In the supplementary material, protocols for adding new datasets (Appendix D.3) or models (Appendix D.4) are given, demonstrating the thorough documentation. I was impressed by the fact that the authors shared rich metadata and even model prediction outputs – this openness enables others to perform further analysis (e.g., ensembling different authors’ models or error analysis) without rerunning everything. In summary, the reproducibility is excellent: all key resources are publicly available and clearly organized. This is a strong point in favor of the submission.

**Ethical Considerations:**

No, there are no or only very minor ethics concerns

**Limitations Weaknesses:**

1. By applying very strict selection criteria, TabArena-v0.1 includes only 51 datasets, which, although high in quality, is a relatively small number. This drastically reduced dataset count may limit the diversity of domains and problem types represented. In particular, the benchmark currently covers only IID tabular tasks in the small/medium data range – it explicitly excludes non-IID scenarios (e.g. time-series or grouped data), very small datasets (<500 samples), very large datasets (>250k samples), and other ML tasks like clustering or anomaly detection. As a result, the conclusions drawn (and the leaderboard rankings) may not generalize to those important settings. For example, a model that excels on time-series data or extremely large data is not tested here. Constructive suggestion: This narrow focus is understandable for an initial version, but expanding the benchmark to include more diverse datasets is important. The authors already plan to incorporate “tiny, large, and non-IID data” and additional tasks in future versions, which is great. Working with the community to contribute a broader set of datasets (as invited in the paper) will increase the benchmark’s representativeness. In the meantime, authors might consider explicitly reporting the domain coverage of the current 51 datasets and ensure that future additions fill in gaps (e.g., more datasets from under-represented domains or with different distributions) to improve the benchmark’s generality.

2. Constraints of the Hyperparameter Tuning Protocol: TabArena’s model evaluation relies on a fixed budget of 200 randomly-sampled hyperparameter configurations per model (plus the default configuration). This uniform random search approach, while fair and simple, has some drawbacks. It prevents using more sophisticated hyperparameter optimization strategies (e.g. Bayesian optimization or successive halving) that might find better model settings more efficiently. It also means we don’t see how performance might improve beyond 200 trials, nor the variance in outcome if a different random seed were chosen for the search (the authors note that the variance of random choices isn’t analyzed due to this fixed set). Additionally, some models with many hyperparameters or slower training times might be disadvantaged by the fixed budget relative to simpler models. Constructive suggestion: For future iterations, the authors could consider more adaptive HPO methods or at least analyze the returns of scaling up the number of configurations. For instance, a study on whether 200 trials is enough for each model or if certain models plateau earlier would be valuable. If certain models consistently underperform due to under-tuning, allowing model-specific tuning budgets or smarter search (even if not in the main leaderboard, perhaps in an appendix experiment) could ensure the comparison reflects each model’s true potential. Alternatively, employing techniques to ensemble or reuse trials across similar models might be interesting. The current approach is workable, but there is room to make the HPO both more efficient and thorough, which would strengthen the benchmark’s assessment of “peak performance” for each method.

**Strengths Contributions:**

1. This work addresses a clear need for better tabular ML benchmarks. It is the first continuously maintained “living” benchmark for tabular data models, which is a notable paradigm shift from static benchmarks. The benchmark is treated as a community-driven, versioned software project with maintainers, ensuring that new models, new data, and fixes can be incorporated over time. This approach is highly significant because it tackles the common problems of stale benchmarks and unaddressed flaws. The authors systematically compare TabArena to prior benchmarks (Table 2) and demonstrate that TabArena consolidates the best aspects of previous studies while adding active maintenance and openness that others lacked. No prior tabular benchmark had built-in update protocols or a public live leaderboard, so TabArena’s open-source, evolving benchmark approach is novel and likely to set a new standard in the field.
2. The evaluation design in TabArena is carefully considered and clearly explained. To ensure robust comparisons, the authors employ repeated k-fold cross-validation with multiple repeats (instead of a single split) – using up to 10×3-fold CV on smaller datasets – so that results are not unduly influenced by random train/test splits. They adopt a pairwise Elo rating system (inspired by Chatbot Arena) to aggregate performance across many datasets in a statistically sound way; a 400-point Elo difference corresponds to a 91% win probability, and they calibrate 1000 Elo to a baseline random forest’s performance. This yields a stable leaderboard with confidence intervals (obtained via bootstrapping) rather than just reporting raw accuracy numbers. The protocols also include nested evaluation: for example, they highlight that using a simple hold-out validation would underestimate performance and bias rankings (favoring methods that inherently ensemble) – an issue they illustrate in Figure 5. By avoiding the “holdout fallacy” and using nested cross-validation, TabArena’s benchmarking is much more rigorous and fair. The paper details many such precautions (evaluation metrics per task type, tracking of training/inference time, etc.), and even provides scripts to regenerate the leaderboard and plots, indicating a high level of clarity and reproducibility in the evaluation setup. Overall, the evaluation protocols are comprehensive, well-justified, and likely to produce reliable comparisons.
3. he submission strongly emphasizes reproducibility and openness, which is a vital strength. The entire TabArena benchmark is implemented as an open-source platform (the paper mentions that it is available as code, with a public leaderboard at tabarena.ai and versioned as TabArena-v0.1). The authors have released all their code, pipelines, and even precomputed results for the initial benchmark, enabling others to reproduce the findings or easily evaluate new models against the benchmark. They log rich metadata from each run (training times, hardware used, hyperparameters, predictions, etc.) and share this data, which aids in reproducibility and further analysis. Notably, because the full benchmark is computationally expensive (by their estimate, ~15 years of total CPU/GPU time were used in all experiments), the authors have released the result artifacts (model predictions, performance metrics, etc.) so that others can “compare novel models cheaply” without rerunning all baselines from scratch. They also introduce a lighter variant, TabArena-Lite, which uses a single CV fold on all datasets to enable faster benchmarking for academic studies. This level of openness – code, data, results, and an avenue for community submissions (Appendix D.4 describes how researchers can submit new model results to the live leaderboard) – demonstrates a strong commitment to reproducibility. Anyone in the community can build upon TabArena, which greatly increases the utility and trustworthiness of the benchmark.

---

> ### Author Rebuttal · Authors · 2025-07-30
>
> Dear Reviewer txew,
>
> We appreciate your constructive feedback, which already helped us improve the manuscript. We reply to your comments on “Limitations Weaknesses” and “Additional Feedback” below, using your numbers where possible.
>
> ---
>
> ###  W1) Domain Coverage
>
> We provide an overview of domain coverage below. Moreover, we will add this overview and its discussion to the appendix. We follow the domain categorization of the TALENT benchmark [1] and apply it to our datasets.
>
> | Domain | Count | % |
> |---|---:|---:|
> | business & marketing | 16 | 31.37% |
> | finance | 8 | 15.69% |
> | chemistry & material science | 6 | 11.76% |
> | medical & healthcare | 6 | 11.76% |
> | biology & life sciences | 5 | 9.80% |
> | technology & internet | 4 | 7.84% |
> | physics & astronomy | 3 | 5.88% |
> | education | 1 | 1.96% |
> | environmental science & climate | 1 | 1.96% |
> | industry & manufacturing | 1 | 1.96% |
>
> ---
>
> ### W2) Hyperparameter Tuning Protocol
>
> We agree that an extended analysis of the hyperparameter tuning protocol would be an interesting area for future work. We provide an extended discussion on this below.
>
> For the camera-ready version, we will add two new figures in the main text: 1) showing the Pareto front of Improvability and inference time, and 2) showing the performance over (tuning) time related to Improvability. Figure 2) will show the (average) convergence behavior of models with random search, showing a mixed picture: some models seem to have converged (e.g., ExtraTrees, XGBoost); some seem they could still benefit from more HPO (e.g., RealMLP, Torch MLP); and some even got worse with more HPO due to overtuning [7] (e.g., ModernNCA, CatBoost).
>
> We aim to explore more sophisticated HPO methods and facilitate their use, particularly as part of reference pipelines. Following the arguments by Tschalzev et al. [2], we aim to establish reference pipelines that can fully harness the potential of HPO, thereby providing a stronger dataset-specific reference point.
>
> For TabArena-v0.1, we decided to use a fixed set of random hyperparameter configurations (called dense random search) instead of other strategies due to two reasons: 1) trivial parallelization, and 2) easier support for simulation and meta-learning. We believe that the latter reason outweighs limitations introduced by dense random search (as mentioned in Section 5, Limitations (1)).
>
> To clarify in more detail, 1) allows for running the benchmark faster in parallel. 2) allows us to use post-hoc simulation and meta-learning to build portfolios. Thus, dense random search not only enables a plethora of exciting future research, but it also matches the way models are used in state-of-the-art predictive machine learning systems (such as AutoGluon or Auto-Sklearn 2). Lastly, with dense random search, we can simulate meta-learned portfolios of each model in future iterations of TabArena, which we would expect to achieve stronger performance over time than any other hyperparameter tuning protocol.
>
> ---
>
> ### Additional Feedback
>
> > The scope of TabArena’s initial study is very large
>
> We appreciate the recognition of our efforts to make the benchmark more accessible to users with limited computational resources, as well as the suggestions on how we could further improve on that.
> Furthermore, we would like to highlight that, in addition to TabArena-Lite, we do not require future users to run 200 configurations for all datasets (to publish or develop their models), thereby further reducing the usage costs. Users can decide for themselves how much computation they invest. This decision will be further improved by our new plots (mentioned above), which can help to compare models at different tuning budgets.
> Furthermore, we began investigating approaches to identify representative subsets of the datasets (similar to [3]), but this requires more research.
>
> > environmental impact
>
> We are regrettably aware of the negative environmental impact of our benchmark. We had several internal discussions about trading off environmental impact and research, but we have not converged on an official conclusion to this (philosophical) topic.
> To share some insights from our discussions, we provide several thoughts about the positive environmental impact of TabArena below:
>
> * We save and share the predictions (among other metadata) of all models so that others can avoid wasting energy on rerunning our experiments to obtain the predictions for future studies.
> * We only simulate post-hoc ensembling on the saved predictions and thus save the computation overhead that could come from post-hoc ensembling, cf. [4,5,6].
> * We impose time limits on the training time of a model (per split). Thus, we avoid running configurations of models that would potentially take a very long time to converge, without improving predictive performance. A tighter time limit would save more energy.
> * The large (one-time) cost of running TabArena enables us to find more efficient and better models. Moreover, TabArena enables us to identify portfolios that improve the Pareto frontier in terms of both quality and efficiency.
> These portfolio configurations can be (and already are) implemented in predictive machine learning systems widely used in industry (e.g., AutoGluon), with applications that result in compute usage far exceeding that of the TabArena benchmark.
> As a result, the negative environmental impact of TabArena may be offset through its future applications, resulting in a net reduction of compute usage and a positive environmental impact.
>
> We plan to expand this to a more comprehensive discussion in the appendix of the manuscript.
>
> ---
>
> Thank you for providing valuable feedback that enabled more insightful analyses.
>
> Kind regards,
>
> The Authors
>
> **References:**
> * [1] Ye, Han-Jia, et al. "A closer look at deep learning methods on tabular datasets." arXiv preprint arXiv:2407.00956 (2024).
> * [2] Tschalzev, Andrej, et al. "Unreflected use of tabular data repositories can undermine research quality." arXiv preprint arXiv:2503.09159 (2025).
> * [3] Aitchison, Matthew, Penny Sweetser, and Marcus Hutter. "Atari-5: Distilling the arcade learning environment down to five games." International Conference on Machine Learning. PMLR, 2023.
> * [4] Purucker, Lennart, et al. "Q (D) O-ES: Population-based quality (diversity) optimisation for post hoc ensemble selection in AutoML." International Conference on Automated Machine Learning. PMLR, 2023.
> * [5] Purucker, Lennart, and Joeran Beel. "CMA-ES for post hoc ensembling in AutoML: a great success and salvageable failure." International Conference on Automated Machine Learning. PMLR, 2023.
> * [6] Purucker, Lennart, and Joeran Beel. "Assembled-OpenML: creating efficient benchmarks for ensembles in AutoML with OpenML." arXiv preprint arXiv:2307.00285 (2023).
> * [7] Schneider, Lennart, Bernd Bischl, and Matthias Feurer. "Overtuning in Hyperparameter Optimization." arXiv preprint arXiv:2506.19540 (2025).

---

> ### Comment · Area_Chair_vZEz · 2025-08-04
> **Follow-up on rebuttal**
>
> Dear Reviewer txew,
>
> Has the author’s response adequately addressed the points you raised?
>
> As a reminder, the author-reviewer discussion period ends on August 6. If you have further comments or questions, please post them in the relevant thread.
>
> Many thanks,
>
> Area Chair

---

### Official Review · Reviewer_dAmV · 2025-06-27

**Rating:** 6
**Confidence:** 4

**Summary:**

This paper proposes **TabArena** as a novel benchmark for supervised machine learning on tabular data.
The main contributions of this work is (a) the selection of 51 datasets and the (b) creation initialization of a leaderboard with performances of modern tabular models.

**Additional Feedback:**

### Question 1: Limiting Foul Play by Model Providers
Since TabArena is a new benchmark designed to be used in the future als with tabular foundation models (which can be trained/fine-tuned on closed data) in mind.
Then I think it would be important to discuss if or how the benchmark can be used in a way that limits foul play on the model provider side.
Recently, the Llama 4 Maverick model was scrutinized for "cheating" on ChatbotArena.
Do you think there is any way that TabArena could prevent such behavior?
I feel that with the advent of tabular foundation models, tabular ML will soon face similar difficulties like the language or vision domain in assessing "state-of-the-art".

**Note:**
This is quite of an open-endend question, but I think it would be very important to touch on this topic in the paper and hear the authors' thoughts on this.
I do not think that the responsibility here lies with the benchmark providers, but rather with the model providers, which is why this point should not be consiererd a weakness of this paper the paper.

### Q2 Sources of Uncertainty in plots
Can you list all the sources of uncertainty that create the confidence bands in your plots (Figure 1, 5, 6, etc.)?
Of course the training, but also the evaluation with ELO, right?
Having a table of all "randomness" sources would be helpful.

### Q3: Comment on W1
Can you comment on the W1 weakness?

**Dataset Code Accessibility:**

Yes

**Dataset Code Comments:**

Selecting "Yes" here was surprisingly hard for me, since I actually do had some problems and challenges getting the code and data to set up.
However, I am very confident that this is merely an artifact of the project being under review and under active development.
After installing tabarena via uv with the goal of exploring the library I soon realized that this was an empty repository and that all listings actually read from "tabrepo import ...".
The library is then split into multiple subprojects (_tabrepo_, _tabarena_dataset_curation_, _tabarena_benchmarking_examples_) which felt quite confusing to start with.
The fact that the documentation leads only to a Gradio site and not to a technical API documentation complicated this further.

This being said, why I selected yes:
The datasets are carefully curated and collected in a well-organized project (in _tabarena_dataset_curation_ each dataset comes with a loading/preprocessing/checking script and important metadata).
The datasets are selected without licensing issues and come with scientific usage rights.
Hence, while using the benchmark right now feels a bit awkward to use, I do not see any glaring issues of accessing and re-evaluating the benchmark in the future with everything build around it.

**Ethical Considerations:**

No, there are no or only very minor ethics concerns

**Final Justification:**

This paper is a strong submission to NeurIPS DB track that probably deserves a highlight at the conference. The response by the authors makes the submission even stronger. My most important point regarding ELO was addressed. My open-ended question regarding foul-play was also addressed.

**Limitations Weaknesses:**

The paper is very strong, but I see two weaknesses and limitations that should be addressed:

### W1: Comparison to ChatbotArena
The authors compare their work to ChatbotArena.
I get that TabArena is a catchy name with good recognition potential, however, I wonder if the comparison to ChatbotArena is actually helping the paper.
ChatbotArena is a "messy" tool to do benchmarks in a domain where doing good evaluations is very hard with a lot of drawbacks.
In the tabular domain this is not the case: evaluation is easy and well understood.
In ChatbotArena models are "battling" against each other resulting in pairwise comparisons.
The pairwise comparisons are then aggregated into a ranking / leaderboard.
Yet, these pairwise comparison are much more noisy and less reliable performance signals than traditional evaluation with metrics such as AUROC or RMSE (used in TabArena).
The authors take the idea of using ELO ranking as a basis for their leaderboard.
This feels like artificially limiting the evaluation just to be able to be like ChatbotArena.
I know that the authors also store and present the raw evaluation metrics (Appendix A.1, Line 169, ...), but I would have liked to see a more substation discussion of why ELO ranking should be the default choice and/or is better suited than just using the raw evaluation metrics.
To finish this point and relating TabArena back to the LLM domain, I think TabArena might actually be more similar to something like RewardBench (you upload a new reward model and see if it performs better/worse than the rest) than ChatbotArena.

#### References:
- RewardBench: https://arxiv.org/abs/2403.13787

### W2: Onboarding new Users (Documentation)
While the appendix shows how new users could code up new models to the benchmark, the paper still is quite vague (or I missed it ... the paper is quite long) on how to actually run and evaluate the models with the benchmark.
There are a lot of listings showing important steps, but I would have liked to see more documentation / examples on how to work with the library.
I would expect a software library of this scale to have a more detailed technical documentation (e.g., API documentation) instead of just a Gradio site.
This is not necessarily a weakness of the paper, but rather a limitation of the current state of the library, which should be addressed.

**Strengths Contributions:**

In my opinion, the paper is a very strong contribution to the benchmark track with a lot of contributions that are very relevant to the ML community as a whole. A couple of contributions that are worth highlighting:

### Novelty and Relevance
The paper proposes a novel benchmark for supervised machine learning on tabular data.
The work is very timely and relevant, as the field of tabular data is currently experiencing a resurgence in interest, after the success of tabular foundation models.
Therein, the evaluation on this benchmark paints a very much needed picture of the current state-of-the-art.

### Dataset Selection
The datasets are carefully selected and curated.
The filtering criteria are well-defined.
I second the authors' choice in forgoing the creation of very large benchmarks with many but often times poor-quality datasets and instead focus on a smaller set of high-quality and real-world datasets.
Looking through _tabarena_dataset_curation.dataset_creation_scripts.datasets_ shows a very good selection of datasets (often from kaggle) that are derived from real-world problems.

### Running the Benchmark (Leaderboard Initialization)
The authors provide a very good initialization of the leaderboard with results from many modern tabular models.
The models include both classical models (e.g., XGBoost, LightGBM), baselines such as KNN or Linear, but also modern neural network foundation models such as TabPFNv2 or TabICL.
Modeling includes hyperparameter optimization (tuning) and ensembling.
This is a very important contribution to the community which shows that deep learning models like the authors put it "caught up" to classical models.
Especially when ensembled.
The effort put into running all of these experiments must be commended!

### Good Presentation, Quality of the Paper, and Quality of the Software
The quality of the paper is very high.
It's well-structured, easy to read, and the figures are of high quality.
The design choices are well documented and presented.
The technical appendix is very detailed and provides a lot of additional information (more detailed evaluation, how to run the code, reproducibility, etc.).
With minor issues/weaknesses, the software is also well polished.

### TabArena Ensemble
As another contribution, the authors post-hoc bundle the results (models) from the leaderboard into a TabArena ensemble.
This is quite interesting, as it shows that the ensemble outperforms all individual models on the benchmark (+ the AutoGluon 4h baseline).
This offers a very good starting point for future work with the benchmark and model ensembling.
The average weight plot of the ensemble is a good first-step towards analyzing what contributes to the performance of the ensemble, but other methods from the XAI literature could be applied here as well (e.g. Shapley values) with more potential insights.
This shows how layered the contributions of the paper are.

### Subsets of the Benchmark
The authors provide sensible cuts of the benchmark into subsets that allow for different use cases:
The evaluation of TabPFN in small data regimes or TabICL in medium data regimes.

---

> ### Author Rebuttal · Authors · 2025-07-30
>
> Dear dAmV,
>
> We sincerely thank you for the valuable suggestions and insights provided. We reply to W1/Q3, Q1, and Q2 below.
>
> We see W2 and the “Dataset Code Comments” as important and are actively working on improving the documentation, with a focus on creating a technical API / onboarding page. We were already able to refine the onboarding process a bit in collaboration with the first users from the community.
>
> ---
>
> ### W1/Q3: Comparison to ChatbotArena & Elo
>
> We have updated the related work section in accordance with your assessment of the differences between TabArena and ChatbotArena. We fully agree with this assessment. In detail, we clarified how we differ from ChatbotArena and in how far tabular data is a distinct evaluation domain. Moreover, we added several more references to other arena-like efforts, including RewardBench, and clarified the relationships to our work.
>
> Regarding Elo:
> We want to give a clear ranking of models as part of the leaderboard. Thus, we decided on one “main” metric while allowing users to re-sort the leaderboard according to several other metrics. This was not a trivial decision as part of the evaluation design because every aggregation metric has its own pitfalls, and none is perfect. However, we have now realized, based on your feedback, that Section 2.3 conveys (wrongly) that we blindly followed ChatbotArena. We addressed this issue in the updated manuscript. We summarize the discussion for why we slightly favor Elo as the main metric below.
>
> Elo is based on pairwise comparison scoring, which only considers wins, ties, or losses, and neglects the magnitude of performance differences. This can be a disadvantage, since minor performance differences could be considered irrelevant by practitioners in specific applications. At the same time, this means that each dataset contributes equally to Elo, hence the aggregation is not biased towards certain domains or dataset properties (e.g., small or non-noisy datasets). For TabArena, this is a key advantage because we want to create a benchmark whose results are representative of all domains and datasets. In general, we expect Elo to be more robust towards choices in dataset selection, and it has more statistical power (as seen by the small confidence intervals).
>
> To add a complementary point of view to the main paper, we will add plots with improvability, a metric that measures the performance of methods relative to the best method, and is therefore sensitive to the magnitude of performance differences. Specifically, we plan to add two new figures in the main text: 1) showing the Pareto front of Improvability and inference time, and 2) showing the performance over (tuning) time related to Improvability.
> Improvability is an aggregation that shows the relative reduction in a metric, thereby providing an overview of the magnitude of differences between models.
>
> Lastly, we would like to note that we are open to revising the metric in future versions of the leaderboard as part of our living benchmark. However, we first need more research on evaluation metrics (and statistical tests) to reach a constructive conclusion. We view TabArena as being very useful in this research, and it may inspire further research on metrics and tabular ML evaluation, similar to what ChatbotArena did for Elo and LLM evaluation.
>
> ---
>
> ### Q1 Limiting Foul Play by Model Providers
>
> We have added an extended discussion about foul play to the appendix. Here is a shorter version of this discussion:
>
> We think foul play will inevitably affect TabArena as well. We, as maintainers, are very aware of this problem. As maintainers, we have considered future guards against this in four ways:
> 1. A simple “solution” is to provide leaderboards excluding models with potential contamination or foul play. Additionally, we can still provide a leaderboard that includes all models.
> 2. We will generally investigate reasons for better (or worse) performance per dataset, given model outliers. Given that model providers can train on TabArena’s datasets, we expect that for LLM-based approaches, model providers should do memorization tests at all times [1]. For tabular foundation models, we currently have no way of detecting contamination and do not know if such models can “remember” the data in a significant way, as is the case for text- or vision-based modelling. Here, future research might help.
> 3. One big difference with ChatbotArena and its problems with limiting foul play is that our official leaderboard requires an open mode (code, data, result artifacts), which will make the abuse mentioned harder and easier to spot by the community. We do not want to rule out benchmarking API-based closed-source models in the future, so this might also not be a silver bullet.
> 4. Lastly, we believe that the living benchmark itself will detect foul play across future iterations as new datasets, potentially changing (the seed of) the dataset splits, or new methods to detect foul play will also inevitably be added to the TabArena ecosystem.
>
> ---
>
> ### Q2 Sources of Uncertainty
>
> We have added a discussion on the source of randomness to the appendix.
> In short, to the best of our knowledge, the following sources exist:
>
> * Model Randomness, resulting from: initialization, training, non-deterministic computations (on GPU or due to precision), inner validation splits (e.g., for early stopping), hyperparameter configurations, and the sampling of the hyperparameter configurations.
> * Data Randomness, resulting from: the selected datasets and splits used for outer repeated cross-validation.
> * Evaluation Randomness, resulting from: metric calculation (such as Bradley-Terry Elo with bootstrapping), and the precision of calculating metrics when the metric is used for ranking or normalization.
>
> ---
>
> Thank you for inciting extended discussions that improve the manuscript.
>
> Kind regards,
>
> The Authors
>
> **References:**
> * [1] Bordt, Sebastian, et al. "Elephants never forget: Memorization and learning of tabular data in large language models." arXiv preprint arXiv:2404.06209 (2024).

---

> > ### Comment · Reviewer_dAmV · 2025-08-04
> > **Thank you to the authors!**
> >
> > I **thank** the authors for their response! I appreciate the discussions and improvements to the manuscript.
> >
> > > For TabArena, this is a key advantage because we want to create a benchmark whose results are representative of all domains and datasets. In general, we expect Elo to be more robust towards choices in dataset selection, and it has more statistical power (as seen by the small confidence intervals).
> >
> > I think so too. Another variant of this aggregation metric could also incorporate a softer (potentially even variable) _tie_ mechanism, where epsilon differences in performance are seen as ties in the aggregation. Furthermore, I also think that looking at _win-rates_ between the top-models could also be interesting.
> >
> > Ultimately, as the authors also have put it, the **living** aspect of the benchmark **is** and **will be** a very important contribution of this work. Keep up the good work!

---

### Official Review · Reviewer_SJuG · 2025-06-30

**Rating:** 4
**Confidence:** 3

**Summary:**

This paper proposes TabArena, the first dynamically updated benchmark testing platform for tabular data, featuring continuous maintenance mechanisms and public leaderboards. By selecting 51 representative tasks from real-world scenarios, it constructs a benchmark covering small-to-medium real-world scenarios for unified evaluation of 16 models, encompassing tree models, deep networks, and foundation models' performance on tabular data. Through 25 million training runs and carefully designed evaluation protocols, this paper presents a standardized testing scheme that is scalable for the entire community. Additionally, the authors provide public leaderboards, scoring systems, rich metadata, and standardized evaluation interfaces, offering a transparent, reproducible, and continuously evolving evaluation platform for tabular learning.

**Additional Feedback:**

Compared to previous tabular data benchmarks such as OpenML-CC18 and TALENT, this paper has three core advances:

1. Introduces a "dynamic update + open community participation" mechanism, breaking the closed nature of static benchmarks;
2. Constructs a systematic evaluation framework based on large-scale experiments and standard protocols, with significant reproducibility;
3. Emphasizes ensemble perspectives rather than single-model optimality, bringing more reasonable performance evaluation approaches to tabular learning research.

Openness, sustainability, and community-driven nature are the key features that distinguish this platform from traditional static benchmarks.

**Dataset Code Accessibility:**

Yes

**Dataset Code Comments:**

No special comments

**Ethical Considerations:**

No, there are no or only very minor ethics concerns

**Final Justification:**

As authors have addressed my concerns, I will keep this positive score.

**Limitations Weaknesses:**

1. Data task selection: While strict dataset filtering criteria enhance the representativeness of real-world tasks, they also significantly reduce the number of datasets and only consider the ideal case of independent and identically distributed data.

2. Chart design lacks focus: Although Figures 2-6 are numerous, they fail to form clear comparative logic and struggle to support conclusions. Most only display Elo scores or rankings without providing common metrics like AUC or Accuracy.

3. Tuned + Ensemble is a major highlight of the paper, but it appears more like a post-hoc trick, lacking systematic modeling logic, and fails to explain how different model weights are learned in the ensemble.

4. In the last part of Chapter 3, the authors argue that practitioners generally prefer using GBDT over deep learning methods, primarily because deep learning methods often have inferior code quality and maintainability. This argument lacks supporting evidence.

**Strengths Contributions:**

1. Filling research gaps: This paper addresses the lack of systematic benchmark evaluation and unified evaluation platforms in the tabular data field, which has extremely high practical significance.

2. Strong systematicity and thorough explanation: From model selection, data filtering, ensemble strategies to scoring systems, the design is reasonable and comprehensive in coverage.

3. Open-source friendly with complete toolchain: APIs, code documentation, and evaluation scripts are clear and user-friendly, making it easy for community adoption and extension.

4. New research perspective: Past benchmark research focused too much on which single model is better, while this paper is dedicated to finding models that perform better in ensembles, rather than individual models that beat GBDT.

---

> ### Author Rebuttal · Authors · 2025-07-30
>
> Dear SJuG,
>
> Thank you for your thoughtful feedback. We respond to your comments on Limitations and Weaknesses below.
>
> ---
>
> ### 1) Data task selection
> We fully agree that the focus on IID data and the resulting reduction in the number of datasets is a limitation. However, we had to make decisions to keep the scope of the paper focused. For incorporating non-iid datasets, several conceptual changes would be required, such as defining and evaluating suitable validation and ensembling protocols. Similarly, since we have already assessed over 1,000 datasets, finding more suitable datasets that have not been used in prior work would be a significant contribution in itself. Therefore, we decided to narrow the scope and leave the mentioned extensions for future work.
>
> ---
>
> ### 2) Chart design
> To be precise, let us distinguish between metrics (such as ROC AUC / logloss / RMSE, computed on a single dataset) and aggregations (such as Elo, Rank, or any other method for aggregating these per-dataset metrics into a single score).
> We compute metrics, such as ROC AUC, and then aggregate per-dataset results using ELO or mean improvability (see Section 2.3). Besides ELO, we provide figures with additional aggregations and the full leaderboard (in the supplements: Appendix A1 and Table A.1). We agree that providing common metrics is important. Therefore, we present the metric results for ROC AUC, logloss, and RMSE at the dataset level in the supplements, Appendix E.
>
> To add a complementary point of view next to Elo in the main paper, we will add plots with improvability, a metric that measures the performance of methods relative to the best method, and is therefore sensitive to the magnitude of performance differences. Specifically, we plan to add two new figures in the main text: 1) showing the Pareto front of Improvability and inference time, and 2) showing the performance over (tuning) time related to Improvability.
> Improvability is an aggregation that shows the relative reduction in a metric of the best method compared to the current method, thereby providing an overview of the magnitude of differences between models.
>
> > Although Figures 2-6 are numerous, they fail to form clear comparative logic and struggle to support conclusions.
>
> We struggle to understand the criticism of this point. Could you clarify what you mean?
> Figures 2 and 3 describe our benchmark datasets. Figures 4-6 are used directly in our text for comparisons and to support conclusions across subsections of the results.
>
> ---
>
> ### 3) Tuned + Ensemble
> The post-hoc ensembling logic we employ to obtain Tuned + Ensemble is, to the best of our knowledge, the SOTA for ensembling tabular machine learning models after tuning. We use post-hoc ensembling and thereby follow prior work on ensembling, Kaggle expertise, and SOTA predictive machine learning systems. We concur that we did not sufficiently explain the ensembling method in the paper and added the following explanation to the appendix.
>
> **Tuned + Ensemble Explanation**
>
> Post-hoc ensembling (PHE) aims to combine a set of models previously evaluated on holdout data or with cross-validation during model selection (e.g., HPO) to improve performance over any single model [2,3]. In particular, PHE relies on using data collected while evaluating models to build its ensemble, such as predictions on the validation data.
>
> In practice, predictive machine learning systems most often [3] combine a set of models by aggregating their predictions with a weighted arithmetic mean whereby the weights of the models are commonly obtained using greedy ensemble selection (GES) [4,5]. Likewise, multiple hyperparameter configurations of an individual tabular model can be ensembled with GES, as done in TabRepo [6] or by TabPFN v2 [7].
>
> Post-hoc ensembling with GES has four key advantages: (1) GES is very efficient due to reusing predictions on validation data previously collected while evaluating models [3, 8]; (2) GES optimizes a user-defined target metric using an anytime algorithm; (3) the final ensemble is usually small since GES produces sparse weight vectors [3, 9]; 4) the predictive performance of post-hoc ensembling with GES is superior to the best individual model under mild assumptions [3, 8, 9, 10].
>
> We build an ensemble of models using GES. In detail, we create an ensemble using the 200 hyperparameter configurations that were evaluated during the tuning process. To train the ensemble, we reuse the predictions on validation data that were computed during (inner) cross-validation. Then, we obtain a weight vector using GES for all configurations. Finally, we return the weighted average predictions of all non-zero-weighted configurations.
>
> GES learns a weight vector $W = (w_1, ..., w_m)$ to combine multiple models $f_i \in F$ from a pool of $m$-many models $F = (f_1, ..., f_m)$ as $\sum_i w_i f_i$. GES ensures that $\forall i, 0 \leq w_i \leq 1$ and $\sum_i w_i = 1$. The vector $W$ is learned via a greedy algorithm that runs for a fixed number of iterations $N$ ($N=40$ for TabArena). In each step $n \leq N$, GES finds $i$ such that increasing $w_i$ by $\frac{1-w_i}{n+1}$ and decreasing all other weights by $\frac{w_j}{n+1}, \forall\, j \neq i$ most reduces the validation error.
>
> ---
>
> ### 4) Supporting evidence
> We thank you for pointing out that our hypothesis lacked supporting evidence. We added this statement as our (subjective) perception, but did not adequately express why we perceived it. Thus, we will update the section with the corresponding supporting evidence. PyTorch Tabular [11], a significant effort to reimplement many of the older deep learning methods, also remarks on the “lack on the software side of things” as driving motivation. Furthermore, when surveying the popularity (in terms of GitHub stars) and the maintenance of various implementations, we also obtain a clear picture where older packages are not actively maintained:
>
> * Examples for tree-based models: XGBoost (27k, active), CatBoost (9k, active), LightGBM (17k, active), EBM (7k, active)
> * Examples for old deep learning models: FT-Transformer (400 across the official and various re-implementations, inactive), PyTorch Tabular (2k, semi-active to inactive), TabNet (3k, inactive), many other published models do not have officially maintained code or are re-implemented in PyTorch Tabular.
> * Examples for new deep learning models: TabPFN (4k, active), TabM (600, active), RealMLP (200, active)
>
> In addition to adding the reference, we will also mention alternative hypotheses in the paper:
> >  Finally, we hypothesize that a major reason for practitioners to rely on GBDTs instead of deep learning is the insufficient code quality and maintenance of deep learning methods compared to GBDT frameworks [11]. TabArena also tackles this problem by wrapping existing deep learning methods in an easy-to-use interface, see Appendix D.5. However, other reasons may also contribute to the slow adoption of deep learning methods, such as longer training time (especially on CPU), the lack of some functionalities, habit, the existence of more educational material for GBDTs, the large number of different deep learning methods, and overclaims of deep learning in academic papers.
>
> ---
>
> Thank you again for your valuable feedback!
>
> Kind regards,
>
> The Authors
>
> **References:**
>
> * [1] Provost et al., "The case against accuracy estimation for comparing induction algorithms." ICML. Vol. 98. 1998.
> * [2] Shen, Yu, et al. "Divbo: Diversity-aware CASH for ensemble learning." Advances in Neural Information Processing Systems 35 (2022): 2958-2971.
> * [3] Purucker, Lennart, et al. "Q (D) O-ES: Population-based quality (diversity) optimisation for post hoc ensemble selection in AutoML." International Conference on Automated Machine Learning. PMLR, 2023.
> * [4] Caruana, Rich, et al. "Ensemble selection from libraries of models." Proceedings of the twenty-first international conference on Machine learning. 2004.
> * [5] Caruana, Rich, Art Munson, and Alexandru Niculescu-Mizil. "Getting the most out of ensemble selection." Sixth International Conference on Data Mining (ICDM'06). IEEE, 2006.
> * [6] Salinas, David, and Nick Erickson. "Tabrepo: A large scale repository of tabular model evaluations and its AutoML applications." arXiv preprint arXiv:2311.02971 (2023).
> * [7] Hollmann, Noah, et al. "Accurate predictions on small data with a tabular foundation model." Nature 637.8045 (2025): 319-326.
> * [8] Feurer, Matthias, et al. "Efficient and robust automated machine learning." Advances in neural information processing systems 28 (2015).
> * [9] Purucker, Lennart, and Joeran Beel. "CMA-ES for post hoc ensembling in AutoML: a great success and salvageable failure." International Conference on Automated Machine Learning. PMLR, 2023.
> * [10] Purucker, Lennart, and Joeran Beel. "Assembled-OpenML: creating efficient benchmarks for ensembles in AutoML with OpenML." arXiv preprint arXiv:2307.00285 (2023).
> * [11] Joseph, Manu. "Pytorch tabular: A framework for deep learning with tabular data." arXiv preprint arXiv:2104.13638 (2021).

---

> > ### Comment · Reviewer_SJuG · 2025-08-03
> >
> > The authors have in general resolved my questions, I hope these discussions can be added in the revised version.

---

### Decision · Program_Chairs · 2025-09-18

**Decision:**

Accept (spotlight)

**Comment:**

**Summary**. This paper introduces TabArena, a continuously maintained “living” benchmark for tabular machine learning. It curates a collection of datasets and tabular models, and establish the initial public leaderboard through an initial large-scale evaluation.


**Strengths**:
- *Filling an important research gap* (*SJuG*, *dAmV*, *txew*, *NWc4*). The paper addresses a major gap in tabular ML research, given that (i) existing benchmarks have limitations in dataset curation and performance evaluation, and (ii) there is renewed interest in benchmarking tabular ML models.
- *Systematicity* (*SJuG*, *dAmV*, *txew*, *NWc4*).
  - The benchmark achieves extensive breadth, curating 51 datasets from an initial pool of 1,053 candidates through a manual selection process.
  - The benchmark uses sensible and robust evaluation protocols.
  - The leaderboard is initialized with comprehensive results from 16 models, spanning tree-based approaches, neural networks, and tabular FMs.
- *Reproducibility/Accessibility* (*SJuG*, *txew*, *NWc4*). The benchmark includes a public leaderboard (as well as a light version), precomputed predictions, high-quality reproducible code, and well-established maintenance protocols.
- *Significance of paper's highlights* (*SJuG*, *dAmV*, *txew*, *NWc4*). The experiments deliver significant findings: (i) the benefits of ensembling, (ii) the value of model diversity in ensembling, and (iii) the identification of conditions under which each model family performs best.
 - *Clarity of presentation* (*dAmV*, *txew*, *NWc4*). The paper and its appendix are clearly written, well-structured, and include extensive documentation.

**Weaknesses**:
- *Limitations in data selection* (*SJuG*, *NWc4*). The benchmark currently focuses on small- and medium-sized i.i.d. datasets. The authors acknowledge this limitation and mention plans to collaborate with the community to broaden the benchmark's scope.
- *Limitations in the ELO metric* (*dAmV*, *NWc4*). Two reviewers expressed reservations regarding the choice of ELO metric, which only accounts for signs of performance differences and not magnitudes. In response, the authors argue that ELO is expected to be robust to dataset selection. They also indicate plans to complement their results with the “improvability” metric.


**Additional questions**:
- Have the authors considered using metrics like accuracy instead of log-loss, which may be penalizing overconfidence in overparameterized NNs?
- Could the mismatch between individual model rankings and their ensemble weights in Figure 6 arise from spreading weights across similar models within the same family, so that the ensemble maximizes overall diversity?

**Justification**. The reviews were unanimously positive, recognizing that this work is of high quality and is likely to have high impact in the tabular ML community. It certainly deserves to be highlighted at NeurIPS D&B 2025.